# Contextual Bandits with Knapsacks beyond Worst Cases via Re-Solving

## Abstract

Contextual Bandits with Knapsacks (CBwK) is a fundamental and essential framework for modeling a dynamic decision-making scenario with resource constraints. Under this framework, an agent selects an action in each round upon observing a request, leading to a reward and resource consumption that are further associated with an unknown external factor. The agent's target is to maximize the total reward under the initial inventory. While previous research has already established an $\widetilde{O}(\sqrt{T})$ worst-case regret for this problem, this work offers two results that go beyond the worst-case perspective, one for worst-case locations, and another for logarithmic regret rates. We start by demonstrating that the unique-optimality and degeneracy of the fluid LP problem, which is both succinct and easily verifiable, is a sufficient condition for the existence of an $\Omega(\sqrt{T})$ regret lower bound. To supplement this worst-case location result, we merge the re-solving heuristic with distribution estimation skills and propose an algorithm that achieves an $\widetilde{O}(1)$ regret as long as the fluid LP has a unique and non-degenerate solution. This condition is mild as it is satisfied for most problem instances. Furthermore, we prove our algorithm maintains a near-optimal $\widetilde{O}(\sqrt{T})$ regret even in the worst cases, and extend these results to the setting where request and external factor are continuous. Regarding information, our regret results are obtained under two feedback models, respectively, where the algorithm accesses the external factor at the end of each round and at the end of a round only when a non-null action is executed.

## 1 Introduction

In the contextual bandits problem with knapsack constraints (CBwK problem for short), an agent is required to make sequential decisions over a finite time horizon to maximize the accumulated reward under initial resource constraints. To be more specific, in each round $t = 1, \cdots, T$, a request $\theta_t$ and an external factor $\gamma_t$ are independently generated from two distributions, and only $\theta_t$ is revealed to the agent. Based on the request, the agent should irrevocably choose an action $a_t$, which would result in a reward $r(\theta_t, a_t, \gamma_t)$ and a consumption vector $\boldsymbol{c}(\theta_t, a_t, \gamma_t)$ of resources. The agent's target is to optimize the sum of rewards $\sum_{t=1}^{T} r(\theta_t, a_t, \gamma_t)$ before the resources are depleted.

The CBwK problem presents two key challenges when compared to closely related problems (e.g., network revenue management problem) : (1) choices are made without observing external factors, and (2) distributions of requests and external factors are *unknown*. However, the complexity of CBwK makes it a suitable mathematical abstraction for many real-life scenarios, such as dynamic bidding in repeated second-price auctions with budgets [Balseiro et al., 2021, Balseiro and Gur, 2019]. In this circumstance, an advertiser (the agent) acquires the value of the ad slot (the request) at the start of each auction, and would choose a bid (the action) accordingly. The agent's utility and payment in

this auction, as a consequence, are collaboratively determined by the value, the bid, and the highest competing bid (the external factor). It is to be noted that the highest competing bid is inaccessible to the agent before committing to the bid, as all advertisers bid simultaneously. Meanwhile, its distribution is decided by other advertisers, which is also unknown to the agent before the auctions. The CBwK model can also capture other well-discussed problems including multi-secretary, online linear programming, online matching, as discussed in Balseiro et al. [2021].

Previous studies of the CBwK problem have shown that the worst-case regret of any online strategy is $\widetilde{O}(\sqrt{T})$ when the initial resources are linearly proportional to the horizon length $T$ [Slivkins and Foster, 2022, Han et al., 2022]. [1] However, it is still unclear where worst-case scenarios occur, meaning under which condition(s) an $\widetilde{\Omega}(\sqrt{T})$ regret is inevitable. Furthermore, we do not know whether we can achieve a better regret guarantee for the CBwK problem beyond worst-case scenarios. In particular, can we design algorithms to obtain an $o(\sqrt{T})$ regret only under mild assumptions that hold for almost all possible CBwK instances? This work takes the first step in addressing these questions.

## 1.1 Our Contributions

This work mainly makes three contributions, summarized as follows.

**A precise sufficient condition for an $\widetilde{\Omega}(\sqrt{T})$ regret lower bound.** To move beyond worst-case analysis, we establish a precise sufficient condition for the $\widetilde{\Omega}(\sqrt{T})$ regret lower bound to hold. Specifically, we demonstrate that when the fluid benchmark (also known as the deterministic LP) has a unique and degenerate solution, then an $\Omega(\sqrt{T})$ regret is unavoidable for any online decision strategy (Theorem 2.1). While Han et al. [2022] have also provided a regret lower bound result for the CBwK problem, their condition depends on the inseparability of the possible expected reward/consumption function set. In other words, their condition may not perform well when this feasible set is small. Furthermore, their condition is rather complicated to verify. In contrast, our condition only depends on the underlying problem instance, is concise, and is easy to check. The proof of our result extends the approach of Vera and Banerjee [2021] to the CBwK problem.

**An $\widetilde{O}(1)$ regret via re-solving under mild assumptions with full/partial information feedback.** With the above result, we investigate how well an online algorithm can perform beyond worst cases, by applying the re-solving heuristic in conjunction with distribution estimation techniques, as given in Algorithm 1. Although this method has been considered in the problem of bandits with knapsacks (BwK) [Flajolet and Jaillet, 2015], to the best of our knowledge, we are the first to extend this method to the CBwK problem, which poses new challenge as decisions should be made according to the request. To avoid worst cases, we explicitly suppose that the fluid problem has a unique and non-degenerate solution (Assumption 3.1). This assumption is mild in three aspects: (1) it captures almost all CBwK problem instances, as slightly perturbing any LP can help it satisfy the unique optimality and non-degeneracy conditions; (2) it is almost necessary for an $o(\sqrt{T})$ regret bound to establish by Theorem 2.1 as we just discussed, only left the case that the fluid problem has multiple optimal solutions; and (3) it is far less restrictive than the assumptions given in Sankararaman and Slivkins [2021], which require that there are at most two resources and the best-arm optimality, and almost surely excludes all problem instances. Under the assumption, our main results show that the re-solving heuristic reaches an $O(1)$ regret with full information (Theorem 3.1) and an $O(\log T)$ regret with partial information (Theorem 4.1). To our knowledge, these are the first $\widetilde{O}(1)$ regret results in the CBwK problem beyond the worst case with only mild assumptions. Importantly, these regret bounds are also independent to the number of actions, unlike previous results.

Within our results, the full information model assumes that the agents sees the external factor at the end of each round, while in the partial information model, the agent acquires the external factor only when a non-null action is adopted. Other state-of-the-art results consider bandit information feedback, in which the agent only sees the reward and the consumption rather than the external factor.

---

[1]In this work, a strategy's regret is defined as the gap between its expected total reward and the fluid benchmark (to be introduced in Section 2), which has known to be an upper bound of the former. Such a definition is implicitly yet widely adopted in the literature [Slivkins and Foster, 2022, Han et al., 2022, Sivakumar et al., 2022].

However, they explicitly assume a specific (e.g., linear) relationship between the conditional expected reward-consumption pair and the request [Agrawal and Devanur, 2016, Sankararaman and Slivkins, 2021, Han et al., 2022, Slivkins and Foster, 2022], whereas our results do not impose any underlying distribution structures. On this side, our information model are comparable to those in existing work.

**A near-optimal regret even in worst cases with full/partial information feedback, and an extension to continuous randomness.** We further explore how well our Algorithm 1 performs even in worst-case scenarios. With full information feedback, we show that an $O(\sqrt{T \log T})$ regret is achieved (Theorem 5.1). This bound is asymptotically equal to the state-of-the-arts with this information model [Han et al., 2022, Slivkins and Foster, 2022]. Even with partial information, we can still guarantee a universal $O(\sqrt{T} \log T)$ regret (Theorem 5.2), which is optimal up to a logarithmic factor. These results demonstrate the applicability of the re-solving heuristic in CBwK problems, regardless of the specific instance. For completeness, we also extend our algorithm and analysis to the situation in which the randomness of request and external factor are continuous, and derive corresponding regret results (Theorems A.1 and A.2).

## 1.2 Literature Review

**Contextual bandits with knapsacks.** The contextual bandits with knapsacks framework was introduced by Agrawal and Devanur [2016]. Along this research line, two main methodologies have been proposed to solve the problem. The first approach aims to select the best probabilistic strategy within the policy set [Badanidiyuru et al., 2014], and Agrawal et al. [2016] adopts this approach to achieve an $\widetilde{O}(\sqrt{T})$ regret. This heuristic originates from the subject of contextual bandits [Dudik et al., 2011, Agarwal et al., 2014], and requires a cost-sensitive classification oracle to achieve computation efficiency.

On the other hand, another approach views the problem from the perspective of the Lagrangian dual space. It uses a dual update method that reduces the CBwK problem to the online convex optimization (OCO) problem. In particular, some work [Agrawal and Devanur, 2016, Sankararaman and Slivkins, 2021, Sivakumar et al., 2022, Liu and Grigas, 2022] assumes a linear relationship between the conditional expectation of the reward-consumption pair and the request-action pair. This line adopts techniques for estimating linear function classes [Abbasi-Yadkori et al., 2011, Auer, 2002, Sivakumar et al., 2020, Elmachtoub and Grigas, 2022] and combines them with OCO methods to achieve sub-linear regret. Among these works, [Sankararaman and Slivkins, 2021] shows that when there are only two resources and a best-arm, this method can obtain an $O(\log T)$ regret. Compared with their results, our assumptions are much milder, as we only assume non-degeneracy.

From another angle, depending on the difficulty of overcoming the lack of distribution knowledge on the external factor, there are two types of feedback models in the literature: full or bandit information. In the former [Liu and Grigas, 2022], the agent sees the external factor at the end of each round and can derive the reward and consumption of each possible decision in the round ex-post. Meanwhile, in the latter, the agent can only observe the reward-consumption pair brought by the decision. Apparently, the bandit information feedback is harder to deal with since less information can be accessed. Our work further considers a partial feedback model, in which the agent observes the external factor when a non-null action is chosen. This model acts as an intermediate between full and partial information feedback models.

Apart from the above work, two results [Han et al., 2022, Slivkins and Foster, 2022] concurrent with this work are not restricted to linear expectation functions. To deal with more general problems with bandit feedback, they plug model-reliable online regression methods [Foster et al., 2018, Foster and Rakhlin, 2020] into the dual update framework. As a result, the regret of their algorithms is the sum of the regret on online regression and online convex optimization, respectively. Nevertheless, the online regression technique still limits the conditionally expected reward-consumption functions.

**Network revenue management and the re-solving heuristic.** Unlike the above approaches, our work adopts the re-solving method, also known as the "certainty equivalence" (CE) heuristic. Under this approach, the agent frequently solves the fluid optimization problem with the remaining resources to obtain a probability control in each round. This method comes from the literature on the network revenue management problem, which can be seen as a simplification of the CBwK problem without the existence of external factors, or that the external factor not getting involved in the resource

137 consumption [Wu et al., 2015]. Some work in this setting also assumes known request distributions.
138 This line of research originates from Jasin and Kumar [2012], and also includes Jasin [2015], Ferreira
139 et al. [2018], Bumpensanti and Wang [2020], Li and Ye [2021], Chen et al. [2022], Besbes et al.
140 [2022]. They show that the re-solving method can obtain constant regret under certain non-degeneracy
141 assumptions and can generally obtain square root regret [Chen et al., 2022]. Recently, the re-solving
142 method is also extended to the general dynamic resource-constrained reward collection (DRCRC)
143 problem in Balseiro et al. [2021], which assumes the knowledge of request and external factor
144 distributions and achieves $O(1)$ to $O(\log T)$ regret for different action space cardinalities.

145 We should mention that the re-solving technique has also been adopted to the bandits with knapsacks
146 (BwK) problem [Flajolet and Jaillet, 2015] to achieve an $O(\log T)$ regret. However, CBwK is a more
147 challenging problem than BwK in the sense that the decision has to be made based on the received
148 request. Thus, there is no optimal static action mode that is irrelevant to the round, which adds a layer
149 of complexity to the re-solving method.

## 2  Preliminaries

151 We consider an agent interacting with the environment for $T$ rounds. There are $n$ kinds of resources,
152 with an average amount of $\boldsymbol{\rho}^i$ for resource $i$ in each round, resulting in a total of $\boldsymbol{\rho}^i T$ amount of
153 resource $i$. We suppose that $\mathbf{0} < \boldsymbol{\rho} = \boldsymbol{\rho}_1 = (\boldsymbol{\rho}^i)_{i \in [n]} \leq \mathbf{1}$ is independent of $T$, with a maximum
154 entry of $\rho^{\max}$ and a minimum entry of $\rho^{\min}$. At the beginning of each round $t \geq 1$, the agent observes
155 a request $\theta_t \in \Theta$ drawn i.i.d. from a distribution $\mathcal{U}$, and should choose an action $a_t$ from a set of
156 actions $A$. Given the request $\theta_t$ and the action $a_t$, the agent receives a random reward $r_t \in [0, 1]$ and
157 consumption vector of resources $\boldsymbol{c}_t \in [0, 1]^n$, both of which are related to an external factor $\gamma_t \in \Gamma$
158 drawn i.i.d. from a distribution $\mathcal{V}$. In other words, there is a reward function $r : \Theta \times A \times \Gamma \to [0, 1]$
159 and a consumption vector function $\boldsymbol{c} : \Theta \times A \times \Gamma \to [0, 1]^n$, such that $r_t = r(\theta_t, a_t, \gamma_t)$ and
160 $\boldsymbol{c}_t = \boldsymbol{c}(\theta_t, a_t, \gamma_t)$. We suppose these two functions are pre-known to the agent. We further define
161 $R(\theta, a) := \mathbb{E}_\gamma[r(\theta, a, \gamma)]$, and $\boldsymbol{C}(\theta, a) := \mathbb{E}_\gamma[\boldsymbol{c}(\theta, a, \gamma)]$.

162 We impose minimum restrictions on the distributions $\mathcal{U}$ and $\mathcal{V}$. Specifically, in the main body of this
163 work, we suppose that both distributions are discrete without any further assumptions. In other words,
164 $\Theta$ and $\Gamma$ are finite. We denote the mass function of $\mathcal{U}$ and $\mathcal{V}$ by $u(\theta)$ and $v(\gamma)$, respectively. We will
165 extend to the situation that these two distributions can be continuous in Appendix A.

166 The agent's objective is to maximize her cumulative rewards over the period under initial resource
167 constraints, which is a sequential decision-making problem. To ensure feasibility, we assume the
168 existence of a null action (denoted by 0) in the action set $A$. Under the null action, the reward and the
169 consumption of any resource are zero, regardless of the request and the external factor. In other words,
170 we have $r(\theta_t, 0, \gamma_t) = 0$ and $\boldsymbol{c}(\theta_t, 0, \gamma_t) = \boldsymbol{0}$ for any $(\theta_t, \gamma_t) \in \Theta \times \Gamma$. We use $A^+ := A \setminus \{0\}$ to
171 denote the set of non-null actions, and let $m := |A^+|$ be its size.

172 We consider the set of non-anticipating strategies $\Pi$. In particular, let $\mathcal{H}_t$ be the history the agent
173 could access at the start of round $t$. Then, for any non-anticipating strategy $\pi \in \Pi$, $a_t$ should depend
174 only on $\widetilde{\mathcal{H}}_t := (\theta_t, \mathcal{H}_t)$, that is, $a_t = a_t^\pi(\theta_t, \mathcal{H}_t)$. For abbreviation, we write $a_t^\pi = a_t^\pi(\theta_t, \mathcal{H}_t)$ when
175 there is no confusion.

176 Therefore, we can define the agent's optimization problem as below:

$$V^{\mathrm{ON}} := \max_{\pi \in \Pi} \mathbb{E}_{\boldsymbol{\theta} \sim \mathcal{U}^T, \boldsymbol{\gamma} \sim \mathcal{V}^T} \left[ \sum_{t=1}^T r(\theta_t, a_t^\pi, \gamma_t) \right],$$

$$\text{s.t.} \quad \sum_{t=1}^T \boldsymbol{c}(\theta_t, a_t^\pi, \gamma_t) \leq \boldsymbol{\rho} T, \quad \forall \boldsymbol{\theta} \in \Theta^T, \boldsymbol{\gamma} \in \Gamma^T.$$

177 **Benchmark.** In practice, however, computing the expected reward of the optimal online strategy
178 would require high-dimension (probably infinite) dynamic programming, which is intractable. Hence,
179 we turn to consider the fluid benchmark to measure the performance of a strategy, which is defined as

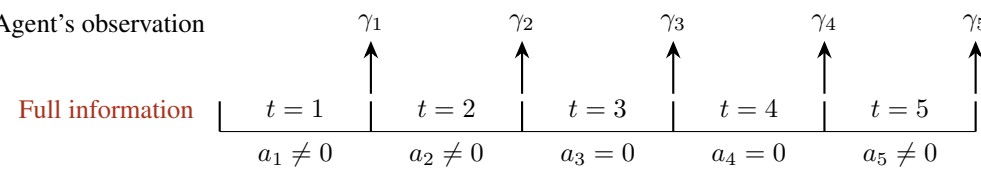

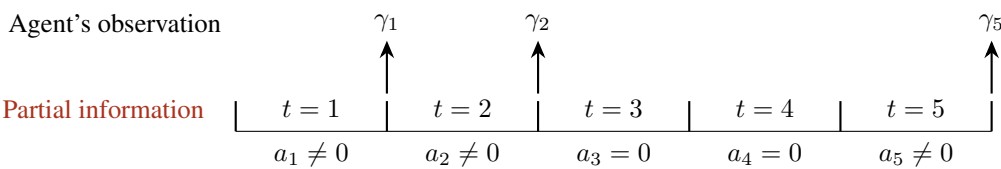

Figure 1: An illustration of the two information feedback models we consider in this work.

follows:

$$V^{\mathrm{FL}} := T \cdot \max_{\phi:\Theta \times A^+ \to \mathbb{R}} \mathbb{E}_{\theta \sim \mathcal{U}} \left[ \sum_{a \in A^+} R(\theta, a)\phi(\theta, a) \right],$$

$$\text{s.t.} \quad \mathbb{E}_{\theta \sim \mathcal{U}} \left[ \sum_{a \in A^+} \boldsymbol{C}(\theta, a)\phi(\theta, a) \right] \le \boldsymbol{\rho},$$

$$\sum_{a \in A^+} \phi(\theta, a) \le 1, \quad \forall \theta \in \Theta,$$

$$\phi(\theta, a) \ge 0, \quad \forall (\theta, a) \in \Theta \times A^+.$$

For a better understanding, $V^{\mathrm{FL}}$ reflects the maximum expected total rewards an agent can win when a static strategy is adopted and the resource constraints are only to be satisfied in expectation. Therefore, this optimization problem is a linear programming, in which the decision variable $\phi(\theta, a)$ represents the probability that the agent chooses action $a$ upon seeing request $\theta$. It is a well-known result that $V^{\mathrm{FL}}$ gives an upper bound on $V^{\mathrm{ON}}$.

**Proposition 2.1** ([Balseiro et al., 2021]). $V^{\mathrm{FL}} \ge V^{\mathrm{ON}}$.

Thus, we evaluate the performance of a non-anticipating strategy $\pi$ by comparing its expected accumulated reward $Rew^\pi$ with the fluid benchmark $V^{\mathrm{FL}}$. We call their difference the regret of $\pi$ for convenience. In this context, we prove that an $\Omega(\sqrt{T})$ regret lower bound is inevitable as long as $V^{\mathrm{FL}}$ is degenerate.

**Theorem 2.1** (worst-case location). *When $V^{\mathrm{FL}}$ has a unique and degenerate optimal solution,* $V^{\mathrm{FL}} - V^{\mathrm{ON}} = \Omega(\sqrt{T})$.

Despite the worst-case lower bound, we prove in this work that for any CBwK instance in which $V^{\mathrm{FL}}$ has a unique *non-degenerate* optimal solution (Assumption 3.1), we can obtain an $O(1)$ regret via the re-solving approach.

**Information feedback model.** In this work, we consider two types of information feedback models, with increasing levels of difficulty in obtaining a sample of the external factor $\gamma$.

- [Full information feedback.] The agent is able to observe $\gamma_t$ at the end of each round $t$.

- [Partial information feedback.] The agent can observe $\gamma_t$ at the end of round $t$ *only if $a_t \ne 0$*.

The above two information feedback models are illustrated in Figure 1. In general, with full information feedback, the agent can observe an i.i.d. sample from $\mathcal{V}$ each round, which is the optimal scenario for learning the distribution. Nevertheless, such an assumption may be overly strong since the reward and consumption vector are irrelevant to the external factor when the agent chooses the

**Algorithm 1:** Re-Solving with Empirical Estimation.

**Input:** $\rho$, $T$.

**Initialization:** $\mathcal{I}_1 \leftarrow \emptyset$, $\boldsymbol{B}_1 \leftarrow \boldsymbol{\rho} T$.

1 **for** $t \leftarrow 1$ **to** $T$ **do**

2  Observe $\theta_t$;

  /* Solve a linear programming with estimates. */

3  $\boldsymbol{\rho}_t \leftarrow \boldsymbol{B}_t/(T - t + 1)$;

4  $\widehat{\phi}_t^* \leftarrow$ the solution to $\widehat{J}(\boldsymbol{\rho}_t, \mathcal{H}_t)$;

5  Choose $a_t \in A$ randomly such that for $a \in A^+$, $\Pr[a_t = a] = \widehat{\phi}_t^*(\theta_t, a)$, and
  $\Pr[a_t = 0] = 1 - \sum_{a \in A^+} \widehat{\phi}_t^*(\theta_t, a)$;

  /* Observe the sample. */

6  **if** *(FULL-INFO)* $\lor$ *(PARTIAL-INFO* $\land a_t \neq 0$*)* **then**

7    Observe $\boldsymbol{\gamma}_t$;

8    $\mathcal{I}_{t+1} \leftarrow \mathcal{I}_t \cup \{t\}$;

9  **end**

10  **else**

11    $\mathcal{I}_{t+1} \leftarrow \mathcal{I}_t$;

12  **end**

  /* Update the remaining budget vector. */

13  $\boldsymbol{B}_{t+1} \leftarrow \boldsymbol{B}_t - \boldsymbol{c}_t$;

14  **if** $\boldsymbol{B}_{t+1}^i < 1$ *for some* $i \in [n]$ **then**

15    **break**;

16  **end**

17 **end**

---

204 null action $a = 0$. Thereby, a more realistic information model is partial feedback, where the external
205 factor is only accessible when $a \neq 0$. This limitation also increases the difficulty of learning the
206 distribution $\mathcal{V}$ since the agent observes fewer samples under this model than under full information
207 feedback. It is important to note that the partial information model represents a transition from full to
208 bandit information feedback, under which only the reward and consumption vector are accessible in
209 each round, rather than the external factor.

## 210 3 The Re-Solving Heuristic

211 In this work, we introduce the re-solving heuristic to the CBwK problem. The resulting algorithm is
212 presented in Algorithm 1.

213 To briefly describe the algorithm, we start by defining an optimization problem that captures the
214 optimal fluid control for each round, assuming complete knowledge of $\mathcal{U}$ and $\mathcal{V}$. For any $\boldsymbol{\kappa} \in [0, 1]^n$,
215 we define $J(\boldsymbol{\kappa})$ be the following optimization problem:

$$J(\boldsymbol{\kappa}) := \max_{\phi: \Theta \times A^+ \to \mathbb{R}} \mathbb{E}_{\theta \sim \mathcal{U}} \left[ \sum_{a \in A^+} R(\theta, a) \phi(\theta, a) \right],$$

$$\text{s.t.} \quad \mathbb{E}_{\theta \sim \mathcal{U}} \left[ \sum_{a \in A^+} \boldsymbol{C}(\theta, a) \phi(\theta, a) \right] \leq \boldsymbol{\kappa},$$

$$\sum_{a \in A^+} \phi(\theta, a) \leq 1, \quad \forall \theta \in \Theta,$$

$$\phi(\theta, a) \geq 0, \quad \forall (\theta, a) \in \Theta \times A^+.$$

216 Evidently, we have $V^{\mathrm{FL}} = T \cdot J(\boldsymbol{\rho}) = T \cdot J(\boldsymbol{\rho}_1)$ by definition. Intuitively, in each round $t$, the best
217 fluid choice of the agent is given by the optimal solution $\phi_t^*$ of LP $J(\boldsymbol{\rho}_t)$, where $\boldsymbol{\rho}_t$ is the average

budget of the remaining rounds, including round $t$. Nevertheless, since the agent lacks full knowledge of the exact distributions $\mathcal{U}$ and $\mathcal{V}$, she can only solve an estimated programming $\widehat{J}(\boldsymbol{\rho}_t, \mathcal{H}_t)$ as outlined in Algorithm 1, with the following realization:

$$\widehat{J}(\boldsymbol{\rho}_t, \mathcal{H}_t) := \max_{\phi:\Theta \times A^+ \to \mathbb{R}} \mathbb{E}_{\theta \sim \widehat{\mathcal{U}}_t} \left[ \sum_{a \in A^+} \mathbb{E}_{\gamma \sim \widehat{\mathcal{V}}_t} \left[ r(\theta, a, \gamma) \right] \phi(\theta, a) \right],$$

$$\text{s.t.} \quad \mathbb{E}_{\theta \sim \widehat{\mathcal{U}}_t} \left[ \sum_{a \in A^+} \mathbb{E}_{\gamma \sim \widehat{\mathcal{V}}_t} \left[ \boldsymbol{c}(\theta, a, \gamma) \right] \phi(\theta, a) \right] \leq \boldsymbol{\rho}_t,$$

$$\sum_{a \in A^+} \phi(\theta, a) \leq 1, \quad \forall \theta \in \Theta,$$

$$\phi(\theta, a) \geq 0, \quad \forall(\theta, a) \in \Theta \times A^+.$$

Here, $\widehat{\mathcal{U}}_t$ and $\widehat{\mathcal{V}}_t$ represent the empirical distribution of $\theta$ and $\gamma$, respectively, according to the sample history given by $\mathcal{H}_t$. Specifically, the mass functions of these two estimated distributions are standard as follows:

$$\widehat{u}_t(\theta) := \frac{\#[\theta \text{ appears in previous } t-1 \text{ rounds}]}{t-1};$$

$$\widehat{v}_t(\gamma) := \frac{\#[\gamma \text{ appears in } \mathcal{I}_t]}{|\mathcal{I}_t|}.$$

It is worth noting that the estimated distribution of $\theta$, $\widehat{\mathcal{U}}_t$, is always based $t-1$ samples since the agent received an independent sample from $\mathcal{U}$ at the beginning of each round. On the other hand, the empirical distribution of the external factor $\gamma$, $\widehat{\mathcal{V}}_t$, is estimated from $|\mathcal{I}_t|$ independent samples. With full information feedback, $|\mathcal{I}_t| = t-1$; whereas with partial information feedback, $|\mathcal{I}_t| \leq t-1$ equals the number of times the agent chooses an action $a \neq 0$ before round $t$. For brevity, for the estimated programming, we write $\widehat{\boldsymbol{C}}_t(\theta, a) := \mathbb{E}_{\gamma \sim \widehat{\mathcal{V}}_t}[\boldsymbol{c}(\theta, a, \gamma)]$ and $\widehat{R}_t(\theta, a) := \mathbb{E}_{\gamma \sim \widehat{\mathcal{V}}_t}[r(\theta, a, \gamma)]$.

As per Algorithm 1, the agent's decision mode in round $t$ is given by the optimal solution $\widehat{\phi}_t^*$ of programming $\widehat{J}(\boldsymbol{\rho}_t, \mathcal{H}_t)$. The algorithm stops when the resources are near depletion, that is, $\boldsymbol{B}^i \leq 1$ for some resource $i \in [n]$, and we use $T_0$ to denote the stopping time of Algorithm 1, i.e., $T_0 := \min\{T, \min\{t : \exists i \in [n], \boldsymbol{B}_{t+1}^i < 1\}\}$.

For an analysis beyond the worst-case scenario, a crucial assumption we will make is that the fluid problem possesses good regularity properties, i.e., it is an LP with a unique and non-degenerate solution.

**Assumption 3.1.** The optimal solution to $J(\boldsymbol{\rho}_1)$ is unique and non-degenerate.

The regularity assumption made Assumption 3.1 is commonplace in the linear programming literature [Chen et al., 2022, Li and Ye, 2021]. Further, any LP can easily avoid non-uniqueness or degeneracy through a slight perturbation [Megiddo and Chandrasekaran, 1989].

With the assumption, below we present the main result of this work, which is proved in Appendix C.1.

**Theorem 3.1.** *Under Assumption 3.1, with full information feedback, the expected accumulated reward $Rew$ brought by Algorithm 1 when $T \to \infty$ satisfies:*

$$V^{\mathrm{FL}} - Rew = O(1), \quad T \to \infty,$$

*which is independent of $T$.*

One of the key implications of Theorem 3.1 is that the re-solving heuristic's regret is independent of the number of rounds beyond the worst-case with full information. This result represents a significant improvement over previous state-of-the-art results under mild assumptions, surpassing the solutions proposed by Slivkins and Foster [2022], Han et al. [2022]. In particular, their solutions come from the bandits with knapsacks (BwK) literature and rely on dual update and upper confidence bound (UCB) heuristics, which only provide a worst-case regret of $O(\sqrt{T \log T})$ even with full information feedback. Furthermore, the reduction proposed by Sankararaman and Slivkins [2021] can only grant an $O(\log T)$ regret for linear CBwK problems, and relies on the strong assumption that there is a

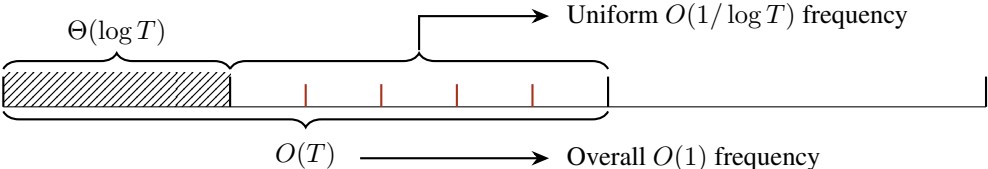

Figure 2: An illustration of Lemma 4.1.

universal best action and only $n = 2$ resources. In contrast, our assumption is more common and less restrictive.

Additionally, as pointed out in Theorem 2.1, an $\Omega(\sqrt{T})$ lower bound is established when the primal LP $J(\boldsymbol{\rho}_1)$ has a unique and degenerate optimal solution, while Theorem 3.1 provides an $O(1)$ upper bound on the optimal regret of CBwK with full information under the uniqueness and non-degeneracy condition. It is interesting to consider the regret bound in the remaining cases when $J(\boldsymbol{\rho}_1)$ has multiple optimal solutions.

It is worth noting the relationship between our theoretical regret and the number of resources $n$ and number of actions $m$. Generally, our analysis shows that the regret scales with (at most) the square of $n$. Further, a surprising result is that the regret is not explicitly related to $m$. This is superior to existing results, which report an $\widetilde{O}(\sqrt{m})$ reliance [Slivkins and Foster, 2022, Han et al., 2022, Badanidiyuru et al., 2014, Agrawal et al., 2016]. As an intuitive reason, the number of actions does not explicitly appear in the re-solving algorithm but only contributes to the dimension of the linear programming. However, this is not the case for other existing algorithms, which explicitly incorporate the number of actions $m$ into their algorithms, resulting in a correlation between the regret and $m$.

# 4 Partial Information Feedback

We now shift to consider the re-solving method's performance with partial information feedback, under which the agent only sees the external factor $\gamma_t$ when her choice is non-null in round $t$, i.e., $a_t \neq 0$. Apparently, with less information, the learning speed of the distribution $\mathcal{V}$ decreases, hindering the re-solving procedure's quick convergence to an optimal solution. Nevertheless, we demonstrate that the performance of the re-solving method only faces an $O(\log T)$ multiplicative degradation under partial information feedback. Our primary theorem in this section is as follows:

**Theorem 4.1.** *Under Assumption 3.1, with partial information feedback, the expected accumulated reward Rew brought by Algorithm 1 satisfies:*

$$V^{\mathrm{FL}} - Rew = O(\log T), \quad T \to \infty.$$

Before we come to the technical parts, we first place Theorem 4.1 within the literature. As previously mentioned, $\Omega(\sqrt{T})$ is a worst-case lower bound on the regret even with full information feedback, and thus also serves as a lower bound with partial information feedback. However, Theorem 4.1 steps beyond the worst-case by providing an $O(\log T)$ upper bound for most regular problem instances. This result outperforms the universal $O(\sqrt{T \log T})$ regret by Slivkins and Foster [2022], Han et al. [2022]. Although the result is asymptotically equivalent to that of Sankararaman and Slivkins [2021], it imposes fewer restrictions on the problem structure, as previously discussed. Moreover, the regret result's dependence on the number of resources $n$ and number of actions $m$ is inherited from Theorem 3.1.

We now provide an intuitive understanding of the proof of Theorem 4.1. The crux lies in analyzing the frequency that Algorithm 1 can access an independent sample of the external factor. To this end, we use $Y_t = |\mathcal{I}_t| \leq t - 1$ to denote the times of choosing action $a \neq 0$ before time $t$, or equivalently, the number of i.i.d. samples from $\mathcal{V}$ observed by the agent before time $t$, under partial information feedback. We have the following important lemma that presents a lower bound on $Y_t$.

**Lemma 4.1.** *There is a constant $0 < C_b < 1/2$, such that with probability $1 - O(1/T)$, the following hold for Algorithm 1:*

    *1. For any $\Theta(\log T) \leq t \leq C_b \cdot T$, $Y_t \geq C_f \cdot (t-1)/\log T$ for some constant $C_f$;*

2. *For any $t > C_b \cdot T$, $Y_t \geq C_r \cdot T$ for some constant $C_r$.*

The proof of Lemma 4.1 is deferred to Appendix D.2, and an illustration is displayed in Figure 2. In simple terms, during the first $\Theta(\log T)$ rounds (the shaded segment), the re-solving method cannot guarantee the accessing frequency since the learning of the request distribution $\mathcal{U}$ has not converged sufficiently. However, after $\Theta(\log T)$ rounds, Algorithm 1 ensures a constant probability of obtaining a new example in each round, provided that the remaining resources are sufficient. As a consequence, before $O(T)$ rounds, we can guarantee an $O(1/\log T)$ accessing frequency at any time step and an overall $O(1)$ frequency with high probability, by a concentration inequality. The remaining proof of Theorem 4.1 is provided in Appendix D.1.

## 5 Relaxing the Regularity Assumption – A Worst-Case Guarantee

In Sections 3 and 4, we have proved that can achieve an $\widetilde{O}(1)$ regret for CBwK problems under full or partial information feedbacks, assuming certain regular conditions (Assumption 3.1). Put differently, the re-solving heuristic nicely deals with regular scenarios. In this section, we complement this by showing that this method can also attain nearly optimal regret in the worst cases. Furthermore, we extend our analysis to cases where the context and external factor distributions can be continuous in Appendix A.

Our main results are given below, and their proofs are provided in Appendices E.1 and E.2, respectively.

**Theorem 5.1.** *With full information feedback, the expected accumulated reward $Rew$ brought by Algorithm 1 satisfies:*

$$V^{\text{FL}} - Rew = O(\sqrt{T \log T}), \quad T \to \infty.$$

**Theorem 5.2.** *With partial information feedback, the expected accumulated reward $Rew$ brought by Algorithm 1 satisfies:*

$$V^{\text{FL}} - Rew = O(\sqrt{T} \log T), \quad T \to \infty.$$

As given by Theorem 2.1, the worst-case regret of any online CBwK algorithm is $\Omega(\sqrt{T})$, while Theorems 5.1 and 5.2 indicate that the re-solving heuristic reaches near-optimality in such cases. Further, state-of-the-art algorithms [Han et al., 2022, Slivkins and Foster, 2022]) can at most obtain an $\widetilde{O}(\sqrt{T})$ regret with full/partial information feedback. Our algorithm also achieves this regret bound in worst cases.

## 6 Concluding Remarks

This work establishes the effectiveness of the re-solving heuristic in the contextual bandits with knapsacks problem. We first prove that any online algorithm incurs a regret of $\Omega(\sqrt{T})$ when the fluid LP has a unique and degenerate optimal solution. Building on this, we demonstrate that the re-solving method reaches an $O(1)$ regret with full information and an $O(\log T)$ regret with partial information when the fluid LP has a unique and non-degenerate optimal solution. Considering the sufficient condition for the $\Omega(\sqrt{T})$ lower bound, our non-degeneracy assumption is mild, especially when combined with the two-resource and best-arm-optimality condition required in Sankararaman and Slivkins [2021].

Further, we show that even in the worst-case, the re-solving method achieves an $O(\sqrt{T \log T})$ regret with full information feedback and an $O(\sqrt{T} \log T)$ regret with partial information feedback. These results are comparable to start-of-the-art results [Slivkins and Foster, 2022, Han et al., 2022]. We also extend our analysis to the continuous randomness case for completeness.

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
