# Appendix of "Contextual Bandits with Knapsacks beyond Worst Cases via Re-Solving"

## A From Discrete Randomness to Continuous Randomness

In the main body of this work, we explicitly assume that both the context set and the external factor set are discrete. Such an assumption can suitably capture most real-life situations. For example, in an agent's online bidding problem with budget constraints, if we presume that the context is the agent's actual value and the external factor is the highest competing bid, it is natural to suppose that all these three values are discrete. Nevertheless, for theoretical completeness, we expand our results in this section to circumstances where these two sets are infinite, i.e., the two underlying randomnesses is continuous. It is imperative to note that the scenario where one randomness is discrete and the other is continuous would be analogous in analysis by incorporating the techniques presented in Section 5.

Conceptually, the re-solving heuristic still works: we solve the optimization problem in each round concerning the remaining resources based on previous estimates. However, technically, since the distributions of context and external factors are continuous, we should further elaborate on the setting. In this section, we suppose that the context set $\Theta = [0,1]^{d_u}$ and the external factor set $\Gamma = [0,1]^{d_v}$. We denote $u(\theta)$ and $v(\gamma)$ as the density function of $\mathcal{U}$ and $\mathcal{V}$, respectively. We assume that $p \in \{u,v\}$ belongs to the $\beta_p$-order $L_p$-Hölder smooth class $\Sigma(\beta_p, L_p)$. Here, for the foundation, given a vector $s = (s_1, ..., s_d)$, define

$$|s| = s_1 + \cdots + s_d, \quad D^s = \frac{\partial^{s_1 + \cdots + s_d}}{\partial x_1^{s_1} \cdots \partial x_d^{s_d}}.$$

Subsequently, for a positive integer $\beta$, the $\beta$-order $L$-Hölder smooth class is defined as

$$\Sigma(\beta, L) := \{g : |D^s g(x) - D^s g(y)| \le L\|x - y\|_2, \text{ for all } s \text{ such that } |s| = \beta - 1, \text{ and all } x, y\}.$$

Now, suppose $X_1, \cdots, X_k$ are $k$ i.i.d. samples from a distribution with density function $p \in \Sigma(\beta, L)$. According to Wasserman [2019], we have the following result, which implies that we can calculate an estimator from these samples that converges to the density function.

**Proposition A.1** ([Wasserman, 2019]). *Suppose $X_1, \cdots, X_k$ are drawn i.i.d. from a $d$-dimension distribution $\mathcal{P}$, with density $p \in \Sigma(\beta, L)$ for some $L > 0$, and $k$ is sufficiently large. Then there exists an estimator $\widehat{p}_k$ such that for any $\epsilon > 0$,*

$$\Pr\left[\sup_x |p(x) - \widehat{p}_k(x)| > \frac{C\sqrt{\log(k/\epsilon)}}{k^{\beta/(2\beta+d)}}\right] \le \epsilon,$$

*with $C$ a constant.*

The details of constructing such a density estimator are postponed to Appendix F.1. We now return to the re-solving heuristic and Algorithm 1. In the algorithm, with continuous randomness, the constrained optimization problem to be solved in each round $\widehat{J}(\boldsymbol{\rho}_t, \mathcal{H}_t)$ for $t = 1, 2, \cdots$ becomes:

$$\widehat{J}(\boldsymbol{\rho}_t, \mathcal{H}_t) := \max_{\phi:\Theta \times A^+ \to \mathbb{R}} \int_\theta \sum_{a \in A^+} \phi(\theta, a) \int_\gamma r(\theta, a, \gamma) \widehat{v}_t(\gamma) \widehat{u}_t(\theta) \, d\gamma \, d\theta,$$

$$\text{s.t.} \quad \int_\theta \sum_{a \in A^+} \phi(\theta, a) \int_\gamma \boldsymbol{c}(\theta, a, \gamma) \widehat{v}_t(\gamma) \widehat{u}_t(\theta) \, d\gamma \, d\theta \le \boldsymbol{\rho}_t,$$

$$\sum_{a \in A^+} \phi(\theta, a) \le 1, \quad \forall \theta \in \Theta,$$

$$\phi(\theta, a) \ge 0, \quad \forall (\theta, a) \in \Theta \times A^+.$$

Correspondingly, the reference optimization problem $J(\boldsymbol{\rho}_t)$ is given below:

$$J(\boldsymbol{\rho}_t) := \max_{\phi:\Theta \times A^+ \to \mathbb{R}} \int_\theta \sum_{a \in A^+} \phi(\theta, a) \int_\gamma r(\theta, a, \gamma) v(\gamma) u(\theta) \, \mathrm{d}\gamma \, \mathrm{d}\theta,$$

$$\text{s.t.} \quad \int_\theta \sum_{a \in A^+} \phi(\theta, a) \int_\gamma \boldsymbol{c}(\theta, a, \gamma) v(\gamma) u(\theta) \, \mathrm{d}\gamma \, \mathrm{d}\theta \le \boldsymbol{\rho}_t,$$

$$\sum_{a \in A^+} \phi(\theta, a) \le 1, \quad \forall \theta \in \Theta,$$

$$\phi(\theta, a) \ge 0, \quad \forall(\theta, a) \in \Theta \times A^+.$$

At this point, it is worth mentioning that solving $\widehat{J}(\boldsymbol{\rho}_t, \mathcal{H}_t)$ in each round could be hard as it could be a continuous yet non-convex constrained optimization problem. Nevertheless, we assume the existence of an oracle that aids us in solving this optimization, and we focus on the regret of the re-solving method. Let $\alpha_u := (\beta_u + d_u)/(2\beta_u + d_u)$ and $\alpha_v := (\beta_v + d_v)/(2\beta_v + d_v)$, and we have the following two results, respectively, under full and partial information feedback.

**Theorem A.1.** *Under continuous randomness, with full information feedback, the expected accumulated reward $Rew$ brought by Algorithm 1 satisfies:*

$$V^{\mathrm{FL}} - Rew = O((T^{\alpha_u} + T^{\alpha_v} + T^{1/2})\sqrt{\log T}), \quad T \to \infty.$$

**Theorem A.2.** *Under continuous randomness, with partial information feedback, the expected accumulated reward $Rew$ brought by Algorithm 1 satisfies:*

$$V^{\mathrm{FL}} - Rew = O((T^{\alpha_u} + T^{1/2})\sqrt{\log T} + T^{\alpha_v} \log^{3/2 - \alpha_v} T), \quad T \to \infty.$$

The proofs of the above theorems are presented in Appendices F.2 and F.3, respectively, which almost follow the threads of Theorems 5.1 and 5.2.

# B  Specifying the Worst-Case Location – Proof of Theorem 2.1

To prove the lemma, we first introduce an intermediate value, which we denote as $V^{\mathrm{Hyb}}$, to upper bound $V^{\mathrm{ON}}$, and show that the gap between $V^{\mathrm{Hyb}}$ and $V^{\mathrm{FL}}$ is $O(\sqrt{T})$ under the given condition. Specifically, we have the following definition:

$$V^{\mathrm{Hyb}} := \mathbb{E}_{\theta_1, \cdots, \theta_T} \left[ \max_{\phi_1, \cdots, \phi_T:A^+ \to \mathbb{R}} \sum_{t=1}^T \sum_{a \in A^+} R(\theta_t, a)\phi_t(a) \right],$$

$$\text{s.t.} \quad \sum_{t=1}^T \sum_{a \in A^+} \boldsymbol{C}(\theta_t, a)\phi_t(a) \le \boldsymbol{\rho} T, \tag{1}$$

$$\sum_{a \in A^+} \phi_t(a) \le 1, \quad \forall t \in [T],$$

$$\phi_t(a) \ge 0, \quad \forall(t, a) \in [T] \times A^+.$$

To see that $V^{\mathrm{Hyb}}$ gives an upper bound on $V^{\mathrm{ON}}$, we fix a request trajectory $\theta_1, \cdots, \theta_T$. Now, for any non-anticipating strategy $\pi$, we let

$$p_t^\pi(a) = \Pr[a_t^\pi = a \mid \theta_1, \cdots, \theta_t]$$

be the total probability that $a_t^\pi = a$ conditioning on the pre-determined request sequence, with respect to $\gamma_1, \cdots, \gamma_{t-1}$ and the randomness of strategy $\pi$. We show that $\{p_t^\pi\}_{t=1,\cdots,T}$ is a feasible solution to $V^{\mathrm{Hyb}}$ under $\theta_1, \cdots, \theta_T$. Here, a key observation is that for any $t \in [T]$:

$$\mathbb{E}\left[\boldsymbol{c}(\theta_t, a_t^\pi, \gamma_t) \mid \theta_1, \cdots, \theta_t\right] = \mathbb{E}_{\gamma_t}\left[\sum_{a \in A^+} \boldsymbol{c}(\theta_t, a_t^\pi, \gamma_t) \cdot \Pr[a_t^\pi = a \mid \theta_1, \cdots, \theta_t]\right]$$

$$= \sum_{a \in A^+} \boldsymbol{C}(\theta_t, a)p_t^\pi(a).$$

In the above, the first expectation is taken on $\gamma_1, \cdots, \gamma_t$ and the random choice of strategy $\pi$. Since $\sum_{t=1}^{T} \boldsymbol{c}(\theta_t, a_t^\pi, \gamma_t) \leq \boldsymbol{\rho}T$ always holds, we derive that

$$\sum_{t=1}^{T} \sum_{a \in A^+} \boldsymbol{C}(\theta_t, a) p_t^\pi(a) = \mathbb{E}\left[\sum_{t=1}^{T} \boldsymbol{c}(\theta_t, a_t^\pi, \gamma_t) \mid \theta_1, \cdots, \theta_T\right] \leq \boldsymbol{\rho}T,$$

which indicates that $\{p_t^\pi\}_{t=1,\cdots,T}$ is feasible to $V^{\text{Hyb}}$ under $\theta_1, \cdots, \theta_T$. To the same reason, we also have

$$\sum_{t=1}^{T} \sum_{a \in A^+} R(\theta_t, a) p_t^\pi(a) = \mathbb{E}\left[\sum_{t=1}^{T} \boldsymbol{r}(\theta_t, a_t^\pi, \gamma_t) \mid \theta_1, \cdots, \theta_T\right]$$

equals the conditional expected reward of strategy $\pi$. Thus, since $V^{\text{Hyb}}$ is a maximization problem for any request trajectory, we conclude that $V^{\text{Hyb}} \geq V^{\text{ON}}$.

It remains to show that when $V^{\text{FL}}$, or $J(\boldsymbol{\rho})$ has a unique and degenerate solution, $V^{\text{FL}} - V^{\text{Hyb}} = \Omega(\sqrt{T})$. We first present a transformation of $V^{\text{Hyb}}$. We let

$$x(\theta) := \frac{\#[\text{appearance of } \theta]}{T}$$

be the random variable indicating the frequency of $\theta$ when $\theta$ is drawn $T$ times i.i.d. from $\mathcal{U}$. Obviously, the mean of $\boldsymbol{x}$ is $\boldsymbol{u}$. We now demonstrate that

$$
\begin{aligned}
V^{\text{Hyb}} = T \cdot \mathbb{E}_{\boldsymbol{x}} &\left[\max_{\phi:\Theta \times A^+ \to \mathbb{R}} \sum_{\theta \in \Theta} x(\theta) \sum_{a \in A^+} R(\theta, a)\phi(\theta, a)\right], \\
\text{s.t.} \quad &\sum_{\theta \in \Theta} x(\theta) \sum_{a \in A^+} \boldsymbol{C}(\theta, a)\phi(\theta, a) \leq \boldsymbol{\rho}, \\
&\sum_{a \in A^+} \phi(\theta, a) \leq 1, \quad \forall \theta \in \Theta, \\
&\phi(\theta, a) \geq 0, \quad \forall(\theta, a) \in \Theta \times A^+.
\end{aligned}
\tag{2}
$$

To see this, in form (1), it is not hard to see that conditioning on $\theta_1, \cdots, \theta_T$, the value of the optimization is only related to the number of times that any $\theta \in \Theta$ appears in the sequence, and irrelevant with their arriving order. Therefore, by taking an average, it is without loss of generality to suppose that $\phi_{t_1}^* = \phi_{t_2}^*$ as long as $\theta_{t_1} = \theta_{t_2}$. Under such an observation, it is natural that (1) is equivalent to (2).

For convenience, we now recall the definition of $V^{\text{FL}}$:

$$
\begin{aligned}
V^{\text{FL}} = T \cdot &\max_{\phi:\Theta \times A^+ \to \mathbb{R}} \sum_{\theta \in \Theta} u(\theta) \sum_{a \in A^+} R(\theta, a)\phi(\theta, a), \\
\text{s.t.} \quad &\sum_{\theta \in \Theta} u(\theta) \sum_{a \in A^+} \boldsymbol{C}(\theta, a)\phi(\theta, a) \leq \boldsymbol{\rho}, \\
&\sum_{a \in A^+} \phi(\theta, a) \leq 1, \quad \forall \theta \in \Theta, \\
&\phi(\theta, a) \geq 0, \quad \forall(\theta, a) \in \Theta \times A^+.
\end{aligned}
$$

By Sierksma [2001], we know that when $J(\boldsymbol{\rho})$ has a unique and degenerate solution, then its dual form has multiple solutions. We then adopt the framework of Vera and Banerjee [2021]. In particular, we let $\boldsymbol{\lambda} \geq \boldsymbol{0}$ be the dual variable vector for the resource constraints, and $\boldsymbol{\mu} \geq \boldsymbol{0}$ be the dual variable vector for the probability feasibility constraints. If we take $\omega(\theta) = \mu(\theta)/u(\theta)$, then the dual programming of $V^{\text{FL}}/T$ is the following as a function of $\boldsymbol{u}$:

$$
\begin{aligned}
\mathcal{D}[Z(\boldsymbol{u})] = \min_{\boldsymbol{\lambda}, \boldsymbol{\omega}} &\boldsymbol{\rho}^\top \boldsymbol{\lambda} + \boldsymbol{u}^\top \boldsymbol{\omega}, \\
\text{s.t.} \quad &\boldsymbol{\lambda}^\top \boldsymbol{C}(\theta, a) + \omega(\theta) \geq R(\theta, a), \quad \forall(\theta, a) \in \Theta \times A^+, \\
&\boldsymbol{\lambda} \geq \boldsymbol{0}, \quad \boldsymbol{\omega} \geq \boldsymbol{0}.
\end{aligned}
$$

481  Now, suppose $(\boldsymbol{\lambda}^1, \boldsymbol{\omega}^1)$ and $(\boldsymbol{\lambda}^2, \boldsymbol{\omega}^2)$ are two different optimal solutions to $\mathcal{D}[Z(\boldsymbol{u})]$, which directly
482  leads to $\boldsymbol{\lambda}^1 \neq \boldsymbol{\lambda}^2$ by the programming formation. We let $\boldsymbol{\lambda}' = \boldsymbol{\lambda}^1 - \boldsymbol{\lambda}^2$ and $\boldsymbol{\omega}' = \boldsymbol{\omega}^1 - \boldsymbol{\omega}^2$. Then,

$$\boldsymbol{\rho}^\top \boldsymbol{\lambda}^1 + \boldsymbol{u}^\top \boldsymbol{\omega}^1 = \boldsymbol{\rho}^\top \boldsymbol{\lambda}^2 + \boldsymbol{u}^\top \boldsymbol{\omega}^2 \implies \boldsymbol{\rho}^\top \boldsymbol{\lambda}' + \boldsymbol{u}^\top \boldsymbol{\omega}' = 0. \tag{3}$$

483  Further, notice that $(\boldsymbol{\lambda}^1, \boldsymbol{\omega}^1)$ and $(\boldsymbol{\lambda}^2, \boldsymbol{\omega}^2)$ are both feasible for $\mathcal{D}[Z(\boldsymbol{x})]$ for any $\boldsymbol{x}$. Since $\mathcal{D}[Z(\boldsymbol{x})]$
484  is a minimization problem, by a convex combination, we have

$$\mathcal{D}[Z(\boldsymbol{x})] \leq (\boldsymbol{\rho}^\top \boldsymbol{\lambda}^1 + \boldsymbol{x}^\top \boldsymbol{\omega}^1)\mathbf{1}[\boldsymbol{\rho}^\top \boldsymbol{\lambda}' + \boldsymbol{x}^\top \boldsymbol{\omega}' \leq 0] + (\boldsymbol{\rho}^\top \boldsymbol{\lambda}^2 + \boldsymbol{x}^\top \boldsymbol{\omega}^2)\mathbf{1}[\boldsymbol{\rho}^\top \boldsymbol{\lambda}' + \boldsymbol{x}^\top \boldsymbol{\omega}' > 0].$$

485  Further, by optimality, we know that for any $\boldsymbol{x}$,

$$\mathcal{D}[Z(\boldsymbol{u})] = (\boldsymbol{\rho}^\top \boldsymbol{\lambda}^1 + \boldsymbol{u}^\top \boldsymbol{\omega}^1)\mathbf{1}[\boldsymbol{\rho}^\top \boldsymbol{\lambda}' + \boldsymbol{x}^\top \boldsymbol{\omega}' \leq 0] + (\boldsymbol{\rho}^\top \boldsymbol{\lambda}^2 + \boldsymbol{u}^\top \boldsymbol{\omega}^2)\mathbf{1}[\boldsymbol{\rho}^\top \boldsymbol{\lambda}' + \boldsymbol{x}^\top \boldsymbol{\omega}' > 0].$$

486  Now, by weak duality, since $V^{\mathrm{Hyb}}/T$ for any given $\boldsymbol{x}$ is a maximization problem, we know from the
487  above two equations that

$$\begin{aligned}
&(V^{\mathrm{FL}} - V^{\mathrm{Hyb}})/T \\
&\geq \mathcal{D}[Z(\boldsymbol{u})] - \mathbb{E}_{\boldsymbol{x}}\left[\mathcal{D}[Z(\boldsymbol{x})]\right] \\
&\geq \mathbb{E}_{\boldsymbol{x}}\left[((\boldsymbol{u}-\boldsymbol{x})^\top \boldsymbol{\omega}^1)\mathbf{1}[\boldsymbol{\rho}^\top \boldsymbol{\lambda}' + \boldsymbol{x}^\top \boldsymbol{\omega}' \leq 0] + ((\boldsymbol{u}-\boldsymbol{x})^\top \boldsymbol{\omega}^2)\mathbf{1}[\boldsymbol{\rho}^\top \boldsymbol{\lambda}' + \boldsymbol{x}^\top \boldsymbol{\omega}' > 0]\right] \\
&\stackrel{(a)}{=} \mathbb{E}_{\boldsymbol{x}}\left[((\boldsymbol{u}-\boldsymbol{x})^\top \boldsymbol{\omega}^1)\mathbf{1}[(\boldsymbol{u}-\boldsymbol{x})^\top \boldsymbol{\omega}' \geq 0] + ((\boldsymbol{u}-\boldsymbol{x})^\top \boldsymbol{\omega}^2)(1 - \mathbf{1}[(\boldsymbol{u}-\boldsymbol{x})^\top \boldsymbol{\omega}' \geq 0])\right] \\
&\stackrel{(b)}{=} \mathbb{E}_{\boldsymbol{x}}\left[((\boldsymbol{u}-\boldsymbol{x})^\top \boldsymbol{\omega}')\mathbf{1}[(\boldsymbol{u}-\boldsymbol{x})^\top \boldsymbol{\omega}' \geq 0]\right]
\end{aligned}$$

488  Here, (a) is due to (3), and (b) is since the mean of $\boldsymbol{x}$ is $\boldsymbol{u}$. Now, we let $\xi = \sqrt{T}(\boldsymbol{u} - \boldsymbol{x})^\top \boldsymbol{\omega}'$ be
489  the normalized scaled variable. By Central Limit Theorem, $\xi\mathbf{1}[\xi \geq 0]$ converges to a half-normal
490  distribution, which has constant expectation. Thus, we arrive at $V^{\mathrm{FL}} - V^{\mathrm{Hyb}} = \Omega(\sqrt{T})$, which
491  finish the proof.

## C  Missing Proofs in Section 3

### C.1  Proof of Theorem 3.1

494  We now give a proof of Theorem 3.1. The proof draws inspiration from that of Chen et al. [2022], but
495  significantly diverges in terms of the problem setting.

#### C.1.1  Regret Decomposition

497  We start by presenting a regret decomposition approach, which stands on the dual viewpoint. We first
498  recall the optimization problem $V^{\mathrm{FL}} = T \cdot J(\boldsymbol{\rho}_1)$:

$$J(\boldsymbol{\rho}_1) \coloneqq \max_{\phi: \Theta \times A^+ \to \mathbb{R}} \mathbb{E}_{\theta \sim \mathcal{U}}\left[\sum_{a \in A^+} R(\theta, a)\phi(\theta, a)\right],$$

$$\text{s.t.} \quad \mathbb{E}_{\theta \sim \mathcal{U}}\left[\sum_{a \in A^+} \boldsymbol{C}(\theta, a)\phi(\theta, a)\right] \leq \boldsymbol{\rho}_1,$$

$$\sum_{a \in A^+} \phi(\theta, a) \leq 1, \quad \forall \theta \in \Theta,$$

$$\phi(\theta, a) \geq 0, \quad \forall (\theta, a) \in \Theta \times A^+.$$

499  Recall that $u(\theta)$ denotes the mass function of $\mathcal{U}$, then the above linear programming can be expanded
500  as

$$J(\boldsymbol{\rho}_1) \coloneqq \max_{\phi: \Theta \times A^+ \to \mathbb{R}} \sum_{\theta \in \Theta, a \in A^+} u(\theta)R(\theta, a)\phi(\theta, a),$$

$$\text{s.t.} \quad \sum_{\theta \in \Theta, a \in A^+} u(\theta)\boldsymbol{C}(\theta, a)\phi(\theta, a) \leq \boldsymbol{\rho}_1,$$

$$\sum_{a \in A^+} \phi(\theta, a) \leq 1, \quad \forall \theta \in \Theta,$$

$$\phi(\theta, a) \geq 0, \quad \forall (\theta, a) \in \Theta \times A^+.$$

Now let $\boldsymbol{\lambda} \geq \boldsymbol{0}$ be the dual vector for the consumption constraint and $\{\mu^*(\theta)\}_{\theta \in \Theta} \geq \boldsymbol{0}$ be the dual variables for the action distribution constraint. By the strong duality of linear programming, there is an optimal dual variable tuple $(\boldsymbol{\lambda}^*, \{\mu^*(\theta)\}_{\theta \in \Theta}) \geq \boldsymbol{0}$ such that:

$$
\begin{aligned}
J(\boldsymbol{\rho}_1) &= \sum_{\theta \in \Theta, a \in A^+} \left( u(\theta) \left( R(\theta, a) - (\boldsymbol{\lambda}^*)^\top \boldsymbol{C}(\theta, a) \right) - \mu^*(\theta) \right) \phi_1^*(\theta, a) + (\boldsymbol{\lambda}^*)^\top \boldsymbol{\rho}_1 + \sum_{\theta \in \Theta} \mu^*(\theta) \\
&= \sum_{\theta \in \Theta, a \in A^+} u(\theta) \left( R(\theta, a) - (\boldsymbol{\lambda}^*)^\top \boldsymbol{C}(\theta, a) \right) \phi_1^*(\theta, a) + (\boldsymbol{\lambda}^*)^\top \boldsymbol{\rho}_1.
\end{aligned}
\tag{4}
$$

Here $\phi_1^*$ is the optimal solution to $J(\boldsymbol{\rho}_1)$. With (4), we have the following lemma for regret decomposition.

**Lemma C.1.** *For any stopping time $T_e \leq T_0$ adapted to the process $\{\boldsymbol{B}_t\}$'s, we have*

$$
\begin{aligned}
& V^{\mathrm{FL}} - Rew \\
& \leq \mathbb{E}\left[ \sum_{t=1}^{T_e} \sum_{\theta \in \Theta, a \in A^+} \left( u(\theta) \left( R(\theta, a) - (\boldsymbol{\lambda}^*)^\top \boldsymbol{C}(\theta, a) \right) - \mu^*(\theta) \right) \left( \phi_1^*(\theta, a) - \widehat{\phi}_t^*(\theta, a) \right) \right] \\
& \quad + \mathbb{E}\left[ \sum_{t=1}^{T_e} \sum_{\theta \in \Theta} \mu^*(\theta) \left( 1 - \sum_{a \in A^+} \widehat{\phi}_t^*(\theta, a) \right) \right] \\
& \quad + (\boldsymbol{\lambda}^*)^\top \mathbb{E}\left[ \boldsymbol{B}_{T_e+1} \right] + \max_{\theta \in \Theta, a \in A^+} \left( R(\theta, a) - (\boldsymbol{\lambda}^*)^\top \boldsymbol{C}(\theta, a) \right) \cdot \mathbb{E}\left[ T - T_e \right].
\end{aligned}
\tag{5}
$$

The proof of Lemma C.1 is deferred to Appendix C.2. We now give a brief explanation on this result. The first two terms in (5) depicts the gap between the choice of Algorithm 1 and the optimal decision. This is apparent for the first term. For the second term, we should notice that by complementary slackness, for each $\theta \in \Theta$,

$$
\mu^*(\theta) \cdot \left( 1 - \sum_{a \in A^+} \phi_1^*(\theta, a) \right) = 0.
$$

Therefore, the second term in (5) is bounded if $\widehat{\phi}_t^*$ is close to $\phi_1^*$.

On the other hand, the last two terms are closely related to the choice of stopping time $T_e$ and the consumption behavior of Algorithm 1. Intuitively, if $T_e$ is sufficiently close to $T$, then $\mathbb{E}[T - T_e]$ should be appropriately bounded. Nevertheless, if the algorithm spends the resources too fast, then such a sufficiently large $T_e$ would be impossible. Conversely, if the resources are consumed substantially slower than the optimal, then the term $\mathbb{E}[\boldsymbol{B}_{T_e+1}]$, the remaining resources at the stopping time, would be unbounded.

In the following, we will deal with these two parts correspondingly. A crux to the analysis is to pick a satisfying stopping time $T_e$, which we will first cover.

### C.1.2 The Gap to Optimal Decision

We first give a realization of the stopping time $T_e$, which relies on Assumption 3.1. As is shown by Mangasarian and Shiau [1987], Chen et al. [2022], local stability holds for an LP with unique and non-degenerate optimal solution, that is, the basic variables and binding constraints are kept within a minor purturbation on the coefficients. To this end, we first explicitly define the relevant concepts.

**Definition C.1.** A context-action pair $(\theta, a)$ is a basic variable for $J(\boldsymbol{\rho}_1)$ if $\phi_1^*(\theta, a) > 0$, or else, it is a non-basic variable. Similarly define basic/non-basic variables for $\widehat{J}(\boldsymbol{\rho}_t, \mathcal{H}_t)$.

**Definition C.2.** $i \in [n]$ is a binding constraint for $J(\boldsymbol{\rho}_1)$ if

$$
\sum_{\theta \in \Theta, a \in A^+} u(\theta) \boldsymbol{C}^i(\theta, a) \phi_1^*(\theta, a) = \boldsymbol{\rho}_1^i,
$$

or else it is a non-binding constraint. We let

$$
\begin{aligned}
\mathcal{S} &:= \{ i \in [n] : i \text{ is a binding constraint for } J(\boldsymbol{\rho}_1) \}, \\
\mathcal{T} &:= \{ i \in [n] : i \text{ is a non-binding constraint for } J(\boldsymbol{\rho}_1) \},
\end{aligned}
$$

and use $\boldsymbol{\kappa}|_{\mathcal{S}}$ to define the sub-vector of $\boldsymbol{\kappa}$ confined on $\mathcal{S}$, similar for $\boldsymbol{\kappa}|_{\mathcal{T}}$. Further, $\theta \in \Theta$ is a binding constraint for $J(\boldsymbol{\rho}_1)$ if

$$\sum_{a \in A^+} \phi_1^*(\theta, a) = 1,$$

or else it is a non-binding constraint. Similarly define binding/non-binding constraints for $\widehat{J}(\boldsymbol{\rho}_t, \mathcal{H}_t)$.

Under the above definitions, we have the following lemma, which is a derivation of the result in Chen et al. [2022]. We will provide the proof in Appendix C.3:

**Lemma C.2** (Stability). *Under Assumption 3.1, there is a $D > 0$, such that when the following holds:*

$$\begin{aligned} \max \left\{ \|(u(\theta) - \widehat{u}_t(\theta))_{\theta \in \Theta}\|_\infty, \|(v(\gamma) - \widehat{v}_t(\gamma))_{\gamma \in \Gamma}\|_1 \right\} &\leq D, \\ \max \left\{ \|\boldsymbol{\rho}_1|_{\mathcal{S}} - \boldsymbol{\rho}_t|_{\mathcal{S}}\|_\infty, \max \{\boldsymbol{\rho}_1|_{\mathcal{T}} - \boldsymbol{\rho}_t|_{\mathcal{T}}\} \right\} &\leq D, \end{aligned} \tag{6}$$

*$J(\boldsymbol{\rho}_1)$ and $\widehat{J}(\boldsymbol{\rho}_t, \mathcal{H}_t)$ share the same sets of basic/non-basic variables and binding/non-binding constraints.*

With Lemma C.2 in hand, we can derive that when condition (6) is met, it holds that

$$\left( u(\theta) \left( R(\theta, a) - (\boldsymbol{\lambda}^*)^\top \boldsymbol{C}(\theta, a) \right) - \mu^*(\theta) \right) \left( \phi_1^*(\theta, a) - \widehat{\phi}_t^*(\theta, a) \right) = 0, \tag{7}$$

$$\sum_{\theta \in \Theta} \mu^*(\theta) \left( 1 - \sum_{a \in A^+} \widehat{\phi}_t^*(\theta, a) \right) = 0. \tag{8}$$

To see these, notice that by the dual feasibility of $J(\boldsymbol{\rho}_1)$, we have $u(\theta) \left( R(\theta, a) - (\boldsymbol{\lambda}^*)^\top \boldsymbol{C}(\theta, a) \right) - \mu^*(\theta) \leq 0$. When $u(\theta) \left( R(\theta, a) - (\boldsymbol{\lambda}^*)^\top \boldsymbol{C}(\theta, a) \right) - \mu^*(\theta) < 0$, by primal optimality, $\phi_1^*(\theta, a) = 0$ and thus $(\theta, a)$ is non-basic for $J(\boldsymbol{\rho}_1)$. By Lemma C.2, $(\theta, a)$ is also non-basic for $\widehat{J}(\boldsymbol{\rho}_t, \mathcal{H}_t)$ and $\widehat{\phi}_t^*(\theta, a) = 0$ holds as well. This finishes the deduction of (7). A similar reasoning on binding constraints would help us achieve (8), which we omit here.

As the above goes, it is then natural for us to define $T_e$ the stopping time in our analysis as follows:

$$T_e := \min\{T_0, \min\{t : \max\{\|\boldsymbol{\rho}_1|_{\mathcal{S}} - \boldsymbol{\rho}_t|_{\mathcal{S}}\|_\infty, \max\{\boldsymbol{\rho}_1|_{\mathcal{T}} - \boldsymbol{\rho}_t|_{\mathcal{T}}\}\} > D\} - 1\}, \tag{9}$$

where $T_0$ is the stopping time of Algorithm 1. That is to say, we always have $\max\{\|\boldsymbol{\rho}_1|_{\mathcal{S}} - \boldsymbol{\rho}_t|_{\mathcal{S}}\|_\infty, \max\{\boldsymbol{\rho}_1|_{\mathcal{T}} - \boldsymbol{\rho}_t|_{\mathcal{T}}\}\} \leq D$ when $t \leq T_e$. What we are left is to bound the situation when $\max\{\|(u(\theta) - \widehat{u}_t(\theta))_{\theta \in \Theta}\|_\infty, \|(v(\gamma) - \widehat{v}_t(\gamma))_{\gamma \in \Gamma}\|_1\} > D$ for $1 \leq t \leq T_e$. In total, we arrive at the following result for this part, with the proof given in Appendix C.4:

**Lemma C.3.** *Under Assumption 3.1, with full information feedback, we have when $T \to \infty$:*

$$\mathbb{E}\left[ \sum_{t=1}^{T_e} \sum_{\theta \in \Theta, a \in A^+} \left( u(\theta) \left( R(\theta, a) - (\boldsymbol{\lambda}^*)^\top \boldsymbol{C}(\theta, a) \right) - \mu^*(\theta) \right) \left( \phi_1^*(\theta, a) - \widehat{\phi}_t^*(\theta, a) \right) \right] = O(1),$$

$$\mathbb{E}\left[ \sum_{t=1}^{T_e} \sum_{\theta \in \Theta} \mu^*(\theta) \left( 1 - \sum_{a \in A^+} \widehat{\phi}_t^*(\theta, a) \right) \right] = O(1).$$

We are now only left to bound the last two terms in (5).

### C.1.3 The Gap to Optimal Consumption

As presented in (5), we now bound the remaining two terms, respectively $\mathbb{E}[\boldsymbol{B}_{T_e+1}]$ and $\mathbb{E}[T - T_e]$ for $T_e$ defined in (9). It turns out that these two terms are closely related. Due to this observation, we would first bound $(\boldsymbol{\lambda}^*)^\top \cdot \mathbb{E}[\boldsymbol{B}_{T_e+1}]$ by $\mathbb{E}[T - T_e]$, and then bound $\mathbb{E}[T - T_e]$.

Now by the strong duality of $J(\boldsymbol{\rho}_1)$, we know that complementary slackness holds, that is $\boldsymbol{\lambda}^*|_{\mathcal{T}} = \boldsymbol{0}$. We therefore have

$$\begin{aligned} (\boldsymbol{\lambda}^*)^\top \mathbb{E}[\boldsymbol{B}_{T_e+1}] &\leq (\boldsymbol{\lambda}^*)^\top \mathbb{E}[\boldsymbol{B}_{T_e}] = (\boldsymbol{\lambda}^*|_{\mathcal{S}})^\top \mathbb{E}[\boldsymbol{B}_{T_e}|_{\mathcal{S}}] = (\boldsymbol{\lambda}^*|_{\mathcal{S}})^\top \mathbb{E}[(T - T_e + 1)\boldsymbol{\rho}_{T_e}|_{\mathcal{S}}] \\ &\overset{(a)}{\leq} n(\rho^{\max} + D)\|\boldsymbol{\lambda}^*\|_\infty \cdot \mathbb{E}[T - T_e + 1]. \end{aligned} \tag{10}$$

557 In the above, recall that $\rho^{\max}$ denotes the maximum coordinate of $\boldsymbol{\rho}_1$, and $D$ is specified in
558 Lemma C.2. Consequently, (a) is due to the definition of $T_e$ and that $\|\boldsymbol{\rho}_1 + D\mathbf{1}\|_\infty \le \rho^{\max} + D$.

559 We are left to bound $\mathbb{E}[T - T_e]$. Nevertheless, this part would be rather technical and involved.
560 Therefore we defer the analysis to Appendix C.5, and only give the final bounds.

561 **Lemma C.4.** *Under Assumption 3.1, with full information feedback, we have when $T \to \infty$:*

$$(\boldsymbol{\lambda}^*)^\top \mathbb{E}\left[\boldsymbol{B}_{T_e+1}\right] + \max_{\theta\in\Theta, a\in A^+} \left(R(\theta, a) - (\boldsymbol{\lambda}^*)^\top \cdot \boldsymbol{C}(\theta, a)\right) \mathbb{E}\left[T - T_e\right] = O(1).$$

562 Combining Lemmas C.1, C.3 and C.4, we arrive at Theorem 3.1.

### C.2 Proof of Lemma C.1

564 The proof is obtained by the following set of (in)equalities.

$$V^{\mathrm{FL}} - Rew$$

$$= T \cdot J(\boldsymbol{\rho}_1) - \mathbb{E}\left[\sum_{t=1}^{T_0} r(\theta_t, a_t, \gamma_t)\right]$$

$$\overset{(a)}{\le} T \cdot J(\boldsymbol{\rho}_1) - \mathbb{E}\left[\sum_{t=1}^{T_e} r(\theta_t, a_t, \gamma_t)\right]$$

$$\overset{(b)}{=} T \cdot J(\boldsymbol{\rho}_1) - \mathbb{E}\left[\sum_{t=1}^{T_e} \sum_{\theta\in\Theta, a\in A^+} u(\theta)R(\theta, a)\widehat{\phi}_t^*(\theta, a)\right]$$

$$\overset{(c)}{=} T \cdot \left(\sum_{\theta\in\Theta, a\in A^+} \left(u(\theta)(R(\theta, a) - (\boldsymbol{\lambda}^*)^\top \boldsymbol{C}(\theta, a)) - \mu^*(\theta)\right)\phi_1^*(\theta, a) + (\boldsymbol{\lambda}^*)^\top \boldsymbol{\rho}_1 + \sum_{\theta\in\Theta}\mu^*(\theta)\right)$$

$$- \mathbb{E}\left[\sum_{t=1}^{T_e} \sum_{\theta\in\Theta, a\in A^+} u(\theta)R(\theta, a)\widehat{\phi}_t^*(\theta, a)\right]$$

$$\overset{(d)}{=} \mathbb{E}\left[\sum_{t=1}^{T_e} \sum_{\theta\in\Theta, a\in A^+} \left(u(\theta)\left(R(\theta, a) - (\boldsymbol{\lambda}^*)^\top \boldsymbol{C}(\theta, a)\right) - \mu^*(\theta)\right)\left(\phi_1^*(\theta, a) - \widehat{\phi}_t^*(\theta, a)\right)\right]$$

$$+ \mathbb{E}\left[\sum_{t=1}^{T_e} \sum_{\theta\in\Theta}\mu^*(\theta)\left(1 - \sum_{a\in A^+}\widehat{\phi}_t^*(\theta, a)\right)\right] + \left(\sum_{\theta\in\Theta^*}\mu^*(\theta)\left(1 - \sum_{a\in A^+}\phi_1^*(\theta, a)\right)\right) \cdot \mathbb{E}\left[T - T_e\right]$$

$$+ (\boldsymbol{\lambda}^*)^\top \mathbb{E}\left[T\boldsymbol{\rho}_1 - \sum_{t=1}^{T_e} \sum_{\theta\in\Theta, a\in A^+} u(\theta)\boldsymbol{C}(\theta, a)\widehat{\phi}_t^*(\theta, a)\right]$$

$$+ \left(\sum_{\theta\in\Theta, a\in A^+} \left(u(\theta)(R(\theta, a) - (\boldsymbol{\lambda}^*)^\top \boldsymbol{C}(\theta, a))\right)\phi_1^*(\theta, a)\right) \cdot \mathbb{E}\left[T - T_e\right]$$

$$\overset{(e)}{\le} \mathbb{E}\left[\sum_{t=1}^{T_e} \sum_{\theta\in\Theta, a\in A^+} \left(u(\theta)\left(R(\theta, a) - (\boldsymbol{\lambda}^*)^\top \boldsymbol{C}(\theta, a)\right) - \mu^*(\theta)\right)\left(\phi_1^*(\theta, a) - \widehat{\phi}_t^*(\theta, a)\right)\right]$$

$$+ \mathbb{E}\left[\sum_{t=1}^{T_e} \sum_{\theta\in\Theta}\mu^*(\theta)\left(1 - \sum_{a\in A^+}\widehat{\phi}_t^*(\theta, a)\right)\right]$$

$$+ (\boldsymbol{\lambda}^*)^\top \mathbb{E}\left[\boldsymbol{B}_{T_e+1}\right] + \max_{\theta\in\Theta, a\in A^+}\left(R(\theta, a) - (\boldsymbol{\lambda}^*)^\top \boldsymbol{C}(\theta, a)\right) \cdot \mathbb{E}\left[T - T_e\right].$$

In the above set of derivations, (a) holds since $T_0 \geq T_e$, (b) is due to Optional Stopping Theorem since $T_e$ is a stopping time, (c) is by the strong duality of $J(\boldsymbol{\rho}_1)$ as given by (4), (d) establishes by rearranging terms. At last, for (e), the diminishing term is by strong duality, the transformation from the fourth term in (d) to the third term in (e) is derived by another application of Optional Stopping Theorem on the accumulated consumption vector, and for the last term, the upper bound is achieved since $\sum_{a \in A^+} \phi_1^*(\theta, a) \leq 1$ for any $\theta \in \Theta$ and $\sum_{\theta \in \Theta} u(\theta) = 1$.

## C.3 Proof of Lemma C.2

We will apply the stability result in Chen et al. [2022] as an intermediate to prove our version. As given, we know that $J(\boldsymbol{\rho}_1)$ and $\widehat{J}(\boldsymbol{\rho}_t, \mathcal{H}_t)$ has the same set of basic/non-basic variables and binding/non-binding constraints as long as the following conditions hold for some constant $D_0 > 0$:

$$
\left\| \left( u(\theta) \sum_{\gamma} v(\gamma) r(\theta, a, \gamma) - \widehat{u}_t(\theta) \sum_{\gamma} \widehat{v}_t(\gamma) r(\theta, a, \gamma) \right)_{(\theta, a) \in \Theta \times A^+} \right\|_\infty \leq D_0,
$$

$$
\left\| \left( u(\theta) \sum_{\gamma} v(\gamma) \boldsymbol{c}^i(\theta, a, \gamma) - \widehat{u}_t(\theta) \sum_{\gamma} \widehat{v}_t(\gamma) \boldsymbol{c}^i(\theta, a, \gamma) \right)_{(\theta, a) \in \Theta \times A^+} \right\|_\infty \leq D_0, \quad \forall i \in [n], \tag{11}
$$

$$
\|\boldsymbol{\rho}_1|_{\mathcal{S}} - \boldsymbol{\rho}_t|_{\mathcal{S}}\|_\infty \leq D_0, \quad \max\{\boldsymbol{\rho}_1|_{\mathcal{T}} - \boldsymbol{\rho}_t|_{\mathcal{T}}\} \leq D_0.
$$

Now, by a standard insertion technique, we have

$$
u(\theta) \sum_{\gamma} v(\gamma) r(\theta, a, \gamma) - \widehat{u}_t(\theta) \sum_{\gamma} \widehat{v}_t(\gamma) r(\theta, a, \gamma)
$$

$$
= (u(\theta) - \widehat{u}_t(\theta)) \sum_{\gamma} v(\gamma) r(\theta, a, \gamma) + \widehat{u}_t(\theta) \sum_{\gamma} (v(\gamma) - \widehat{v}_t(\gamma)) r(\theta, a, \gamma)
$$

$$
\overset{(a)}{\leq} \|(u(\theta) - \widehat{u}_t(\theta))_{\theta \in \Theta}\|_\infty + \|(v(\gamma) - \widehat{v}_t(\gamma))_{\gamma \in \Gamma}\|_1. \tag{12}
$$

For (a), the first term is bounded since $r(\theta, a, \gamma) \leq 1$ and $\sum_{\gamma} v(\gamma) = 1$. The second term is similarly bounded as $\widehat{u}_t(\theta) \leq 1$. Therefore, we let $D = D_0/2$, then when we have

$$
\|(u(\theta) - \widehat{u}_t(\theta))_{\theta \in \Theta}\|_\infty \leq D, \quad \|(v(\gamma) - \widehat{v}_t(\gamma))_{\gamma \in \Gamma}\|_1 \leq D,
$$

the first condition in (11) is met. An almost identical reasoning also holds for the second condition in (11). Consequently we finish the proof of the lemma.

## C.4 Proof of Lemma C.3

Recall that we are going to prove that

$$
\mathbb{E}\left[ \sum_{t=1}^{T_e} \sum_{\theta \in \Theta, a \in A^+} \left( u(\theta) \left( R(\theta, a) - (\boldsymbol{\lambda}^*)^\top \boldsymbol{C}(\theta, a) \right) - \mu^*(\theta) \right) \left( \phi_1^*(\theta, a) - \widehat{\phi}_t^*(\theta, a) \right) \right] = O(1),
$$

$$
\mathbb{E}\left[ \sum_{t=1}^{T_e} \sum_{\theta \in \Theta} \mu^*(\theta) \left( 1 - \sum_{a \in A^+} \widehat{\phi}_t^*(\theta, a) \right) \right] = O(1),
$$

when $T \to \infty$ under Assumption 3.1. For simplicity, we give the following abbreviations:

$$
P_t := \sum_{\theta \in \Theta, a \in a^+} \left( u(\theta) \left( R(\theta, a) - (\boldsymbol{\lambda}^*)^\top \boldsymbol{C}(\theta, a) \right) - \mu^*(\theta) \right) \left( \phi_1^*(\theta, a) - \widehat{\phi}_t^*(\theta, a) \right),
$$

$$
Q_t := \sum_{\theta \in \Theta} \mu^*(\theta) \left( 1 - \sum_{a \in A^+} \widehat{\phi}_t^*(\theta, a) \right),
$$

$$
\mathcal{E}_{u,t} := [\|(u(\theta) - \widehat{u}_t(\theta))_{\theta \in \Theta}\|_\infty \leq D], \quad \mathcal{E}_{v,t} := [\|(v(\gamma) - \widehat{v}_t(\gamma))_{\gamma \in \Gamma}\|_1 \leq D].
$$

On this end, we first utilize Lemma C.2 to show that when condition (6) holds, we have

$$P_t = Q_t = 0.$$

Specifically, for $P_t$, by the dual feasibility of $J(\boldsymbol{\rho}_1)$, we have $u(\theta)\left(R(\theta,a) - (\boldsymbol{\lambda}^*)^\top \boldsymbol{C}(\theta,a)\right) - \mu^*(\theta) \leq 0$. When $u(\theta)\left(R(\theta,a) - (\boldsymbol{\lambda}^*)^\top \boldsymbol{C}(\theta,a)\right) - \mu^*(\theta) < 0$, by primal optimality, $\phi_1^*(\theta,a) = 0$ and thus $(\theta,a)$ is non-basic for $J(\boldsymbol{\rho}_1)$. By Lemma C.2, $(\theta,a)$ is also non-basic for $\widehat{J}(\boldsymbol{\rho}_t, \mathcal{H}_t)$ and $\widehat{\phi}_t^*(\theta,a) = 0$ holds as well. In conjunction with the case that $u(\theta)\left(R(\theta,a) - (\boldsymbol{\lambda}^*)^\top \boldsymbol{C}(\theta,a)\right) - \mu^*(\theta) = 0$, we obtain that $P_t = 0$.

For $Q_t$, notice that we have $\mu^*(\theta) \geq 0$ for any $\theta \in \Theta$. The case that $\mu^*(\theta) = 0$, again, does not contribute to the total sum. When $\mu^*(\theta) > 0$, by complementary slackness, $\sum_{a \in A^+} \phi_1^*(\theta,a) = 1$, i.e., $\theta$ is a binding constraint for $J(\boldsymbol{\rho}_1)$. This, by Lemma C.2, implies that $\theta$ is also binding for $\widehat{J}(\boldsymbol{\rho}_t, \mathcal{H}_t)$, which shows that the second term is also zero.

With the above, it remains to consider the situation that condition (6) does not hold when $t \leq T_e$, or in other words, $\mathcal{E}_{u,t} \wedge \mathcal{E}_{v,t}$ does not hold. Note that $P_t \leq 1$ and $Q_t \leq 1$ always hold. Thus, we only need to bound the probability that $\neg(\mathcal{E}_{u,t} \wedge \mathcal{E}_{v,t})$. By a union bound, we have

$$\Pr[\neg(\mathcal{E}_{u,t} \wedge \mathcal{E}_{v,t})] = \Pr[\neg\mathcal{E}_{u,t} \vee \neg\mathcal{E}_{v,t}] \leq \Pr[\neg\mathcal{E}_{u,t}] + \Pr[\neg\mathcal{E}_{v,t}].$$

For the first term above, we apply the Hoeffding's inequality and a union bound to derive that

$$\Pr[\neg\mathcal{E}_{u,t}] = \Pr[\|(u(\theta) - \widehat{u}_t(\theta))_{\theta \in \Theta}\|_\infty > D] \leq 2|\Theta| \exp\left(-2D^2(t-1)\right).$$

Whereas for the second term, we use the concentration result in Weissman et al. [2003] to derive that

$$\Pr[\neg\mathcal{E}_{v,t}] = \Pr[\|(v(\gamma) - \widehat{v}_t(\gamma))_{\gamma \in \Gamma}\|_1 > D] \leq \left(2^{|\Gamma|} - 2\right) \exp\left(-D^2(t-1)/2\right).$$

Synthesizing the above all, we have

$$\begin{aligned}
\mathbb{E}[P_t] &= \mathbb{E}[P_t \mid \mathcal{E}_{u,t} \wedge \mathcal{E}_{v,t}] \cdot \Pr[\mathcal{E}_{u,t} \wedge \mathcal{E}_{v,t}] + \mathbb{E}[P_t \mid \neg(\mathcal{E}_{u,t} \wedge \mathcal{E}_{v,t})] \cdot \Pr[\neg(\mathcal{E}_{u,t} \wedge \mathcal{E}_{v,t})] \\
&\leq 0 + 1 \cdot \Pr[\neg(\mathcal{E}_{u,t} \wedge \mathcal{E}_{v,t})] \\
&\leq 2|\Theta| \exp\left(-2D^2(t-1)\right) + \left(2^{|\Gamma|} - 2\right) \exp\left(-D^2(t-1)/2\right), \quad (13) \\
\mathbb{E}[Q_t] &\leq 2|\Theta| \exp\left(-2D^2(t-1)\right) + \left(2^{|\Gamma|} - 2\right) \exp\left(-D^2(t-1)/2\right). \quad (14)
\end{aligned}$$

Summing (13) and (14) from 1 to $T_e$, we achieve that

$$\begin{aligned}
&\left\{\mathbb{E}\left[\sum_{t=1}^{T_e} P_t\right], \mathbb{E}\left[\sum_{t=1}^{T_e} Q_t\right]\right\} \\
&\leq \sum_{t=1}^{T}\left(2|\Theta| \exp\left(-2D^2(t-1)\right) + \left(2^{|\Gamma|} - 2\right) \exp\left(-D^2(t-1)/2\right)\right) \\
&\leq \frac{2|\Theta|}{1 - \exp\left(-2D^2\right)} + \frac{2^{|\Gamma|} - 2}{1 - \exp\left(-D^2/2\right)},
\end{aligned}$$

which conclude the proof of the lemma.

## C.5 Proof of Lemma C.4

As implied by (10), the proof of this lemma reduces to bound $\mathbb{E}[T - T_e]$, i.e., showing that $T_e$ is sufficiently close to $T$. On this side, we first recall the definition of $T_e$ in (9):

$$T_e := \min\{T_0, \min\{t : \max\{\|\boldsymbol{\rho}_1|_{\mathcal{S}} - \boldsymbol{\rho}_t|_{\mathcal{S}}\|_\infty, \max\{\boldsymbol{\rho}_1|_{\mathcal{T}} - \boldsymbol{\rho}_t|_{\mathcal{T}}\}\} > D\} - 1\},$$

where $T_0$ is the stopping time of Algorithm 1, and $\mathcal{S}$ and $\mathcal{T}$ correspondingly represent the set of binding/non-binding resource constraints in LP $J(\boldsymbol{\rho}_1)$. For simplicity, we define

$$\mathcal{N}(\boldsymbol{\rho}_1, D, \mathcal{S}) := \{\boldsymbol{\kappa} : \max\{\|\boldsymbol{\rho}_1|_{\mathcal{S}} - \boldsymbol{\kappa}|_{\mathcal{S}}\|_\infty, \max\{\boldsymbol{\rho}_1|_{\mathcal{T}} - \boldsymbol{\kappa}|_{\mathcal{T}}\}\} \leq D\}.$$

606  It is without loss of generality to suppose that $D < \rho^{\min}$. We let

$$T_D := \min\{t : \boldsymbol{\rho}_t \notin \mathcal{N}(\boldsymbol{\rho}_1, D, \mathcal{S})\} - 1, \quad T_- = \lfloor T + 1 - 1/(\rho^{\min} - D) \rfloor.$$

607  We show that if $t \leq T_-$ and $t \leq T_D$, then $t \leq T_e$. In fact, under the condition, we derive that

$$\boldsymbol{B}_t \geq (T - t + 1)(\boldsymbol{\rho}_1 - D\mathbf{1}) \geq \frac{1}{\rho^{\min} - D}(\boldsymbol{\rho}_1 - D\mathbf{1}) \geq \mathbf{1},$$

608  which implies that $t \leq T_0$, and therefore $t \leq T_e$. As a result, we have

$$\mathbb{E}[T_e] = \sum_{t=1}^{T} \Pr[T_e \geq t] \geq \sum_{t=1}^{T_-} \Pr[T_e \geq t] \geq \sum_{t=1}^{T_-} \Pr[T_D \geq t] = T_- - \sum_{t=1}^{T_-} \Pr[t > T_D]. \quad (15)$$

609  Before we continue to bound (15), we first give an observation on the dynamics of $\boldsymbol{\rho}_t$. By the update
610  process of the budget, we have for any $t \geq 1$,

$$\boldsymbol{B}_{t+1} = \boldsymbol{B}_t - \boldsymbol{c}_t \implies \boldsymbol{\rho}_{t+1}(T - t) = \boldsymbol{\rho}_t(T - t + 1) - \boldsymbol{c}_t$$

$$\implies \boldsymbol{\rho}_{t+1} = \boldsymbol{\rho}_t + \frac{\boldsymbol{\rho}_t - \boldsymbol{c}_t}{T - t}.$$

611  Now let

$$\boldsymbol{M}_t^C := \frac{\boldsymbol{\rho}_t - \mathbb{E}_{\theta \sim \mathcal{U}}\left[\sum_{a \in A^+} \widehat{\phi}_t^*(\theta, a)\boldsymbol{C}(\theta, a)\right]}{T - t}, \quad \boldsymbol{N}_t^C := \frac{\mathbb{E}_{\theta \sim \mathcal{U}}\left[\sum_{a \in A^+} \widehat{\phi}_t^*(\theta, a)\boldsymbol{C}(\theta, a)\right] - \boldsymbol{c}_t}{T - t}.$$

612  We then have

$$\boldsymbol{\rho}_{t+1} - \boldsymbol{\rho}_t = \frac{\boldsymbol{\rho}_t - \boldsymbol{c}_t}{T - t} = \boldsymbol{M}_t^C + \boldsymbol{N}_t^C. \quad (16)$$

613  We now define an auxiliary process which benefits the analysis. Specifically, for $t \in [T]$, let

$$\tilde{\boldsymbol{\rho}}_t := \begin{cases} \boldsymbol{\rho}_t, & t \leq T_D; \\ \boldsymbol{\rho}_{T_D}, & t > T_D. \end{cases}$$

614  Therefore,

$$\tilde{\boldsymbol{\rho}}_{t+1} - \tilde{\boldsymbol{\rho}}_t = \begin{cases} \boldsymbol{M}_t^C + \boldsymbol{N}_t^C, & t \leq T_D; \\ 0, & t > T_D. \end{cases}$$

615  We further define the following two auxiliary variables for $t \in [T]$:

$$\widetilde{\boldsymbol{M}}_t^C := \begin{cases} \boldsymbol{M}_t^C, & t \leq T_D; \\ 0, & t > T_D. \end{cases}, \quad \widetilde{\boldsymbol{N}}_t^C := \begin{cases} \boldsymbol{N}_t^C, & t \leq T_D; \\ 0, & t > T_D. \end{cases}$$

616  As a result, we have

$$\tilde{\boldsymbol{\rho}}_{t+1} - \tilde{\boldsymbol{\rho}}_t = \widetilde{\boldsymbol{M}}_t^C + \widetilde{\boldsymbol{N}}_t^C.$$

617  Now we come back to (15). Notice that

$$\Pr[t > T_D] \quad (17)$$
$$= \Pr[\boldsymbol{\rho}_s \notin \mathcal{N}(\boldsymbol{\rho}_1, D, \mathcal{S}) \text{ for some } s \leq t] = \Pr[\tilde{\boldsymbol{\rho}}_t \notin \mathcal{N}(\boldsymbol{\rho}_1, D, \mathcal{S})]$$
$$\leq \Pr\left[\left\|\sum_{\tau=1}^{t-1}\left(\widetilde{\boldsymbol{M}}_\tau^C + \widetilde{\boldsymbol{N}}_\tau^C\right)|_{\mathcal{S}}\right\|_\infty > D \text{ or } \min \sum_{\tau=1}^{t-1}\left(\widetilde{\boldsymbol{M}}_\tau^C + \widetilde{\boldsymbol{N}}_\tau^C\right)|_{\mathcal{T}} < -D\right]$$
$$\leq \Pr\left[\left\|\sum_{\tau=1}^{t-1}\widetilde{\boldsymbol{M}}_\tau^C|_{\mathcal{S}}\right\|_\infty > D/2 \text{ or } \min \sum_{\tau=1}^{t-1}\widetilde{\boldsymbol{M}}_\tau^C|_{\mathcal{T}} < -D/2\right] + \Pr\left[\left\|\sum_{\tau=1}^{t-1}\widetilde{\boldsymbol{N}}_\tau^C\right\|_\infty \geq D/2\right].$$
$$(18)$$

618  For the second term in (18), we observe that each entry of $\{\sum_{\tau < t} \widetilde{\boldsymbol{N}}_\tau^C\}_t$ is a martingale with the
619  absolute value of the $\tau$-th increment bounded by $1/(T - \tau)$. Since

$$\sum_{\tau=1}^{t-1} \frac{1}{(T - \tau)^2} \leq \frac{1}{T - t},$$

by applying the Azuma–Hoeffding inequality and a union bound, we achieve that

$$\Pr\left[\left\|\sum_{\tau=1}^{t-1}\widetilde{\boldsymbol{N}}_\tau^C\right\|_\infty \geq D/2\right] \leq 2n\exp\left(-\frac{(T-t)D^2}{8}\right).$$

We now come back to the first term in (18), for any $\{D_1,\cdots,D_{t-1}\}$ such that $\sum_{\tau=1}^{t-1} D_\tau/(T-\tau) \leq D/2$, we have

$$\left\{\left\|\sum_{\tau=1}^{t-1}\widetilde{\boldsymbol{M}}_\tau^C|_{\mathcal{S}}\right\|_\infty > D/2 \text{ or } \min\sum_{\tau=1}^{t-1}\widetilde{\boldsymbol{M}}_\tau^C|_{\mathcal{T}} < -D/2\right\}$$

$$\implies \left\{\left\|\widetilde{\boldsymbol{M}}_\tau^C|_{\mathcal{S}}\right\|_\infty > \frac{D_\tau}{T-\tau} \text{ or } \min\widetilde{\boldsymbol{M}}_\tau^C|_{\mathcal{T}} < -\frac{D_\tau}{T-\tau}\right\} \text{ for some } \tau \in [T-1].$$

We now define

$$\mathcal{E}_\tau(D_\tau) := \left(\left\|\boldsymbol{M}_\tau^C|_{\mathcal{S}}\right\|_\infty \leq \frac{D_\tau}{T-\tau}\right) \wedge \left(\min\boldsymbol{M}_\tau^C|_{\mathcal{T}} \geq -\frac{D_\tau}{T-\tau}\right) \text{ holds for } \forall\boldsymbol{\rho}_\tau \in \mathcal{N}(\boldsymbol{\rho}_1,D,\mathcal{S}).$$

Since $\widetilde{\boldsymbol{M}}_\tau^C \neq 0$ only when $t \leq T_D$, i.e., $\boldsymbol{\rho}_t \in \mathcal{N}(\boldsymbol{\rho}_1,D,\mathcal{S})$, by the definition of $\mathcal{E}_\tau(D_\tau)$, we have the following claim:

$$\left\{\left\|\sum_{\tau=1}^{t-1}\widetilde{\boldsymbol{M}}_\tau^C|_{\mathcal{S}}\right\|_\infty > D/2 \text{ or } \min\sum_{\tau=1}^{t-1}\widetilde{\boldsymbol{M}}_\tau^C|_{\mathcal{T}} < -D/2\right\} \subseteq \bigcup_{\tau=1}^{t-1}\neg\mathcal{E}_\tau(D_\tau), \quad \forall\sum_{\tau=1}^{t-1}\frac{D_\tau}{T-\tau} \leq D/2. \tag{19}$$

Thus, we forward to bound $\Pr[\neg\mathcal{E}_\tau(D_\tau)]$ for a suitable choice of $\{D_\tau\}_{1\leq\tau\leq T}$. Recall that we have defined events $\mathcal{E}_{u,\tau}$ and $\mathcal{E}_{v,\tau}$ as follows:

$$\mathcal{E}_{u,\tau} := [\|(u(\theta)-\widehat{u}_\tau(\theta))_{\theta\in\Theta}\|_\infty \leq D], \quad \mathcal{E}_{v,\tau} := [\|(v(\gamma)-\widehat{v}_\tau(\gamma))_{\gamma\in\Gamma}\|_1 \leq D].$$

We have the following lemma, which we are going to prove in Appendix C.6:

**Lemma C.5.** *When* $\boldsymbol{\rho}_\tau \in \mathcal{N}(\boldsymbol{\rho}_1,D,\mathcal{S})$ *and* $\mathcal{E}_{u,\tau}\wedge\mathcal{E}_{v,\tau}$ *hold,*

$$(T-\tau)\left\|\boldsymbol{M}_\tau^C|_{\mathcal{S}}\right\|_\infty \leq \|(u(\theta)-\widehat{u}_\tau(\theta))_{\theta\in\Theta}\|_1 + \|(v(\gamma)-\widehat{v}_\tau(\gamma))_{\gamma\in\Gamma}\|_1,$$
$$(T-\tau)\min\boldsymbol{M}_\tau^C|_{\mathcal{T}} \geq -\|(u(\theta)-\widehat{u}_\tau(\theta))_{\theta\in\Theta}\|_1 - \|(v(\gamma)-\widehat{v}_\tau(\gamma))_{\gamma\in\Gamma}\|_1.$$

Further, it is clear that $(T-\tau)\left\|\boldsymbol{M}_\tau^C|_{\mathcal{S}}\right\|_\infty \leq 1$ and $(T-\tau)\boldsymbol{M}_\tau^C|_{\mathcal{T}} \geq -1$ holds. Inspired by the above observations, we let the series of $D_1,\cdots,D_{T-1}$ be the following form:

$$D_\tau = \begin{cases} 1, & \tau \leq \eta T; \\ (\tau-1)^{-1/4}, & \tau > \eta T, \end{cases}$$

where $\eta \in (0,1)$ is a constant to be specified. We need to satisfy the following constraints:

$$\sum_{t=1}^{T-1}\frac{D_t}{T-t} \leq D/2, \quad (\eta T)^{-1/4} < D.$$

Here, the first constraint is instructed by (19), and the second is to guarantee that when $\|(u(\theta)-\widehat{u}_\tau(\theta))_{\theta\in\Theta}\|_1 + \|(v(\gamma)-\widehat{v}_\tau(\gamma))_{\gamma\in\Gamma}\|_1 < (\tau-1)^{-1/4}$ for $\tau > \eta T$, $\mathcal{E}_{u,\tau}\wedge\mathcal{E}_{v,\tau}$ naturally holds, and therefore we can apply Lemma C.5. For the first one, we notice that

$$\sum_{\tau=1}^{T-1}\frac{D_\tau}{T-\tau} = \sum_{\tau=1}^{\eta T}\frac{1}{T-\tau} + \sum_{\tau=\eta T+1}^{T-1}\frac{1}{(T-\tau)(\tau-1)^{1/4}} \leq \log\frac{T-1}{(1-\eta)T-1} + \frac{\log T}{(\eta T)^{1/4}}.$$

634 Therefore, for some $\eta$ such that $\log(1-\eta) \geq -D/4$, $\sum_{t=1}^{T} D_t/(T-t) \leq D/2$ establishes for
635 sufficiently large $T \gg 1$, and the second constraint is also satisfied.

636 We are now prepared to bound $\Pr[\neg\mathcal{E}_\tau(D_\tau)]$ for the $\{D_\tau\}$ we just proposed. To start with, when
637 $\tau \leq \eta T$, $\mathcal{E}_\tau(D_\tau)$ always holds, thus $\Pr[\neg\mathcal{E}_\tau(D_\tau)] = 0$. When $\tau > \eta T$, since $\tau^{-1/4}/2 < D$, by
638 Hoeffding's inequality and union bound, we have

$$\Pr[\neg\mathcal{E}_\tau(D_\tau)]$$

$$\stackrel{(a)}{\leq} \Pr\left[\|(u(\theta) - \widehat{u}_\tau(\theta))_{\theta\in\Theta}\|_1 \leq (\tau-1)^{-1/4}/2\right] + \Pr\left[\|(v(\gamma) - \widehat{v}_\tau(\gamma))_{\gamma\in\Gamma}\|_1 \leq (\tau-1)^{-1/4}/2\right]$$

$$\leq 2|\Theta|\exp\left(-\frac{(\tau-1)^{1/2}}{8|\Theta|^2}\right) + 2|\Gamma|\exp\left(-\frac{(\tau-1)^{1/2}}{8|\Gamma|^2}\right).$$

639 Here, (a) is by Lemma C.5 and a union bound. Therefore, according to (19), we have

$$\Pr\left[\left\|\sum_{\tau=1}^{t-1}\widetilde{M}_\tau^C|_{\mathcal{S}}\right\|_\infty > D/2 \text{ or } \min\sum_{\tau=1}^{t-1}\widetilde{M}_\tau^C|_{\mathcal{T}} < -D/2\right] \leq \sum_{\tau=1}^{t-1}\Pr[\neg\mathcal{E}_\tau(D_\tau)],$$

and therefore,

$$\Pr\left[\left\|\sum_{\tau=1}^{t-1}\widetilde{M}_\tau^C|_{\mathcal{S}}\right\|_\infty > D/2 \text{ or } \min\sum_{\tau=1}^{t-1}\widetilde{M}_\tau^C|_{\mathcal{T}} < -D/2\right] \leq \begin{cases} 0, & t \leq \eta T + 1; \\ \sum_{\tau=\eta T+1}^{t-1}\exp\left\{-\tau^{1/2}\right\}, & t > \eta T + 1. \end{cases}$$

640 Plugging the into (18) and (15), we obtain that when $T \to \infty$,

$$\mathbb{E}[T - T_e]$$

$$\leq T - T_-$$

$$+ \sum_{t=1}^{T_-}\left(\Pr\left[\left\|\sum_{\tau=1}^{t-1}\widetilde{M}_\tau^C|_{\mathcal{S}}\right\|_\infty > D/2 \text{ or } \min\sum_{\tau=1}^{t-1}\widetilde{M}_\tau^C|_{\mathcal{T}} < -D/2\right] + 2n\exp\left(-\frac{(T-t)D^2}{8}\right)\right)$$

$$\leq \frac{1}{\rho^{\min} - D} + 2n(1 - \exp(-D^2/8))^{-1} + O(T^2)\exp\left(-T^{1/2}\right) = O(1).$$

641 At last, combining with (10), we finally finish the proof of Lemma C.4.

## C.6 Proof of Lemma C.5

643 To start with, we notice that

$$(T - \tau)M_\tau^C = \boldsymbol{\rho}_\tau - \mathbb{E}_{\theta\sim\mathcal{U}}\left[\sum_{a\in A^+}\widehat{\phi}_\tau^*(\theta, a)C(\theta, a)\right].$$

644 Now, notice that $\boldsymbol{\rho}_\tau \in \mathcal{N}(\boldsymbol{\rho}_1, D, \mathcal{S})$ and $\mathcal{E}_{u,\tau} \wedge \mathcal{E}_{v,\tau}$ are the condition of Lemma C.2, therefore, the
645 set of resource binding constraints of $\widehat{J}(\boldsymbol{\rho}_t, \mathcal{H}_t)$ are identical to that of $J(\boldsymbol{\rho}_1)$, i.e., $\mathcal{S}$. Hence, for any
646 $i \in [n]$,

$$\boldsymbol{\rho}_\tau^i|_{\mathcal{S}} - \mathbb{E}_{\theta\sim\mathcal{U}}\left[\sum_{a\in A^+}\widehat{\phi}_\tau^*(\theta, a)\boldsymbol{C}^i(\theta, a)|_{\mathcal{S}}\right]$$

$$= \sum_{\theta\in\Theta, a\in A^+}\widehat{u}_\tau(\theta)\widehat{\phi}_\tau^*(\theta, a)\sum_\gamma\widehat{v}_\tau(\gamma)\boldsymbol{c}^i(\theta, a, \gamma)|_{\mathcal{S}} - \sum_{\theta\in\Theta, a\in A^+}u(\theta)\widehat{\phi}_\tau^*(\theta, a)\sum_\gamma v(\gamma)\boldsymbol{c}^i(\theta, a, \gamma)|_{\mathcal{S}}$$

$$= \sum_{\theta\in\Theta, a\in A^+}(u(\theta) - \widehat{u}_\tau(\theta))\widehat{\phi}_\tau^*(\theta, a)\sum_\gamma v(\gamma)\boldsymbol{c}^i(\theta, a, \gamma)|_{\mathcal{S}}$$

$$+ \sum_{\theta\in\Theta, a\in A^+}\widehat{u}_\tau(\theta)\widehat{\phi}_\tau^*(\theta, a)\sum_\gamma(\widehat{v}_\tau(\gamma) - v(\gamma))\boldsymbol{c}^i(\theta, a, \gamma)|_{\mathcal{S}}$$

$$\stackrel{(a)}{\leq} \|(u(\theta) - \widehat{u}_\tau(\theta))_{\theta\in\Theta}\|_1 + \|(v(\gamma) - \widehat{v}_\tau(\gamma))_{\gamma\in\Gamma}\|_1.$$

Here, the bound on the first term in (a) establishes because for any $\theta \in \Theta$,

$$\sum_{a \in A^+} \widehat{\phi}_\tau^*(\theta, a) \sum_\gamma v(\gamma) \boldsymbol{c}^i(\theta, a, \gamma)|_{\mathcal{S}} \leq 1$$

since $\sum_{a \in A^+} \widehat{\phi}_\tau^*(\theta, a) \leq 1$. The bound on the second term is similar. Thus, we achieve the result for binding constraints. The proof for non-binding constraints resembles the above by noticing that

$$\boldsymbol{\rho}_\tau|_{\mathcal{T}} \geq \sum_{\theta \in \Theta, a \in A^+} \widehat{u}_\tau(\theta) \widehat{\phi}_\tau^*(\theta, a) \sum_\gamma \widehat{v}_\tau(\gamma) \boldsymbol{c}(\theta, a, \gamma)|_{\mathcal{T}}.$$

# D  Missing Proofs in Section 4

## D.1  Proof of Theorem 4.1

With Lemma 4.1 in hand, we now show how to derive Theorem 4.1. Specifically, the regret decomposition technique in Lemma C.1 still works fine. We only need to re-derive corresponding results for Lemmas C.3 and C.4. We have the following results on this side, which are proved respectively in Appendices D.3 and D.4.

**Lemma D.1.** *Under Assumption 3.1, with partial information feedback, we have when $T \to \infty$:*

$$\mathbb{E}\left[\sum_{t=1}^{T_e} \sum_{\theta \in \Theta, a \in A^+} \left(u(\theta)\left(R(\theta, a) - (\boldsymbol{\lambda}^*)^\top \boldsymbol{C}(\theta, a)\right) - \mu^*(\theta)\right)\left(\phi_1^*(\theta, a) - \widehat{\phi}_t^*(\theta, a)\right)\right] = O(1),$$

$$\mathbb{E}\left[\sum_{t=1}^{T_e} \sum_{\theta \in \Theta} \mu^*(\theta)\left(1 - \sum_{a \in A^+} \widehat{\phi}_t^*(\theta, a)\right)\right] = O(\log T).$$

**Lemma D.2.** *Under Assumption 3.1, with partial information feedback, we have when $T \to \infty$:*

$$(\boldsymbol{\lambda}^*)^\top \mathbb{E}\left[\boldsymbol{B}_{T_e+1}\right] + \max_{\theta \in \Theta, a \in A^+} \left(R(\theta, a) - (\boldsymbol{\lambda}^*)^\top \cdot \boldsymbol{C}(\theta, a)\right) \mathbb{E}\left[T - T_e\right] = O(1).$$

Lemmas C.1, D.1 and D.2 in together leads to Theorem 4.1.

## D.2  Proof of Lemma 4.1

Some preparations are required before we come to prove the lemma. To start with, we notice that $Y_\tau = \Pr[a_1 \neq 0] + \cdots + \Pr[a_{t-1} \neq 0]$. By the control rule of Algorithm 1, we have

$$\Pr[a_\tau \neq 0] = \mathbb{E}_{\theta \sim \mathcal{U}}\left[\sum_{a \in A^+} \widehat{\phi}_\tau^*(\theta, a) \mid \mathcal{H}_\tau\right].$$

We first give a lower bound on $\mathbb{E}_{\theta \sim \mathcal{U}}[\sum_{a \in A^+} \widehat{\phi}_\tau^*(\theta, a) \mid \mathcal{H}_\tau]$ with $\boldsymbol{\rho}_\tau$, taking $\mathbb{E}_{\theta \sim \widehat{\mathcal{U}}_\tau}[\sum_{a \in A^+} \widehat{\phi}_\tau^*(\theta, a) \mid \mathcal{H}_\tau]$ as an intermediate.

**Lemma D.3.**

$$\mathbb{E}_{\theta \sim \widehat{\mathcal{U}}_\tau}\left[\sum_{a \in A^+} \widehat{\phi}_\tau^*(\theta, a) \mid \mathcal{H}_\tau\right] \geq \min\left\{1, \min \boldsymbol{\rho}_\tau\right\}.$$

*Proof of Lemma D.3.* To start with, when $\boldsymbol{\rho}_\tau \geq 1$, then clearly, all the resource constraints in $\widehat{J}(\boldsymbol{\rho}_\tau, \mathcal{H}_\tau)$ are satisfied even when $\sum_{a \in A^+} \phi(\theta, a) = 1$ holds for any $\theta \in \Theta$. Therefore, an optimal solution should have this form.

We now consider the case that $\min \boldsymbol{\rho}_\tau < 1$. In this case, if there is a feasible solution that $\sum_{a,\in A^+} \widehat{\phi}_\tau^*(\theta, a) = 1$ holds for any $\theta \in \Theta$, then the proof is also finished. Otherwise, there is at least a binding resource constraint in $\widehat{J}(\boldsymbol{\rho}_\tau, \mathcal{H}_\tau)$, which we denote by $i^*$. Consequently,

$$\mathbb{E}_{\theta \sim \widehat{\mathcal{U}}_\tau} \left[ \sum_{a \in A^+} \widehat{\phi}_\tau^*(\theta, a) \right] \geq \mathbb{E}_{\theta \sim \widehat{\mathcal{U}}_\tau} \left[ \sum_{a \in A^+} \widehat{\phi}_\tau^*(\theta, a) \widehat{\boldsymbol{C}}_\tau^{i^*}(\theta, a) \right] = \boldsymbol{\rho}_\tau^{i^*} \geq \min \boldsymbol{\rho}_\tau.$$

This finishes the proof of the lemma. $\qquad \square$

Thus, we have

$$\Pr[a_\tau \neq 0] = \mathbb{E}_{\theta \sim \mathcal{U}} \left[ \sum_{a \in A^+} \widehat{\phi}_\tau^*(\theta, a) \mid \mathcal{H}_\tau \right] \geq \mathbb{E}_{\theta \sim \widehat{\mathcal{U}}_\tau} \left[ \sum_{a \in A^+} \widehat{\phi}_\tau^*(\theta, a) \mid \mathcal{H}_\tau \right] - \|u(\theta) - \widehat{u}_\tau(\theta)\|_1$$

$$\geq \min \{1, \min \boldsymbol{\rho}_\tau\} - \|u(\theta) - \widehat{u}_\tau(\theta)\|_1. \qquad (20)$$

Further, we have the following result bounding $\min \boldsymbol{\rho}_\tau$ when $t$ is no larger than a fraction of $T$.

**Lemma D.4.** *When $t \leq (\rho^{\min}/2) \cdot T$, $\min \boldsymbol{\rho}_\tau \geq \rho^{\min}/2$.*

*Proof of Lemma D.4.* In fact, for $t \leq (\rho^{\min}/2) \cdot T$,

$$\boldsymbol{\rho}_\tau = \frac{T \cdot \boldsymbol{\rho}_1 - \sum_{\tau=1}^{t-1} \boldsymbol{c}_\tau}{T - t + 1} \geq \frac{T \cdot \boldsymbol{\rho}_1 - t \cdot \mathbf{1}}{T} \geq \frac{\boldsymbol{\rho}_1}{2}.$$

This concludes the proof. $\qquad \square$

Now, by Weissman et al. [2003], with probability $1 - O(1/T)$, we have

$$\|u(\theta) - \widehat{u}_\tau(\theta)\|_1 \leq \frac{\rho^{\min}}{4}, \quad \forall \tau \geq \Theta(\log T).$$

Taking into (20), we derive that

$$\Pr[a_\tau \neq 0] \geq \frac{\rho^{\min}}{4}, \quad \forall \Theta(\log T) \leq t \leq \frac{\rho^{\min}}{2} \cdot T.$$

Consequently, within the period, the probability that there are $\Omega(\log T)$ consecutive rounds in which the agent chooses to quit in all these rounds is $O(1/T)$. This proves the first part. Meanwhile, at time $t = \lceil (\rho^{\min}/2) \cdot T \rceil + 1$, by Azuma–Hoeffding inequality, we derive that with probability $1 - O(1/T)$, $Y_t = \sum_{\tau=1}^{t-1} \Pr[a_\tau \neq 0] \geq \Omega(T)$, which proves the second part.

## D.3 Proof of Lemma D.1

We concentrate on adapting the proof of Lemma C.3 into the partial information feedback setting. To start with, we suppose that the conditions given in Lemma 4.1 hold. In fact, since the failure probability is only $O(1/T)$, and the sum is upper bounded by $O(T)$, therefore the failure case only contributes $O(1)$ to the total expectation.

Now, recall the following definitions:

$$P_t := \sum_{\theta \in \Theta, a \in a^+} \left( u(\theta) \left( R(\theta, a) - (\boldsymbol{\lambda}^*)^\top \boldsymbol{C}(\theta, a) \right) - \mu^*(\theta) \right) \left( \phi_1^*(\theta, a) - \widehat{\phi}_t^*(\theta, a) \right),$$

$$Q_t := \sum_{\theta \in \Theta} \mu^*(\theta) \left( 1 - \sum_{a \in A^+} \widehat{\phi}_t^*(\theta, a) \right),$$

$$\mathcal{E}_{u,t} := [\|(u(\theta) - \widehat{u}_t(\theta))_{\theta \in \Theta}\|_\infty \leq D], \quad \mathcal{E}_{v,t} := [\|(v(\gamma) - \widehat{v}_t(\gamma))_{\gamma \in \Gamma}\|_1 \leq D],$$

and by (13) and (14), we have

$$\mathbb{E}[P_t] \leq \Pr[\neg \mathcal{E}_{u,t}] + \Pr[\neg \mathcal{E}_{v,t}], \quad \mathbb{E}[Q_t] \leq \Pr[\neg \mathcal{E}_{u,t}] + \Pr[\neg \mathcal{E}_{v,t}].$$

Now, the bound on $\Pr[\neg\mathcal{E}_{u,t}]$ inherits the analysis in the proof of Lemma C.3, as partial information feedback does not affect the learning of the request distribution. That is,

$$\Pr[\neg\mathcal{E}_{u,t}] = \Pr[\|(u(\theta) - \widehat{u}_t(\theta))_{\theta\in\Theta}\|_\infty > D] \le 2|\Theta| \exp\left(-2D^2(t-1)\right).$$

For $\Pr[\neg\mathcal{E}_{v,t}]$, when $t \le \Theta(\log T)$, it is obviously bounded by 1. By Lemma 4.1, when $\Theta(\log T) \le t \le C_b \cdot T$, by Weissman et al. [2003], we have

$$\Pr[\neg\mathcal{E}_{v,t}] = \Pr[\|(v(\gamma) - \widehat{v}_t(\gamma))_{\gamma\in\Gamma}\|_1 > D] \le \left(2^{|\Gamma|} - 2\right)\exp\left(-\frac{D^2 C_f(t-1)}{2\log T}\right).$$

Further, when $t > C_b \cdot T$, we correspondingly derive

$$\Pr[\neg\mathcal{E}_{v,t}] \le \left(2^{|\Gamma|} - 2\right)\exp\left(-\frac{D^2 C_r(t-1)}{2}\right).$$

Putting the above together, we achieve that

$$\left\{\mathbb{E}\left[\sum_{t=1}^{T_e} P_t\right], \mathbb{E}\left[\sum_{t=1}^{T_e} Q_t\right]\right\}$$

$$\le \sum_{t=1}^{T} 2|\Theta|\exp\left(-2D^2(t-1)\right) + \Theta(\log T)$$

$$+ \left(2^{|\Gamma|} - 2\right)\left(\sum_{t=\Theta(\log T)}^{C_b\cdot T}\exp\left(-\frac{D^2 C_f(t-1)}{2\log T}\right) + \sum_{t=C_b\cdot T+1}^{T}\exp\left(-\frac{D^2 C_r(t-1)}{2}\right)\right)$$

$$\le \Theta(1) + \Theta(\log T) + \left(2^{|\Gamma|} - 2\right)\left(\frac{\Theta(1)}{1 - \exp\left(-\Theta(1/\log T)\right)} + \exp(-\Theta(T))\right)$$

$$\le O(\log T).$$

Here, for the last inequality, by Taylor expansion, we have $1 - e^{-x} \ge x - x^2/2$ for $x > 0$, therefore,

$$\frac{1}{1 - \exp\left(-\Theta(1/\log T)\right)} \le \frac{1}{\Theta(1/\log T - 1/\log^2 T)} \le \Theta(\log T).$$

This finishes the proof.

## D.4 Proof of Lemma D.2

As in Appendix D.3 when we prove Lemma C.3, we only consider the case when the conditions in Lemma 4.1 establish, as the contribution of the failure cases on the expectation-sum is $O(1)$. We now bound $\mathbb{E}[T - T_e]$ in the good case when the sample accessing frequency under partial information feedback is guaranteed. Specifically, as predefined in the proof of Lemma C.3, we only need to re-calculate the following, as the other terms remain unchanged with partial information:

$$\sum_{t=\eta T+2}^{T_-}\sum_{\tau=\eta T+1}^{t-1}\Pr\left[\|(v(\gamma) - \widehat{v}_\tau(\gamma))_{\gamma\in\Gamma}\|_1 \le (\tau-1)^{-1/4}/2\right].$$

Here, $\eta$ is specified in the definition of $D_\tau$. It is hard for us to directly compare $\eta$ and $C_b$ in Lemma 4.1. Nevertheless, in any case, we know that when $T$ is sufficiently large, $Y_\tau/(\tau-1) = \Omega(1/\log T)$ for $\tau \ge \eta T$. Therefore, we have

$$\Pr\left[\|(v(\gamma) - \widehat{v}_\tau(\gamma))_{\gamma\in\Gamma}\|_1 \le (\tau-1)^{-1/4}/2\right] \le 2|\Gamma|\exp\left(-\frac{(\tau-1)^{1/2}}{|\Gamma|^2 O(\log T)}\right).$$

Hence,

$$\sum_{t=\eta T+2}^{T_-} \sum_{\tau=\eta T+1}^{t-1} \Pr\left[ \|(v(\gamma) - \widehat{v}_\tau(\gamma))_{\gamma\in\Gamma}\|_1 \le (\tau-1)^{-1/4}/2 \right]$$

$$\le 2|\Gamma| \sum_{t=\eta T+2}^{T_-} \sum_{\tau=\eta T+1}^{t-1} \exp\left( -\frac{(\tau-1)^{1/2}}{|\Gamma|^2 O(\log T)} \right)$$

$$= O(T^2) \exp\left( -\Omega\left( \frac{T^{1/2}}{\log T} \right) \right) = O(1).$$

Combining with the other parts, Lemma D.2 is proved.

# E  Missing Proofs in Section 5

## E.1  Proof of Theorem 5.1

We will prove Theorem 5.1 in the following, and we are inspired by the analysis in Chen et al. [2022].

### E.1.1  Another Regret Decomposition

Different from our analysis for the regular cases, in general circumstances, we introduce another regret decomposition method. The reason for involving such an alternative is that without the regularity assumptions, we no longer have any local stability guarantee even when the estimates are close. Therefore, the decision given by Algorithm 1 does not coincides with the optimal decision even when the distribution learning process converges well, and the corresponding analysis in Section 3 does not work out anymore.

We now present a more general regret decomposition as follows:

$$V^{\mathrm{FL}} - Rew = T \cdot J(\boldsymbol{\rho}_1) - \mathbb{E}\left[ \sum_{t=1}^{T_0} r(\theta_t, a_t, \gamma_t) \right]$$

$$\overset{(a)}{=} T \cdot J(\boldsymbol{\rho}_1) - \mathbb{E}\left[ \sum_{t=1}^{T_0} \mathbb{E}_\theta\left[ \sum_{a\in A^+} \widehat{\phi}_t^*(\theta, a) R(\theta, a) \right] \right] \tag{21}$$

$$\overset{(b)}{=} J(\boldsymbol{\rho}_1) \cdot \mathbb{E}\left[ T - T_0 \right] + \mathbb{E}\left[ \sum_{t=1}^{T_0} \left( J(\boldsymbol{\rho}_1) - \mathbb{E}_\theta\left[ \sum_{a\in A^+} \widehat{\phi}_t^*(\theta, a) R(\theta, a) \right] \right) \right].$$

Here, (a) holds due to the Optimal Stopping Theorem, since $T_0$ is a stopping time. Meanwhile, by the decision process, we have for any $\theta_t$:

$$\mathbb{E}_{a_t,\gamma_t}[r(\theta_t, a_t, \gamma_t) \mid \theta_t] = \sum_{a\in A^+} \widehat{\phi}_t^*(\theta_t, a) R(\theta_t, a).$$

Further, (b) is by a re-arrangement. To give a bound for (21), we respectively analyze $\mathbb{E}[T - T_0]$ the stopping time, and difference between the optimal accumulated rewards and the real ones.

### E.1.2  Bounding the Stopping Time

To settle the stopping time, we first reduce it to $\max(\boldsymbol{\rho}_1 - \boldsymbol{\rho}_t, 0)$ for $t \le T_0$, and then deals with these values. We notice that $t \le T_0$ as long as that $\boldsymbol{B}_t \ge \mathbf{1}$, or $\boldsymbol{\rho}_t \ge \mathbf{1}/(T-t+1)$. Now, since for any $i \in [n]$,

$$\boldsymbol{\rho}_t^i = \boldsymbol{\rho}_1^i - (\boldsymbol{\rho}_1 - \boldsymbol{\rho}_t)^i \ge \rho^{\min} - \max(\boldsymbol{\rho}_1 - \boldsymbol{\rho}_t, 0),$$

we have $\min \boldsymbol{\rho}_t \ge \rho^{\min} - \max(\boldsymbol{\rho}_1 - \boldsymbol{\rho}_t, 0)$. Therefore,

$$t \le T_0 \impliedby \rho^{\min} - \max(\boldsymbol{\rho}_1 - \boldsymbol{\rho}_t, 0) \ge \frac{1}{T-t+1}$$

$$\iff \max(\boldsymbol{\rho}_1 - \boldsymbol{\rho}_t, 0) \le \rho^{\min} - \frac{1}{T-t+1}. \tag{22}$$

Since $\mathbb{E}[T_0] \geq \Pr[T_0 \geq t] \cdot t$ for any $t \in [T]$, we only need to bound the following term for some certain $t$:

$$\Pr\left[\max\left(\boldsymbol{\rho}_1 - \boldsymbol{\rho}_t, 0\right) \leq \rho^{\min} - \frac{1}{T - t + 1}\right].$$

We will further prove the following lemma in Appendix E.3:

**Lemma E.1.** *It holds for any $t < T$ that*

$$\Pr\left[\max\left(\boldsymbol{\rho}_1 - \boldsymbol{\rho}_t, 0\right) \geq \Theta\left(\frac{1}{T-1} + \sum_{\tau=2}^{t-1}\sqrt{\frac{\log T}{(T-\tau)^2(\tau-1)}} + \sqrt{\frac{\log T}{T-t}}\right)\right] \leq O\left(\frac{1}{T}\right).$$

With the light of Lemma E.1, it is natural for us to compute

$$\sum_{\tau=2}^{t-1}\sqrt{\frac{\log T}{(T-\tau)^2(\tau-1)}} \leq \begin{cases} \sqrt{\log T} \cdot \dfrac{4\sqrt{t-2}}{T-1}, & 2 \leq t \leq (T+1)/2; \\ \sqrt{\log T} \cdot \dfrac{2}{\sqrt{T-t}}, & t > (T+1)/2. \end{cases}$$

In fact, to derive the above, we notice that when $2 \leq t \leq (T+1)/2$,

$$\sum_{\tau=1}^{t-1}\frac{1}{(T-\tau)(\tau-1)^{1/2}} \leq \frac{2}{T-1}\sum_{\tau=2}^{t-1}\frac{1}{(\tau-1)^{1/2}} \leq \frac{4\sqrt{t-1}}{T-1}.$$

Meanwhile, when $t > (T+1)/2$, we have $T - t < t - 1$, which leads to

$$\sum_{\tau=2}^{t-1}\frac{1}{(T-\tau)(\tau-1)^{1/2}} \leq \sqrt{\frac{8}{T-1}} + \sum_{\tau=(T+1)/2}^{t-1}\frac{1}{(T-\tau)^{3/2}} \leq \frac{2}{\sqrt{T-t}}.$$

With these calculations, we come back to the bound on $\mathbb{E}[T_0]$, we notice that when $T$ is sufficiently large and $t = T - O(\log T)$, it holds that

$$\Theta\left(\frac{1}{T-1} + \sum_{\tau=2}^{t-1}\sqrt{\frac{\log T}{(T-\tau)^2(\tau-1)}} + \sqrt{\frac{\log T}{T-t}}\right) + \frac{1}{T-t+1} = O(1) \leq \rho^{\min}.$$

Thus, we have

$$\mathbb{E}\left[T - T_0\right] = T - \mathbb{E}[T_0] \overset{(a)}{\leq} T - \Pr[T_0 \geq T - O(\log T)] \cdot (T - O(\log T))$$

$$\overset{(b)}{\leq} T - \left(1 - O\left(\frac{1}{T}\right)\right) \cdot (T - O(\log T)) = O(\log T). \tag{23}$$

In the above, (a) is because $\mathbb{E}[T_0] \geq \Pr[T_0 \geq t] \cdot t$ for any fixed $t$, and (b) is due to Lemma E.1. Consequently, we finish the analysis of the stopping time in (21).

### E.1.3 The Gap to the Optimal Reward

The rest part of (21) that we are left to consider is the following:

$$J(\boldsymbol{\rho}_1) - \mathbb{E}_\theta\left[\sum_{a \in A^+}\widehat{\phi}_t^*(\theta, a)R(\theta, a)\right]$$

$$= \left(J(\boldsymbol{\rho}_1) - \widehat{J}(\boldsymbol{\rho}_t, \mathcal{H}_t)\right) + \left(\widehat{J}(\boldsymbol{\rho}_t, \mathcal{H}_t) - \mathbb{E}_\theta\left[\sum_{a \in A^+}\widehat{\phi}_t^*(\theta, a)R(\theta, a)\right]\right). \tag{24}$$

Note that the second difference term in (24) reflects the estimation error on distributions of the context and the external factor, which leads to the following result as to be proved in Appendix E.4:

**Lemma E.2.** *We have for $t \geq 2$:*

$$\mathbb{E}\left[\widehat{J}(\boldsymbol{\rho}_t, \mathcal{H}_t) - \mathbb{E}_\theta\left[\sum_{a \in A^+} \widehat{\phi}_t^*(\theta, a) R(\theta, a)\right]\right] \leq O\left(\sqrt{\frac{\log T}{t-1}} + \frac{1}{T}\right).$$

Lemma E.2 induces an $O(\sqrt{T \log T})$ accumulated regret considering (24) when summing from $t = 2$ to $T_0 \leq T$. While for the first term in (24), our main thread here is to bound $\widehat{J}(\boldsymbol{\rho}_t, \mathcal{H}_t)$ with $J(\boldsymbol{\rho}_t)$. To fix the idea, we compare these two optimization problems:

$$J(\boldsymbol{\rho}_t) := \max_{\phi:\Theta \times A^+ \to \mathbb{R}_+} \sum_{\theta \in \Theta, a \in A^+} u(\theta)\phi(\theta, a) \sum_\gamma r(\theta, a, \gamma) v(\gamma),$$

$$\text{s.t.} \quad \sum_{\theta \in \Theta, a \in A^+} u(\theta)\phi(\theta, a) \sum_\gamma \boldsymbol{c}(\theta, a, \gamma) v(\gamma) \leq \boldsymbol{\rho}_t,$$

$$\sum_{a \in A^+} \phi(\theta, a) \leq 1, \quad \forall \theta \in \Theta,$$

$$\phi(\theta, a) \geq 0, \quad \forall(\theta, a) \in \Theta \times A^+.$$

$$\widehat{J}(\boldsymbol{\rho}_t, \mathcal{H}_t) := \max_{\phi:\Theta \times A^+ \to \mathbb{R}_+} \sum_{\theta \in \Theta, a \in A^+} \widehat{u}_t(\theta)\phi(\theta, a) \sum_\gamma r(\theta, a, \gamma) \widehat{v}_t(\gamma),$$

$$\text{s.t.} \quad \sum_{\theta \in \Theta, a \in A^+} \widehat{u}_t(\theta)\phi(\theta, a) \sum_\gamma \boldsymbol{c}(\theta, a, \gamma) \widehat{v}_t(\gamma) \leq \boldsymbol{\rho}_t,$$

$$\sum_{a \in A^+} \phi(\theta, a) \leq 1, \quad \forall \theta \in \Theta,$$

$$\phi(\theta, a) \geq 0, \quad \forall(\theta, a) \in \Theta \times A^+.$$

Now, conceptually, if there is a $0 < \eta_t \leq 1$ such that for any $(\theta, a) \in \Theta \times A^+$,

$$u(\theta) \sum_\gamma \boldsymbol{c}(\theta, a, \gamma) v(\gamma) \geq \eta_t \widehat{u}_t(\theta) \sum_\gamma \boldsymbol{c}(\theta, a, \gamma) \widehat{v}_t(\gamma),$$

then for an optimal solution $\phi_t^*$ of $J(\boldsymbol{\rho}_t)$, we see that $\eta_t \phi_t^*$ is a feasible solution of the programming $\widehat{J}(\boldsymbol{\rho}_t, \mathcal{H}_t)$. Thus,

$$\widehat{J}(\boldsymbol{\rho}_t, \mathcal{H}_t) \geq \eta_t \sum_{\theta \in \Theta, a \in A^+} \widehat{u}_t(\theta)\phi_t^*(\theta, a) \sum_\gamma r(\theta, a, \gamma) \widehat{v}_t(\gamma)$$

$$\overset{(a)}{\geq} \eta_t \sum_{\theta \in \Theta, a \in A^+} u(\theta)\phi_t^*(\theta, a) \sum_\gamma r(\theta, a, \gamma) v(\gamma)$$

$$- \eta_t(\|(u(\theta) - \widehat{u}_t(\theta))_{\theta \in \Theta}\|_1 + \|(v(\gamma) - \widehat{v}_t(\gamma))_{\gamma \in \Gamma}\|_1)$$

$$= \eta_t J(\boldsymbol{\rho}_t) - (\|(u(\theta) - \widehat{u}_t(\theta))_{\theta \in \Theta}\|_1 + \|(v(\gamma) - \widehat{v}_t(\gamma))_{\gamma \in \Gamma}\|_1).$$

Here, since $r(\theta, a, \gamma) \leq 1$ and $\sum_{a \in A^+} \widehat{\phi}_t^*(\theta, a) \leq 1$ for any $\theta$, (a) is expanded as

$$\sum_{\theta \in \Theta, a \in A^+} \widehat{u}_t(\theta)\phi_t^*(\theta, a) \sum_\gamma \widehat{v}_t(\gamma) r(\theta, a, \gamma) - \sum_{\theta \in \Theta, a \in A^+} u(\theta)\phi_t^*(\theta, a) \sum_\gamma v(\gamma) r(\theta, a, \gamma)$$

$$= \sum_{\theta \in \Theta, a \in A^+} (\widehat{u}_t(\theta) - u(\theta))\phi_t^*(\theta, a) \sum_\gamma \widehat{v}_t(\gamma) r(\theta, a, \gamma)$$

$$+ \sum_{\theta \in \Theta, a \in A^+} u(\theta)\phi_t^*(\theta, a) \sum_\gamma (\widehat{v}_t(\gamma) - v(\gamma)) r(\theta, a, \gamma)$$

$$\leq \|(u(\theta) - \widehat{u}_t(\theta))_{\theta \in \Theta}\|_1 + \|(v(\gamma) - \widehat{v}_t(\gamma))_{\gamma \in \Gamma}\|_1.$$

Consequently,

$$J(\boldsymbol{\rho}_1) - \widehat{J}(\boldsymbol{\rho}_t, \mathcal{H}_t)$$
$$\leq (1 - \eta_t)J(\boldsymbol{\rho}_1) + \eta_t(J(\boldsymbol{\rho}_1) - J(\boldsymbol{\rho}_t)) + \|(u(\theta) - \widehat{u}_t(\theta))_{\theta \in \Theta}\|_1 + \|(v(\gamma) - \widehat{v}_t(\gamma))_{\gamma \in \Gamma}\|_1. \quad (25)$$

On top of this, a key observation is that

$$J(\boldsymbol{\rho}_1) - J(\boldsymbol{\rho}_t) \leq \frac{\max(\boldsymbol{\rho}_1 - \boldsymbol{\rho}_t, 0)}{\rho^{\min}} \cdot J(\boldsymbol{\rho}_1). \quad (26)$$

In fact, when $\boldsymbol{\rho}_1 \leq \boldsymbol{\rho}_t$, (26) is natural as $J(\boldsymbol{\rho}_1) \leq J(\boldsymbol{\rho}_t)$. Otherwise, let $\phi_1^*$ be the optimal solution to the programming $J(\boldsymbol{\rho}_1)$. Let $i^*$ be the index that minimizes $\boldsymbol{\rho}_t^{i^*}/\boldsymbol{\rho}_1^{i^*}$. We have $\boldsymbol{\rho}_1^{i^*} > \boldsymbol{\rho}_t^{i^*}$. Evidently, we know that $\phi_1^* \cdot \boldsymbol{\rho}_t^{i^*}/\boldsymbol{\rho}_1^{i^*}$ is a feasible solution to the programming of $J(\boldsymbol{\rho}_t)$. By the optimality of $J(\boldsymbol{\rho}_t)$, we have

$$J(\boldsymbol{\rho}_t) \geq \frac{\boldsymbol{\rho}_t^{i^*}}{\boldsymbol{\rho}_1^{i^*}} \cdot J(\boldsymbol{\rho}_1),$$

which leads to

$$J(\boldsymbol{\rho}_1) - J(\boldsymbol{\rho}_t) \leq \left(1 - \frac{\boldsymbol{\rho}_t^{i^*}}{\boldsymbol{\rho}_1^{i^*}}\right) \cdot J(\boldsymbol{\rho}_1) = \frac{\boldsymbol{\rho}_1^{i^*} - \boldsymbol{\rho}_t^{i^*}}{\boldsymbol{\rho}_1^{i^*}} \cdot J(\boldsymbol{\rho}_1) \leq \frac{\max(\boldsymbol{\rho}_1 - \boldsymbol{\rho}_t)}{\rho^{\min}} \cdot J(\boldsymbol{\rho}_1).$$

Synthesizing the above two parts, (26) is proved.

As for $\mathbb{E}[\max(\boldsymbol{\rho}_1 - \boldsymbol{\rho}_t, 0)]$, we note that for any non-negative random variable $X$ with upper bound $\bar{X}$ and any positive $\xi$, we have

$$\mathbb{E}[X] \leq \xi \Pr[X \leq \xi] + \bar{X}(1 - \Pr[X \leq \xi]) \leq \xi + \bar{X}(1 - \Pr[X \leq \xi]). \quad (27)$$

Notice that $\max(\boldsymbol{\rho}_1 - \boldsymbol{\rho}_t, 0)$ is certainly upper bounded by 1. Therefore, as a corollary of Lemma E.1, we have

$$\mathbb{E}[\max(\boldsymbol{\rho}_1 - \boldsymbol{\rho}_t, 0)] \leq \begin{cases} O\left(\dfrac{\sqrt{(t-2)\log T}}{T} + \sqrt{\dfrac{\log T}{T-t}} + \dfrac{1}{T}\right), & 2 \leq t \leq (T+1)/2; \\[4mm] O\left(\sqrt{\dfrac{\log T}{T-t}} + \dfrac{1}{T}\right), & t > (T+1)/2. \end{cases}$$

We almost finish the bound now except for determining $\eta_t$ in (25), which we hope is asclose to 1 as possible. Nevertheless, we leave the technical parts to Appendix E.5 which derives the following lemma on the total bound:

**Lemma E.3.**
$$\mathbb{E}\left[\sum_{t=1}^{T_0}\left(J(\boldsymbol{\rho}_1) - \widehat{J}(\boldsymbol{\rho}_t, \mathcal{H}_t)\right)\right] = O(\sqrt{T \log T}).$$

Now, we sum the result in (21) from $t = 2$ to $T_0$, and plus the constant term for $t = 1$ to obtain that

$$\mathbb{E}\left[\sum_{t=1}^{T_0}\left(\widehat{J}(\boldsymbol{\rho}_t, \mathcal{H}_t) - \mathbb{E}_\theta\left[\sum_{a \in A^+} \widehat{\phi}_t^*(\theta, a) R(\theta, a)\right]\right)\right] = O(\sqrt{T \log T}).$$

Synthesizing Lemma E.3, (24), (23), and (21), we derive Theorem 5.1.

## E.2 Proof of Theorem 5.2

The proof of this theorem follows the line of Theorem 5.1, and the only difference is to adopt Lemma 4.1 when considering the concentration of estimates. On this side, we can disregard the cases when $t \leq \Theta(\log T)$, as the accumulated regret in this phase is bounded by $O(\log T)$. On the other hand, the time range that $t \geq \Theta(T)$ is asymptotically identical to the full information setting since

the accessing frequency is a constant. We only need to consider the case that $\Theta(\log T) \leq t \leq \Theta(T)$, when we have

$$
\Pr\left[\|(u(\theta) - \widehat{u}_t(\theta))_{\theta \in \Theta}\|_1 \leq -\Theta\left(\sqrt{\frac{\log T}{t-1}}\right)\right] \leq O\left(\frac{1}{T^2}\right),
$$
$$
\Pr\left[\|(v(\gamma) - \widehat{v}_t(\gamma))_{\gamma \in \Gamma}\|_1 \leq -\Theta\left(\frac{\log T}{\sqrt{t-1}}\right)\right] \leq O\left(\frac{1}{T^2}\right).
$$
(28)

Taking into the proof of Lemma E.1 and then into the main body, we should find a sufficient large $t$ such that

$$
\Theta\left(\frac{\log T}{T-1} + \sum_{\tau=\Theta(\log T)}^{\Theta(T)} \frac{\log T}{\sqrt{(T-\tau)^2(\tau-1)}} + \sum_{\tau=\Theta(T)}^{t-1} \sqrt{\frac{\log T}{(T-\tau)^2(\tau-1)}} + \sqrt{\frac{\log T}{T-t}}\right)
$$
$$
\leq \rho^{\min} - \frac{1}{T-t+1},
$$

and $t = T - O(\log T)$ still suffices. Therefore, $\mathbb{E}[T - T_0] = O(\log T)$ also holds under partial information feedback.

Nevertheless, for the counterpart of Lemma E.2, by (28), when we sum from $t = 1$ to $T_0 \leq T$, we derive that

$$
\mathbb{E}\left[\sum_{t=1}^{T_0} \left(\widehat{J}(\boldsymbol{\rho}_t, \mathcal{H}_t) - \mathbb{E}_{\theta \sim \mathcal{U}}\left[\sum_{a \in A^+} \widehat{\phi}_t^*(\theta, a) R(\theta, a)\right]\right)\right] \leq O(\sqrt{T} \log T).
$$

At last, for $J(\boldsymbol{\rho}_1) - \widehat{J}(\boldsymbol{\rho}_t, \mathcal{H}_t)$, we face the same degradation on the estimation accuracy, which leads to

$$
\mathbb{E}\left[\sum_{t=1}^{T_0} \left(J(\boldsymbol{\rho}_1) - \widehat{J}(\boldsymbol{\rho}_t, \mathcal{H}_t)\right)\right] = O(\sqrt{T} \log T).
$$

Therefore, Theorem 5.2 is achieved.

### E.3 Proof of Lemma E.1

Now that we are going to bound $\max(\boldsymbol{\rho}_1 - \boldsymbol{\rho}_t, 0)$. Recall the definitions below which we give in Appendix C.5 when we prove Lemma C.4:

$$
\boldsymbol{M}_t^C := \frac{\boldsymbol{\rho}_t - \mathbb{E}_{\theta \sim \mathcal{U}}\left[\sum_{a \in A^+} \widehat{\phi}_t^*(\theta, a) \boldsymbol{C}(\theta, a)\right]}{T-t}, \quad \boldsymbol{N}_t^C := \frac{\mathbb{E}_{\theta \sim \mathcal{U}}\left[\sum_{a \in A^+} \widehat{\phi}_t^*(\theta, a) \boldsymbol{C}(\theta, a)\right] - \boldsymbol{c}_t}{T-t}.
$$

By (16), we have

$$
\boldsymbol{\rho}_{t+1} - \boldsymbol{\rho}_t = \frac{\boldsymbol{\rho}_t - \boldsymbol{c}_t}{T-t} = \boldsymbol{M}_t^C + \boldsymbol{N}_t^C.
$$

Consequently,

$$
\max(\boldsymbol{\rho}_1 - \boldsymbol{\rho}_t) = \max\left(-\left(\sum_{\tau=1}^{t-1} \boldsymbol{M}_\tau^C + \sum_{\tau=1}^{t-1} \boldsymbol{N}_\tau^C\right)\right) \leq -\min\sum_{\tau=1}^{t-1} \boldsymbol{M}_\tau^C - \min\sum_{\tau=1}^{t-1} \boldsymbol{N}_\tau^C.
$$

For the second term, we notice that each entry of $\{\sum_{\tau<t} \boldsymbol{N}_\tau^C\}_t$ is a martingale with the absolute value of the $\tau$-th increment bounded by $1/(T-\tau)$. Since

$$
\sum_{\tau=1}^{t-1} \frac{1}{(T-\tau)^2} \leq \frac{1}{T-t},
$$

by applying the Azuma–Hoeffding inequality and a union bound, we achieve that

$$
\Pr\left[-\min\sum_{\tau=1}^{t-1} \boldsymbol{N}_\tau^C \geq \sqrt{\frac{2\log T}{T-t}}\right] \leq \frac{n}{T}.
$$
(29)

On the other hand, for the first term, when $\tau = 1$, it is apparent that $-\min \boldsymbol{M}_1^C \leq 1/(T-1)$. When $\tau \geq 2$, we have for any $i \in [n]$,

$$
(T-\tau)\left(\boldsymbol{M}_\tau^C\right)^i
$$

$$
= \boldsymbol{\rho}_\tau^i - \mathbb{E}_{\theta \sim \mathcal{U}}\left[\sum_{a \in A^+} \widehat{\phi}_\tau^*(\theta, a) \boldsymbol{C}^i(\theta, a)\right]
$$

$$
\overset{(a)}{\geq} \sum_{\theta \in \Theta, a \in A^+} \widehat{u}_\tau(\theta)\widehat{\phi}_\tau^*(\theta, a) \sum_\gamma \widehat{v}_\tau(\gamma)\boldsymbol{c}^i(\theta, a, \gamma) - \sum_{\theta \in \Theta, a \in A^+} u(\theta)\widehat{\phi}_\tau^*(\theta, a) \sum_\gamma v(\gamma)\boldsymbol{c}^i(\theta, a, \gamma)
$$

$$
= \sum_{\theta \in \Theta, a \in A^+} (\widehat{u}_\tau(\theta) - u(\theta))\widehat{\phi}_\tau^*(\theta, a) \sum_\gamma \widehat{v}_\tau(\gamma)\boldsymbol{c}^i(\theta, a, \gamma)
$$

$$
+ \sum_{\theta \in \Theta, a \in A^+} u(\theta)\widehat{\phi}_\tau^*(\theta, a) \sum_\gamma (\widehat{v}_\tau(\gamma) - v(\gamma))\boldsymbol{c}^i(\theta, a, \gamma)
$$

$$
\geq -\|(u(\theta) - \widehat{u}_\tau(\theta))_{\theta \in \Theta}\|_1 - \|(v(\gamma) - \widehat{v}_\tau(\gamma))_{\gamma \in \Gamma}\|_1.
$$

In the above, (a) is because $\widehat{\phi}_\tau^*$ is feasible for $\widehat{J}(\boldsymbol{\rho}_\tau, \mathcal{H}_\tau)$. By Hoeffding's inequality and a union bound, we have

$$
\Pr\left[\|(u(\theta) - \widehat{u}_\tau(\theta))_{\theta \in \Theta}\|_1 \leq -|\Theta|\sqrt{\frac{\log T}{\tau - 1}}\right] \leq \frac{|\Theta|}{T^2},
$$

$$
\Pr\left[\|(v(\gamma) - \widehat{v}_\tau(\gamma))_{\gamma \in \Gamma}\|_1 \leq -|\Gamma|\sqrt{\frac{\log T}{\tau - 1}}\right] \leq \frac{|\Gamma|}{T^2}.
$$

Thus, suppose the above events hold for all $\tau \leq T$ with failure probability only $O(1/T)$,

$$
\Pr\left[-\min \sum_{\tau=1}^{t-1} \boldsymbol{M}_\tau^C \geq \Theta\left(\frac{1}{T-1} + \sum_{\tau=2}^{t-1}\sqrt{\frac{\log T}{(T-\tau)^2(\tau-1)}}\right)\right] \leq O\left(\frac{1}{T}\right). \tag{30}
$$

Combining (29) and (30), we derive the lemma.

## E.4 Proof of Lemma E.2

We notice that
$$
\widehat{J}(\boldsymbol{\rho}_t, \mathcal{H}_t) = \sum_{\theta \in \Theta, a \in A^+} \widehat{u}_t(\theta)\widehat{\phi}_t^*(\theta, a) \sum_\gamma r(\theta, a, \gamma)\widehat{v}_t(\gamma),
$$

and

$$
\sum_{\theta \in \Theta, a \in A^+} \widehat{u}_t(\theta)\widehat{\phi}_t^*(\theta, a) \sum_\gamma \widehat{v}_t(\gamma)r(\theta, a, \gamma) - \sum_{\theta \in \Theta, a \in A^+} u(\theta)\widehat{\phi}_t^*(\theta, a) \sum_\gamma v(\gamma)r(\theta, a, \gamma)
$$

$$
= \sum_{\theta \in \Theta, a \in A^+} (\widehat{u}_t(\theta) - u(\theta))\widehat{\phi}_t^*(\theta, a) \sum_\gamma \widehat{v}_t(\gamma)r(\theta, a, \gamma)
$$

$$
+ \sum_{\theta \in \Theta, a \in A^+} u(\theta)\widehat{\phi}_t^*(\theta, a) \sum_\gamma (\widehat{v}_t(\gamma) - v(\gamma))r(\theta, a, \gamma)
$$

$$
\leq \|(u(\theta) - \widehat{u}_t(\theta))_{\theta \in \Theta}\|_1 + \|(v(\gamma) - \widehat{v}_t(\gamma))_{\gamma \in \Gamma}\|_1.
$$

Thus,

$$
\widehat{J}(\boldsymbol{\rho}_t, \mathcal{H}_t) - \mathbb{E}_{\theta \sim \mathcal{U}}\left[\sum_{a \in A^+} \widehat{\phi}_t^*(\theta, a)R(\theta, a)\right] \leq \|(u(\theta) - \widehat{u}_t(\theta))_{\theta \in \Theta}\|_1 + \|(v(\gamma) - \widehat{v}_t(\gamma))_{\gamma \in \Gamma}\|_1.
$$

By Hoeffding's inequality and a union bound, we have

$$\Pr\left[\|(u(\theta) - \widehat{u}_t(\theta))_{\theta \in \Theta}\|_1 \geq |\Theta|\sqrt{\frac{\log T}{2(t-1)}}\right] \leq \frac{|\Theta|}{T},$$

$$\Pr\left[\|(v(\gamma) - \widehat{v}_t(\gamma))_{\gamma \in \Gamma}\|_1 \geq |\Gamma|\sqrt{\frac{\log T}{2(t-1)}}\right] \leq \frac{|\Gamma|}{T}.$$

Further, the difference we hope to analyze is certainly upper bounded by 1. As a result, with (27), we finish the proof.

### E.5 Proof of Lemma E.3

We come to consider $J(\boldsymbol{\rho}_1) - \widehat{J}(\boldsymbol{\rho}_t, \mathcal{H}_t)$. As per the thread in the main body, we let

$$\delta_t := \frac{\|(u(\theta) - \widehat{u}_t(\theta))_{\theta \in \Theta}\|_\infty + \|(v(\gamma) - \widehat{v}_t(\gamma))_{\gamma \in \Gamma}\|_1}{\min_{\theta \in \Theta, a \in A^+}\{\min\{u(\theta)\boldsymbol{C}(\theta, a) > 0\}\}}.$$

We now claim that for any $(\theta, a, i) \in \Theta \times A^+ \times [n]$,

$$\widehat{u}_t(\theta)\sum_\gamma \boldsymbol{c}^i(\theta, a, \gamma)\widehat{v}_t(\gamma) \leq (1 + \delta_t)u(\theta)\sum_\gamma \boldsymbol{c}^i(\theta, a, \gamma)v(\gamma).$$

The above is obvious if $\boldsymbol{C}^i(\theta, a) = \boldsymbol{0}$, or $\boldsymbol{c}^i(\theta, a, \gamma) = 0$ holds for any $\gamma$. When $\boldsymbol{C}(\theta, a) \neq \boldsymbol{0}$, then for any $i \in [n]$,

$$\widehat{u}_t(\theta)\sum_\gamma \boldsymbol{c}^i(\theta, a, \gamma)\widehat{v}_t(\gamma) - u(\theta)\sum_\gamma \boldsymbol{c}^i(\theta, a, \gamma)v(\gamma)$$

$$= (\widehat{u}_t(\theta) - u(\theta))\sum_\gamma \boldsymbol{c}^i(\theta, a, \gamma)\widehat{v}_t(\gamma) + u(\theta)\sum_\gamma \boldsymbol{c}^i(\theta, a, \gamma)(\widehat{v}_t(\gamma) - v(\gamma))$$

$$\leq \|(u(\theta) - \widehat{u}_t(\theta))_{\theta \in \Theta}\|_\infty + \|(v(\gamma) - \widehat{v}_t(\gamma))_{\gamma \in \Gamma}\|_1$$

$$\leq \delta_t u(\theta)\sum_\gamma \boldsymbol{c}^i(\theta, a, \gamma)v(\gamma).$$

This finish the explanation of the claim. Upon that, if we let $\eta_t := 1 - \delta_t \leq 1/(1 + \delta_t)$, we derive that

$$u(\theta)\sum_\gamma \boldsymbol{c}(\theta, a, \gamma)v(\gamma) \leq \frac{1}{1 + \delta_t}\widehat{u}_t(\theta)\sum_\gamma \boldsymbol{c}(\theta, a, \gamma)\widehat{v}_t(\gamma)$$

$$\leq \eta_t\widehat{u}_t(\theta)\sum_\gamma \boldsymbol{c}(\theta, a, \gamma)\widehat{v}_t(\gamma).$$

With respect to (25) and (26), we obtain that

$$J(\boldsymbol{\rho}_1) - \widehat{J}(\boldsymbol{\rho}_t, \mathcal{H}_t)$$

$$\leq J(\boldsymbol{\rho}_1) \cdot \left(1 - \eta_t + \frac{\max(\boldsymbol{\rho}_1 - \boldsymbol{\rho}_t, 0)}{\rho^{\min}}\right) + \|(u(\theta) - \widehat{u}_t(\theta))_{\theta \in \Theta}\|_1 + \|(v(\gamma) - \widehat{v}_t(\gamma))_{\gamma \in \Gamma}\|_1$$

$$= J(\boldsymbol{\rho}_1) \cdot \left(\delta_t + \frac{\max(\boldsymbol{\rho}_1 - \boldsymbol{\rho}_t, 0)}{\rho^{\min}}\right) + \|(u(\theta) - \widehat{u}_t(\theta))_{\theta \in \Theta}\|_1 + \|(v(\gamma) - \widehat{v}_t(\gamma))_{\gamma \in \Gamma}\|_1. \quad (31)$$

As we have already shown in the main body that

$$\mathbb{E}\left[\max(\boldsymbol{\rho}_1 - \boldsymbol{\rho}_t, 0)\right] \leq \begin{cases} O\left(\frac{\sqrt{(t-2)\log T}}{T} + \sqrt{\frac{\log T}{T-t}} + \frac{1}{T}\right), & 2 \leq t \leq (T+1)/2; \\ O\left(\sqrt{\frac{\log T}{T-t}} + \frac{1}{T}\right), & t > (T+1)/2, \end{cases}$$

it suffices for us to bound

$$\mathbb{E}[\|(u(\theta) - \widehat{u}_t(\theta))_{\theta \in \Theta}\|_\infty], \mathbb{E}[\|(u(\theta) - \widehat{u}_t(\theta))_{\theta \in \Theta}\|_1], \mathbb{E}[(v(\gamma) - \widehat{v}_t(\gamma))_{\gamma \in \Gamma}\|_1].$$

On this side, as we have shown that

$$\Pr\left[\|(u(\theta) - \widehat{u}_t(\theta))_{\theta \in \Theta}\|_1 \geq |\Theta|\sqrt{\frac{\log T}{2(t-1)}}\right] \leq \frac{|\Theta|}{T},$$

$$\Pr\left[\|(v(\gamma) - \widehat{v}_t(\gamma))_{\gamma \in \Gamma}\|_1 \geq |\Gamma|\sqrt{\frac{\log T}{2(t-1)}}\right] \leq \frac{|\Gamma|}{T},$$

it is natural that

$$\{\mathbb{E}[\|(u(\theta) - \widehat{u}_t(\theta))_{\theta \in \Theta}\|_\infty], \mathbb{E}[\|(u(\theta) - \widehat{u}_t(\theta))_{\theta \in \Theta}\|_1], \mathbb{E}[(v(\gamma) - \widehat{v}_t(\gamma))_{\gamma \in \Gamma}\|_1]\}$$

$$\leq O\left(\sqrt{\frac{\log T}{t-1}} + \frac{1}{T}\right).$$

Thus, putting all the above into (31) and summing from $t = 1$ to $T_0 \leq T$, we have

$$\mathbb{E}\left[\sum_{t=1}^{T_0} \left(J(\boldsymbol{\rho}_1) - \widehat{J}(\boldsymbol{\rho}_t, \mathcal{H}_t)\right)\right] = O(\sqrt{T \log T}).$$

This concludes the proof.

# F  Missing Details in Appendix A

## F.1  The Density Estimator

We now present details on the kernel density estimator which we apply in Appendix A for approximating continuous distributions, which comes from Wasserman [2019]. We consider a one-dimensional kernel function $K$ such that

- $\int K(x)\,dx = 1$;

- $\int x^s K(x)\,dx = 0, \quad \forall 1 \leq s \leq \beta$;

- $\int |x|^\beta |K(x)|\,dx < \infty$.

Now, given $k$ independent samples $X_1, \cdots, X_k$ from $P$ and a positive number $h$ called the bandwidth, the kernel density estimator is defined as

$$\widehat{p}_k(x) = \frac{1}{k}\sum_{i=1}^{k}\frac{1}{h^d}K\left(\frac{\|x - X_i\|_2}{h}\right).$$

Furthermore, to satisfy Proposition A.1, we should choose $h \asymp k^{1/(2\beta+d)}\log k$ when $p \in \Sigma(\beta, L)$ is the density of $\mathcal{P}$ on $\mathbb{R}^d$.

## F.2  Proof of Theorem A.1

By (21), we know that

$$V^{\mathrm{FL}} - Rew = J(\boldsymbol{\rho}_1) \cdot \mathbb{E}[T - T_0] + \mathbb{E}\left[\sum_{t=1}^{T_0}\left(J(\boldsymbol{\rho}_1) - \mathbb{E}_\theta\left[\sum_{a \in A^+}\widehat{\phi}_t^*(\theta, a)R(\theta, a)\right]\right)\right],$$

and we bound these terms in order. For the expected stopping time $\mathbb{E}[T_0]$, by the analysis in Section 5, our goal turns into bounding $\max(\boldsymbol{\rho}_1 - \boldsymbol{\rho}_t, 0)$, which further by (16) and (29), reduces to bound

831    $M_\tau^C$. With continuous randomness, we have for any $i \in [n]$,

$$
\begin{aligned}
& (T - \tau)\left(\boldsymbol{M}_\tau^C\right)^i \\
&= \boldsymbol{\rho}_\tau^i - \mathbb{E}_\theta\left[\sum_{a \in A^+} \widehat{\phi}_\tau^*(\theta, a)\boldsymbol{C}^i(\theta, a)\right] \\
&\overset{(a)}{\geq} \int_\theta \sum_{a \in A^+} \widehat{\phi}_\tau^*(\theta, a)\int_\gamma \boldsymbol{c}^i(\theta, a, \gamma)\widehat{v}_\tau(\gamma)\widehat{u}_\tau(\theta)\,\mathrm{d}\gamma\,\mathrm{d}\theta \\
&\quad - \int_\theta \sum_{a \in A^+} \widehat{\phi}_\tau^*(\theta, a)\int_\gamma \boldsymbol{c}^i(\theta, a, \gamma)v(\gamma)u(\theta)\,\mathrm{d}\gamma\,\mathrm{d}\theta \\
&= \int_\theta \sum_{a \in A^+} \widehat{\phi}_\tau^*(\theta, a)\int_\gamma \boldsymbol{c}^i(\theta, a, \gamma)\widehat{v}_\tau(\gamma)(\widehat{u}_\tau(\theta) - u(\theta))\,\mathrm{d}\gamma\,\mathrm{d}\theta \\
&\quad + \int_\theta \sum_{a \in A^+} \widehat{\phi}_\tau^*(\theta, a)\int_\gamma \boldsymbol{c}^i(\theta, a, \gamma)(\widehat{v}_\tau(\gamma) - v(\gamma))u(\theta)\,\mathrm{d}\gamma\,\mathrm{d}\theta \\
&\overset{(b)}{\geq} -\sup_\theta |u(\theta) - \widehat{u}_\tau(\theta)| - \sup_\gamma |(v(\gamma) - \widehat{v}_\tau(\gamma)|.
\end{aligned}
$$

832    In the above, (a) is by the constraint feasibility of $\widehat{\phi}_\tau^*$, and (b) is because $\sum_{a \in A^+} \widehat{\phi}_\tau^*(\theta, a) \leq 1$ holds
833    for any $\theta \in \Theta$. Further, by Proposition A.1, we have for $\tau = \Omega(1)$,

$$
\Pr\left[\sup_\theta |u(\theta) - \widehat{u}_\tau(\theta)| \leq -\Theta\left(\sqrt{\log T}(\tau - 1)^{\alpha_u - 1}\right)\right] \leq \frac{1}{T^2},
$$

$$
\Pr\left[\sup_\gamma |v(\theta) - \widehat{v}_\tau(\theta)| \leq -\Theta\left(\sqrt{\log T}(\tau - 1)^{\alpha_v - 1}\right)\right] \leq \frac{1}{T^2}.
$$

834    Thus, when $t = \Omega(1)$, we derive that with failure probability $O(1/T)$, it holds that

$$
\max\left(\boldsymbol{\rho}_1 - \boldsymbol{\rho}_t, 0\right) \leq \Theta\left(\frac{1}{T - 1} + \sqrt{\log T}\sum_{\tau = \Theta(1)}^{t-1}\left(\frac{(\tau - 1)^{\alpha_u - 1}}{T - \tau} + \frac{(\tau - 1)^{\alpha_v - 1}}{T - \tau}\right) + \sqrt{\frac{\log T}{T - t}}\right).
$$

835    Further, for $p \in \{u, v\}$, when $t \leq (T + 1)/2$,

$$
\sum_{\tau = \Theta(1)}^{t-1} \frac{(\tau - 1)^{\alpha_p - 1}}{T - \tau} \leq \frac{2}{T - 1}\sum_{\tau = 2}^{t-1}(\tau - 1)^{\alpha_p - 1} \leq \frac{2(t - 2)^{\alpha_p}}{\alpha_p(T - 1)};
$$

836    and when $t > (T + 1)/2$, we have

$$
\sum_{\tau = \Theta(1)}^{t-1} \frac{(\tau - 1)^{\alpha_p - 1}}{T - \tau} \leq \frac{1}{\alpha_p}\left(\frac{2}{T - 1}\right)^{1 - \alpha_p} + \sum_{\tau = (T+1)/2}^{t-1}(T - \tau)^{\alpha_p - 2} \leq \frac{(T - t)^{\alpha_p - 1}}{1 - \alpha_p}.
$$

837    Thus, when $t = T - \Theta(\log^{(2(1 - \max\{1/2, \alpha_u, \alpha_v\}))^{-1}} T)$, we have

$$
\begin{aligned}
&\Theta\left(\frac{1}{T - 1} + \sqrt{\log T}\sum_{\tau = \Theta(1)}^{t-1}\left(\frac{(\tau - 1)^{\alpha_u - 1}}{T - \tau} + \frac{(\tau - 1)^{\alpha_v - 1}}{T - \tau}\right) + \sqrt{\frac{\log T}{T - t}}\right) \\
&\leq \rho^{\min} - \frac{1}{T - t + 1},
\end{aligned}
$$

838    which leads to

$$
\mathbb{E}[T - T_0] = O\left(\log^{(2(1 - \max\{1/2, \alpha_u, \alpha_v\}))^{-1}} T\right).
$$

839    This concludes the analysis of the stopping time.

840 For the second part, By (24), we have

$$J(\boldsymbol{\rho}_1) - \mathbb{E}_\theta \left[ \sum_{a \in A^+} \widehat{\phi}_t^*(\theta, a) R(\theta, a) \right]$$

$$= \left( J(\boldsymbol{\rho}_1) - \widehat{J}(\boldsymbol{\rho}_t, \mathcal{H}_t) \right) + \left( \widehat{J}(\boldsymbol{\rho}_t, \mathcal{H}_t) - \mathbb{E}_\theta \left[ \sum_{a \in A^+} \widehat{\phi}_t^*(\theta, a) R(\theta, a) \right] \right).$$

841 On the second difference term, similar to the proof of Lemma E.2, we have

$$\widehat{J}(\boldsymbol{\rho}_t, \mathcal{H}_t) - \mathbb{E}_\theta \left[ \sum_{a \in A^+} \widehat{\phi}_t^*(\theta, a) R(\theta, a) \right]$$

$$= \int_\theta \sum_{a \in A^+} \widehat{\phi}_t^*(\theta, a) \int_\gamma r(\theta, a, \gamma) \widehat{v}_t(\gamma) \widehat{u}_t(\theta) \, d\gamma \, d\theta - \int_\theta \sum_{a \in A^+} \widehat{\phi}_t^*(\theta, a) \int_\gamma r(\theta, a, \gamma) v(\gamma) u(\theta) \, d\gamma \, d\theta$$

$$\leq \sup_\theta |u(\theta) - \widehat{u}_t(\theta)| + \sup_\gamma |(v(\gamma) - \widehat{v}_t(\gamma)|.$$

842 Thus, when $t = \Omega(1)$, by taking $\epsilon = 1/T$ in Proposition A.1 and (27), we arrive that

$$\mathbb{E} \left[ \widehat{J}(\boldsymbol{\rho}_t, \mathcal{H}_t) - \mathbb{E}_\theta \left[ \sum_{a \in A^+} \widehat{\phi}_t^*(\theta, a) R(\theta, a) \right] \right] = O \left( \sqrt{\log T} \left( (t-1)^{\alpha_u - 1} + (t-1)^{\alpha_v - 1} \right) + \frac{1}{T} \right).$$

843 We now focus on $J(\boldsymbol{\rho}_1) - \widehat{J}(\boldsymbol{\rho}_t, \mathcal{H}_t)$. We let

$$\delta_t := \frac{\sup_\theta |u(\theta) - \widehat{u}_t(\theta)| + \sup_\gamma |(v(\gamma) - \widehat{v}_t(\gamma)|}{\min_{\theta \in \Theta, a \in A^+} \{\min\{u(\theta) \boldsymbol{C}(\theta, a) > 0\}\}}.$$

844 We prove that

$$\widehat{u}_t(\theta) \int_\gamma \boldsymbol{c}^i(\theta, a, \gamma) \widehat{v}_t(\gamma) \, d\gamma \leq (1 + \delta_t) u(\theta) \int_\gamma \boldsymbol{c}^i(\theta, a, \gamma) v(\gamma) \, d\gamma$$

845 holds for any $(\theta, a, i)$ tuple, which is obvious if $\boldsymbol{c}^i(\theta, a, \gamma)$ is almost surely zero with respect to $\gamma$.
846 Otherwise, we observe that

$$\widehat{u}_t(\theta) \int_\gamma \boldsymbol{c}^i(\theta, a, \gamma) \widehat{v}_t(\gamma) \, d\gamma - u(\theta) \int_\gamma \boldsymbol{c}^i(\theta, a, \gamma) v(\gamma) \, d\gamma$$

$$= (\widehat{u}_t(\theta) - u(\theta)) \int_\gamma \boldsymbol{c}^i(\theta, a, \gamma) \widehat{v}_t(\gamma) \, d\gamma + u(\theta) \int_\gamma \boldsymbol{c}^i(\theta, a, \gamma) (\widehat{v}_t(\gamma) - v(\gamma)) \, d\gamma$$

$$\leq \sup_\theta |u(\theta) - \widehat{u}_t(\theta)| + \sup_\gamma |(v(\gamma) - \widehat{v}_t(\gamma)|$$

$$\leq \delta_t u(\theta) \int_\gamma \boldsymbol{c}^i(\theta, a, \gamma) v(\gamma) \, d\gamma.$$

847 and thus, with $\eta_t := 1 - \delta_t \leq 1/(1 + \delta_t)$, we derive that

$$u(\theta) \int_\gamma \boldsymbol{c}^i(\theta, a, \gamma) v(\gamma) \, d\gamma \leq \frac{1}{1 + \delta_t} \widehat{u}_t(\theta) \int_\gamma \boldsymbol{c}^i(\theta, a, \gamma) \widehat{v}_t(\gamma) \, d\gamma$$

$$\leq \eta_t \widehat{u}_t(\theta) \int_\gamma \boldsymbol{c}^i(\theta, a, \gamma) \widehat{v}_t(\gamma) \, d\gamma.$$

848 This proves the above inequality. Thus, for an optimal solution $\phi_t^*$ of $J(\boldsymbol{\rho}_t)$, we see that $\eta_t \phi_t^*$ is a
849 feasible solution of the programming $\widehat{J}(\boldsymbol{\rho}_t, \mathcal{H}_t)$. Thus, we notice that

$$\widehat{J}(\boldsymbol{\rho}_t, \mathcal{H}_t) \geq \eta_t \int_\theta \sum_{a \in A^+} \phi_t^*(\theta, a) \int_\gamma r(\theta, a, \gamma) \widehat{v}_t(\gamma) \widehat{u}_t(\theta) \, \mathrm{d}\gamma \, \mathrm{d}\theta$$

$$\geq \eta_t \int_\theta \sum_{a \in A^+} \phi_t^*(\theta, a) \int_\gamma r(\theta, a, \gamma) v(\gamma) u(\theta) \, \mathrm{d}\gamma \, \mathrm{d}\theta$$

$$- \eta_t (\sup_\theta |u(\theta) - \widehat{u}_t(\theta)| + \sup_\gamma |(v(\gamma) - \widehat{v}_t(\gamma)|)$$

$$= \eta_t J(\boldsymbol{\rho}_t) - (\sup_\theta |u(\theta) - \widehat{u}_t(\theta)| + \sup_\gamma |(v(\gamma) - \widehat{v}_t(\gamma)|).$$

850 With respect to (26), we obtain that

$$J(\boldsymbol{\rho}_1) - \widehat{J}(\boldsymbol{\rho}_t, \mathcal{H}_t)$$

$$\leq J(\boldsymbol{\rho}_1) \cdot \left(1 - \eta_t + \frac{\max(\boldsymbol{\rho}_1 - \boldsymbol{\rho}_t, 0)}{\rho^{\min}}\right) + \sup_\theta |u(\theta) - \widehat{u}_t(\theta)| + \sup_\gamma |(v(\gamma) - \widehat{v}_t(\gamma)|$$

$$= J(\boldsymbol{\rho}_1) \cdot \left(\delta_t + \frac{\max(\boldsymbol{\rho}_1 - \boldsymbol{\rho}_t, 0)}{\rho^{\min}}\right) + \sup_\theta |u(\theta) - \widehat{u}_t(\theta)| + \sup_\gamma |(v(\gamma) - \widehat{v}_t(\gamma)|.$$

851 Now, when $t = \Theta(1)$, we have

$$\mathbb{E}\left[\sup_\theta |u(\theta) - \widehat{u}_t(\theta)|\right] = O\left(\sqrt{\log T}(t-1)^{\alpha_u - 1} + \frac{1}{T}\right),$$

$$\mathbb{E}\left[\sup_\gamma |v(\gamma) - \widehat{v}_t(\gamma)|\right] = O\left(\sqrt{\log T}(t-1)^{\alpha_v - 1} + \frac{1}{T}\right).$$

852 By the previous reasoning on $\max(\boldsymbol{\rho}_1 - \boldsymbol{\rho}_t, 0)$, we obtain that when $t = \Omega(1)$,

$$\mathbb{E}\left[\max(\boldsymbol{\rho}_1 - \boldsymbol{\rho}_t, 0)\right]$$

$$\leq \Theta\left(\frac{1}{T-1} + \sqrt{\log T} \sum_{\tau=\Theta(1)}^{t-1} \left(\frac{(\tau-1)^{\alpha_u - 1}}{T - \tau} + \frac{(\tau-1)^{\alpha_v - 1}}{T - \tau}\right) + \sqrt{\frac{\log T}{T - t}}\right).$$

853 Therefore, summing from $t = 1$ to $T_0 \leq T$, we achieve that

$$\mathbb{E}\left[\sum_{t=1}^{T_0} \left(J(\boldsymbol{\rho}_1) - \widehat{J}(\boldsymbol{\rho}_t, \mathcal{H}_t)\right)\right] = O\left((T^{1/2} + T^{\alpha_u} + T^{\alpha_v})\sqrt{\log T}\right).$$

854 Combining with previous bounds on $\mathbb{E}[T - T_0]$ and the estimation errors, we derive the theorem.

### F.3 Proof of Theorem A.2

856 Similar to the proof of Theorem 5.2, we concentrate on re-bounding the three terms under partial
857 information feedback, respectively $\mathbb{E}[T - T_0]$, $\widehat{J}(\boldsymbol{\rho}_t, \mathcal{H}_t) - \mathbb{E}_\theta[\sum_{a \in A^+} \widehat{\phi}_t^*(\theta, a) R(\theta, a)]$, and $J(\boldsymbol{\rho}_1) -$
858 $\widehat{J}(\boldsymbol{\rho}_t, \mathcal{H}_t)$. As for $\mathbb{E}[T - T_0]$, with Lemma 4.1, we argue here that the main term in bounding
859 $\max(\boldsymbol{\rho}_1 - \boldsymbol{\rho}_t, 0)$ when $t = \Theta(T)$ becomes

$$\Theta\left(\sqrt{\log T}\left(\sum_{\tau=\Theta(1)}^{t-1} \frac{(\tau-1)^{\alpha_u - 1}}{T - \tau} + \sum_{\tau=\Theta(1)}^{\Theta(T)} \frac{((\tau-1)/\log T)^{\alpha_v - 1}}{T - \tau} + \sum_{\tau=\Theta(T)}^{t-1} \frac{(\tau-1)^{\alpha_v - 1}}{T - \tau}\right)\right).$$

860 Consequently, when $t$ is close to $T$, we have with failure probability $O(1/T)$,

$$\max(\boldsymbol{\rho}_1 - \boldsymbol{\rho}_t, 0)$$

$$\leq \Theta\left(\frac{1}{T-1} + \sqrt{\log T}\left((T-t)^{\alpha_u - 1} + (T-t)^{-1/2}\right) + (T-t)^{\alpha_v - 1}\log^{3/2 - \alpha_v} T\right).$$

861 This leads to
$$\mathbb{E}[T - T_0] = O\left(\log^{\max(1, 1/(2-2\alpha_u), (3-2\alpha_v)/(2-2\alpha_v))} T\right).$$

862 For the estimation error term $\widehat{J}(\boldsymbol{\rho}_t, \mathcal{H}_t) - \mathbb{E}_\theta[\sum_{a \in A^+} \widehat{\phi}_t^*(\theta, a)R(\theta, a)]$, when $\Omega(1) \leq t \leq \Theta(T)$,
863 the bound now becomes

$$\mathbb{E}\left[\widehat{J}(\boldsymbol{\rho}_t, \mathcal{H}_t) - \mathbb{E}_\theta\left[\sum_{a \in A^+} \widehat{\phi}_t^*(\theta, a)R(\theta, a)\right]\right]$$
$$= O\left(\sqrt{\log T}(t-1)^{\alpha_u - 1} + \log^{3/2 - \alpha_v} T \cdot (t-1)^{\alpha_v - 1} + \frac{1}{T}\right).$$

864 At last, for $J(\boldsymbol{\rho}_1) - \widehat{J}(\boldsymbol{\rho}_t, \mathcal{H}_t)$, we derive that

$$\mathbb{E}\left[\sum_{t=1}^{T_0} \max(\boldsymbol{\rho}_1 - \boldsymbol{\rho}_t, 0)\right] = O\left(\sqrt{\log T}\left(T^{\alpha_u} + T^{1/2}\right) + \log^{3/2 - \alpha_v} T \cdot T^{\alpha_v}\right),$$

$$\mathbb{E}\left[\sum_{t=1}^{T_0} \sup_\theta |u(\theta) - \widehat{u}_t(\theta)|\right] = O\left(\sqrt{\log T} \cdot T^{\alpha_u}\right),$$

$$\mathbb{E}\left[\sum_{t=1}^{T_0} \sup_\gamma |v(\gamma) - \widehat{v}_t(\gamma)|\right] = O\left(\log^{3/2 - \alpha_v} T \cdot T^{\alpha_v}\right).$$

865 Putting together, we obtain that

$$\mathbb{E}\left[\sum_{t=1}^{T_0} \left(J(\boldsymbol{\rho}_1) - \widehat{J}(\boldsymbol{\rho}_t, \mathcal{H}_t)\right)\right] = O\left(\sqrt{\log T}\left(T^{\alpha_u} + T^{1/2}\right) + \log^{3/2 - \alpha_v} T \cdot T^{\alpha_v}\right).$$

866 Synthesizing all the above, we finish the proof of the theorem.