# OpenReview forum: "Contextual Bandits with Knapsacks beyond Worst Cases via Re-Solving"
_NeurIPS.cc/2023/Conference — Submitted to NeurIPS 2023_

### Official Review · Reviewer_VG5f · 2023-07-03

**Soundness:** 3 good
**Presentation:** 3 good
**Contribution:** 2 fair
**Rating:** 4
**Confidence:** 4

**Summary:**

This work studies the CBwK problem beyond the worst case scenario, presenting two results regarding the worst-case locations and log rate respectively. The proposed algorithm utilizes a re-solving heuristic that achieves $O(1)$ under full-information and $O(\log(T))$ regret under partial-information with some regularity conditions. The worst-case guarantee is also presented.

**Strengths:**

This is a well written paper. The key ideas and main contributions are clearly presented. The proofs seems complete and easy to follow.

**Weaknesses:**

This reviewer has doubts on the exact contribution of this paper.
Firstly, the problem studied (CBwK) is well-established in the literature. And the setting that the authors choose might be not so realistic and sort of marginal. For example, one major concern from my understanding is that the budget is "soft", since the budget constraint is only required in expectation, and in summation. However, in reality, "hard" constraints should be considered prevalently, where the constraint should not be compensated, and the expectation might also be unnecessary. The assumption of null action might not be met in reality as well.

Secondly, the algorithm and the re-solving heuristic are from literature. There's something new in the proofs since extending from BwK to CBwK requires a more complicated estimation.

To add to these, there are existing work on constrained reinforcement learning, e.g. [1] that considers similar setting. Since RL is typically considered more general a setting than CB, the results there should also be at least comparable.

Lastly, the assumption for full-information seems overly strong, and the results does not seem very pertinent to the main claims. Maybe it could be better moved to the appendix to make the contribution more clear.

[1] Sobhan Miryoosefi, Chi Jin, A Simple Reward-free Approach to Constrained Reinforcement Learning

**Questions:**

Since this paper is dealing with a subtle topic, a thorough comparison with the existing work should be expected, which seems missing in the current version.

---

> ### Author Rebuttal · Authors · 2023-08-09
>
> We thank the reviewer for the detailed comments. We will answer your doubts in the following. Please also refer to **CQ1 in the global comment**.
>
> **Q1: "Hard" budget and "soft" budget.**
>
> We believe there lies some misunderstanding here. **This paper does consider "hard" budget constraints**, in the sense that our re-solving algorithm never exceeds the initial budget. To see this, one could refer to lines 13-16 in Algorithm 1 on Page 6, which states that the algorithm stops when the budget is near depletion. On the contrary, the "soft" budget constraint in the formulation of $J(\mathbf{\rho}_1)$ and $V^{FL}$ is only used to construct a reward upper bound as given by Proposition 2.1 and Theorem 2.1; and applied in the estimated LP only to instruct the action mode in each round.
>
> **Q2: Assumption of null action.**
>
> In fact, assuming the existence of null (also called "no-op" in some papers) action is a **common practice** in all CBwK papers (e.g., [BLS15], [AD16], [ADL16], [SF22], etc.), which guarantees that there always exists budget-feasible online algorithms. To see this, if there is no null action and the average budget vector is smaller than any possible resource consumption vector, then no feasible online algorithm exists. There are also strong intuitions on the existence of null action in real-life practice. For example, in the scenario of repeated auctions with a total budget constraint, the "null action" is interpreted as bidding 0, ensuring that the agent does not win anything and pay anything. Another example goes to dynamic pricing, where the "null-action" is interpreted as pricing $+\infty$ for the item, and no customer would make the deal in this case.
>
> **Ref:**
> [BLS15] Ashwinkumar Badanidiyuru, John Langford, and Aleksandrs Slivkins. Resourceful contextual bandits.
> [AD16] Shipra Agrawal and Nikhil Devanur. Linear contextual bandits with knapsacks.
> [ADL16] Shipra Agrawal, Nikhil R Devanur, and Lihong Li. An efficient algorithm for contextual bandits with knapsacks, and an extension to concave objectives.
> [SF22] Aleksandrs Slivkins and Dylan Foster. Efficient contextual bandits with knapsacks via regression.
>
>
> **Q3: Full information and partial information.**
>
> The case of full information is also considered in various works relating to online decision-making with resource constraints, e.g., [VBG20], [LG22]. In our work, the full-information setting has two functions: (a) showing that achieving constant regret is possible for the CBwK problem beyond the worst-case in general by applying the re-solving technique, and (b) leading to the introduction of the partial information feedback model (Section 4), which is a major technical novelty of our work. We should mention that **an $O(1)$ regret is never reported in the CBwK literature** to our knowledge, and we believe that this is a good contribution even with full-information feedback (see CQ1). Further, even with partial information feedback, we can achieve an $O(\log T)$ regret, which is comparable to the result of [SS21], but releasing major assumptions, e.g., we are not required only to have two resources.
>
>
> **Ref:**
> [VBG20] Alberto Vera, Siddhartha Banerjee, and Itai Gurvich. Online allocation and pricing: constant regret via bellman inequalities.
> [LG22] Heyuan Liu and Paul Grigas. Online contextual decision-making with a smart predict-then-optimize method.
> [SS21] Karthik Abinav Sankararaman and Aleksandrs Slivkins. Bandits with knapsacks beyond the worst case.
>
>
>
> **Q4: Comparison with works in constrained RL, especially [MJ21].**
>
> Thanks for mentioning. Regarding the comparison with the constrained RL literature, as you mentioned, we have the following differences from [MJ21]:
>
> 1. The main conclusion in [MJ21] considers three problem classes and derives the same sample complexity on these classes. However, we consider an online learning problem with the goal of minimizing regret. The sample complexity bounds in [MJ21] do not clearly guarantee low regret during the learning process. It is possible that during some specific episodes of learning, significant regret may occur due to exploration.
> 2. To add to the above reasoning, some parts of our results are comparable to [MJ21], while some parts are much superior. Generally, in their framework, a sample complexity of $\tilde O(1/\epsilon^2)$ could be accompanied by a total online learning regret of $\tilde \Omega(\sqrt K)$ across $K$ (exploration) episodes. However, our work provides sufficient mild conditions for a total regret of $\tilde O(1)$ beyond the worst-case, as seen in Theorems 3.1 and 4.1. Therefore, directly adopting approaches in [MJ21] in the CB area could result in less persuasive results compared to our paper.
>
> We will add comparisons of our work with related constrained RL literature in the next version.
>
> **Ref:**
> [MJ21] Sobhan Miryoosefi, Chi Jin. A Simple Reward-free Approach to Constrained Reinforcement Learning.

---

> > ### Comment · Reviewer_VG5f · 2023-08-17
> >
> > Thank you for your clarifications on the formulation, assumption, paper structure and related work. I do understand that the authors want to save space for the theoretical development, but I still believe that, in order to give a better positioning of the paper, a more detailed literature review should be expected, (at least at a level of an extended section in the appendix) since this (as well as some closely related) subject has been studied by many, including those mentioned by the reviewers and the AC. Hence I tend to keep my score at this point.

---

> > > ### Author Response · Authors · 2023-08-21
> > >
> > > Thank you for your kind suggestions. We will add a discussion in an appendix to extend the literature review part with a detailed comparison of these works, recommended by both reviewers and ACs, with this submission.

---

### Official Review · Reviewer_CJMH · 2023-07-05

**Soundness:** 2 fair
**Presentation:** 3 good
**Contribution:** 3 good
**Rating:** 6
**Confidence:** 4

**Summary:**

This works considers a general setup of Contextual Bandits with Knapsacks (CBwK). The authors identify sufficient conditions under which constant of logarithmic regrets are possible (while previous authors were mainly dealing with $\sqrt{T}$ worst-case regrets). The studied algorithm is rather intuitive and is based on a sequential refinement of the underlying best static LP problem.



**Strengths:**

1. Identification of the conditions for $o(\sqrt{T})$ regret is a great contribution.
2. The algorithm does not need to know whether the conditions are satisfied---it is adaptive.
3. The main body exposition is rather clear, but more details would be appreciated.
4. The approach is intuitive and computationally tractable.
5. Overall I liked the proofs, they are a bit hard to read due to heavy notation, but once the reader is comfortable, the presentation is rather good.

**Weaknesses:**

Here are the weaknesses in no particular order

1. Bounds are in expectation (which is fine per-se, but from my experience the results are often formulated with high probability)
2. Little to no comments on the actual stopping time of the algorithm, is it much smaller than T? (I only found this information in the appendix)
3. O(1) is great, but what kind of 1 it involves is not present in the main body.
4. Regret is not the only part of the story: e.g., Agrawal and Devour 2016 allow o(1) budgets, instead of constant budgets. In this case comparing to their result is not really fair.
5. While computationally tractable, the algorithm is not very efficient, one needs to solve a potentially huge LP problem at each step.




**Questions:**

1. Proofs of all the results rely heavily on the stability Lemma C.2. This lemma involves a constant $D$, which eventually appears in the upper bounds. The actual dependency on the parameters of the problem is not discussed. Imagine I am in the setup of non-constant budget e.g. 1 / \sqrt{T} budget. Shall $D$ start to depend on $T$ as well in this case? In general, expliciting exact dependency of $D$ on the problem's parameters would be great. Otherwise $O(1)$ is a bit too opaque.

2. Say I want to derive bounds with high probability, then, I feel that the same results are possible, but they will involve additional condition like $T \geq T_0$ with $T_0$ being non-explicit and potentially very large. Is it the case?

3. Could the authors provide a comment on potential extension of the results to algorithms based on OCO?

4. The extension to the continuous case is slightly unsatisfactory. Personally, I expected a more different type of modelling condition e.g. logistic model (instead of smoothness type assumption). Would it be possible to extend the analysis to continuous case, maintaining the $\sqrt{T}$ worst-case regret, assuming that both reward and costs can be modelled by a logistic model?

P.S. I am open to increase my score.

**Limitations:**

I was not aware of the work of Chen et al 2022, but upon skimming it looks very similar to the current submission (tools and techniques-wise). A broader discussion would be appreciated.

While being honest and correct, the extension to the continuous case, is not very satisfactory (but maybe it is a matter of taste).

---

> ### Author Rebuttal · Authors · 2023-08-09
>
> We thank the reviewer for the detailed and constructive comments. We will answer your questions according to the logical order. Please also refer to **CQ1-3 in the global comment**.
>
>
> **Q1: Stopping time of the algorithm.**
>
> Under Assumption 3.1 and full information feedback, $\mathbb{E}[T - T_0]$ is constant as given by Lemma C.4. For a high probability result, by lines 617, 620, and 639, we can derive that for any $t \leq T_- = T - O(1)$, $\Pr[T_0 > t] = \exp(-\Omega(T - t)) + O(T)\exp(-\Omega(\sqrt{T}))$, which decreases exponentially with $t$. With partial information feedback, $\mathbb{E}[T - T_0]$ is still a constant, while the high probability counterpart becomes $\exp(-\Omega(T - t)) + O(T)\exp(-\Omega(\sqrt{T}/\log T))$. Without Assumption 3.1, with full information, line 736 points out that $\mathbb{E}[T - T_0] = O(\log T)$, and $\Pr[T_0 < T - O(\log T)] = O(1 / T)$; the same results also hold with partial information, by line 775. In the continuous setting, lines 838 and 861 respectively bound the expected stopping time in two information feedbacks as well, and corresponding high probability results ($\Pr[t_0 < T - O(\cdots)] = O(1/T)$) also hold. All the above results show that in all cases, the stopping time $T_0$ is sufficiently close to $T$, either in a constant scale (with Assumption 3.1) or in a polylog scale (without Assumption 3.1). We will note this part in the main body in the next version.
>
>
> **Q2: Bounds with high probability.**
>
> Despite that there are high probability results for the stopping time as above, a similar satisfying result for regret (e.g., $O(1)$ regret with probability $1 - O(1/T)$) may not exist. In fact, we believe that the **best possible high probability result** for CBwK problems is $\Pr[reg \geq \tilde{O}(\sqrt{T})] \leq O(1/T)$, which is already given in previous works (e.g., [AD16], [ADL16], [SF22]). The intuition here is that the randomness of the external factors could deviate the total reward from its expectation, and concentration inequalities could best guarantee the above result. Therefore, such a negative result could be unavoidable, and this argument could best lead to an expected regret of $\tilde{O}(\sqrt{T})$, which is far from what we achieve. In fact, eq.(b) and eq.(e) in the Lagrangian regret decomposition of Page 18 are crucial to achieving our constant regret results. There, we use the Optimal Stopping Theorem to equalize different expectations in a martingale. If we otherwise adopt the Azuma--Hoeffding inequality here to bound the difference, an $\tilde{O}(\sqrt{T})$ regret would inevitably occur, ruining the whole analysis.
>
>
> **Q3: Contents of the $O(1)$ regret, $D$, and $o(1)$ average budget.**
>
> The regret of the re-solving algorithm is irrelevant to the total timespan $T$, and it is related to the following key parameters (see Section 2):
>
> 1. increasing with $n$, $|\Theta|$, and $|\Gamma|$;
> 2. decreasing with $\rho^\min$ and $D$.
>
> Here, the stability factor $D$ reflects how far the fluid LP $J(\rho_1)$ is away from any LP with a degenerate optimal solution or multiple optimal solutions in $L_\infty$ norm. This can be found in Lemma C.3 and its proof. We will add that in the main body for the next version of our paper. Thus, when $\rho^\min$ is $o(1)$, $D$ would be $o(1)$ as well, since 0 budget vector always makes the LP degenerate. However, settling the exact relationship between $D$ and $\rho^\min$ when the latter is $o(1)$ is hard since the coordination of different parameters is complicated. Meanwhile, the key Assumption 3.1 would be vague in this case. Due to the above reasons, our analysis may not work well when $\mathbf{\rho}$ and $D$ are $o(1)$. In fact, it is widely supposed in the problem of network revenue management (NRM), which is a simplification of CBwK without external factors, that the average budget vector is constant, e.g., [BW20].
>
>
> **Q4: $\tilde{O}(\sqrt{T})$ regret for logistic reward and consumption models.**
>
> Unluckily, the answer could be negative. The reason here is that a major technical assumption in all CBwK works is that the reward and consumption are **non-negative and bounded** (e.g., [BLS15], [AD16], [ADL16], [SF22], etc.). Otherwise, estimating their means, which is a key step for (almost) all existing CBwK algorithms, could be impossible. This is the same for our algorithm as well. However, the answer could be positive if we change the reward/consumption model to some other **non-negative, bounded**, and parameterized distributions. To see this, we can suppose $r(\theta, a, \mathbf{\gamma}) = \gamma_{\theta, a, 0}$ and $\mathbf{c}^i(\theta, a, \mathbf{\gamma}) = \gamma_{\theta, a, i}$, where all $\gamma$'s follow the corresponding parameterized distributions. To estimate the distribution, we can alternatively use classic parameterized estimation methods like maximum likelihood estimation (MLE). When the distribution satisfies certain assumptions, according to [NM94], MLE guarantees an $O(1/\sqrt{n})$ convergence rate on the density function with $n$ independent samples. Plugging that into our analysis also implies an $\tilde{O}(\sqrt{T})$ regret.
>
>
> **Complete Ref:**
> [AD16] S. Agrawal and N. Devanur. Linear contextual bandits with knapsacks.
> [ADL16] S. Agrawal, N. R Devanur, and L. Li. An efficient algorithm for contextual bandits with knapsacks, and an extension to concave objectives.
> [SF22] A. Slivkins and D. Foster. Efficient contextual bandits with knapsacks via regression.
> [CLY22] G. Chen, X. Li, and Y. Ye. An improved analysis of lp-based control for revenue management.
> [BW20] P. Bumpensanti and H. Wang. A re-solving heuristic with uniformly bounded loss for network revenue management.
> [BLS15] A. Badanidiyuru, J. Langford, and A. Slivkins. Resourceful contextual bandits.
> [NM94] W. K Newey and D. McFadden. Large sample estimation and hypothesis testing.

---

### Official Review · Reviewer_uK79 · 2023-07-06

**Soundness:** 2 fair
**Presentation:** 3 good
**Contribution:** 2 fair
**Rating:** 6
**Confidence:** 2

**Summary:**

The paper studies stochastic contextual Bandits with Knapsacks. The authors provide an algorithm that guarantees regret smaller than the worst-case $\tilde O(\sqrt{T})$ in non-degenerate instances. In particular, they provides an algorithm that achieves $\tilde O(1)$ regret when the fluid LP has a unique and non-degenerate solution. Moreover, in the worst-case the algorithm maintains $\tilde O(\sqrt{T})$ regret.

**Strengths:**

The main paper is clear and well written. The problem under study is interesting. The technical results are not straightforward.

**Weaknesses:**

Most of the results in the paper are based on the existence of an unique and non-degenerate solution (Assumption 3.1). The formal definition of this assumption is missing. Moreover, while this assumption is commonplace in the linear programming literature, it is not clear whether it makes sanse in the bandit with knapsack problem. Finally, previous work suggests that large regret is unavailable in many standard cases.

The main paper does not give a clear intuition of the techniques used to prove the theoretical results and the appendix is difficult to follows. Moreover, the appendix is heavily based on results in other papers, e.g., Chen et al. [2022]. This makes the proofs hard to follow. Moreover, it is not clear which are the main technical contributions of the paper, and to what extend the results follow directly from the application of known techniques.

**Questions:**

Why is Assumption 3.1 reasonable in the bandit with knapsack problem?

**Limitations:**

Yes

---

> ### Author Rebuttal · Authors · 2023-08-09
>
> We thank the reviewer for the meaningful comments. The following are our responses to your questions. Please also refer to **CQ1 in the global comment**.
>
> **Q1: The definition of Assumption 3.1.**
>
> Assumption 3.1 supposes that the optimal solution of $J(\mathbf{\rho}_1)$ is unique and non-degenerate. We believe that uniqueness is easy to understand. Meanwhile, non-degeneracy is defined as the number of non-basic variables (Definition C.1), the number of binding resource constraints (Definition C.2), and the number of binding allocation constraints (Definition C.2) sum to $|\Theta| \times m$. In other words, the total number of binding constraints of the optimum equals the dimension of the LP. We will clarify the meaning of this critical assumption in the next version.
>
> **Q2: Why is Assumption 3.1 important for CBwK?**
>
> The importance of Assumption 3.1 for CBwK partly comes from literature in network revenue management (NRM), which is a simplification of CBwK without external factors. As revealed by [JK12], [CLY22], and [BBP23], uniqueness and non-degeneracy of the fluid LP act as important sufficient conditions for an $O(1)$ regret to hold for re-solving-based methods in this problem.
>
> To add on the above, our Theorem 2.1 shows that uniqueness and degeneracy of $J(\mathbf{\rho}_1)$ is sufficient for the $\Omega(\sqrt{T})$ regret lower-bound to hold, indicating that the structure of $J(\mathbf{\rho}_1)$ could be much crucial for the regret result of CBwK problems, and Assumption 3.1 arises as an close-to-necessary condition in this sense.
>
> **Ref:**
> [JK12] Stefanus Jasin and Sunil Kumar. A re-solving heuristic with bounded revenue loss for network revenue management with customer choice.
> [CLY22] Guanting Chen, Xiaocheng Li, and Yinyu Ye. An improved analysis of lp-based control for revenue management.
> [BBP23] Santiago Balseiro, Omar Besbes, and Dana Pizarro. Survey of dynamic resource constrained reward collection problems: Unified model and analysis.
>
> **Q3: Previous work suggests that large regret is unavoidable (is that the correct understanding?) in many standard cases.**
>
> Indeed, many prior works have suggested that an $\Omega(\sqrt{T})$ regret is inevitable, and our Theorem 2.1, which gives an almost tight sufficient condition for this worst-case lower bound, also clarifies this fact. However, this paper stands on a beyond-the-worst-case perspective like [SS21]. The key aim of this paper is to point out that for most CBwK instances, an $O(1)$ regret is achievable, which is a much positive result for CBwK problems and improves over previous UCB-like results (e.g., [SF22], [HZWXZ22]), which only guarantees a universal $O(\sqrt{T})$ regret for CBwK.
>
> **Ref:**
> [SS21] Karthik Abinav Sankararaman and Aleksandrs Slivkins. Bandits with knapsacks beyond the worst case.
> [SF22] Aleksandrs Slivkins and Dylan Foster. Efficient contextual bandits with knapsacks via regression.
> [HZWXZ22] Yuxuan Han, Jialin Zeng, Yang Wang, Yang Xiang, and Jiheng Zhang. Optimal contextual bandits with knapsacks under realizability via regression oracles.
>
> **Q4: Intuition of techniques.**
>
> The intuition of our algorithm comes from the basic fact that when the estimates of expected rewards and consumption vectors are sufficiently accurate, $\hat{J}(\mathbf{\rho}_t, \mathcal{H}_t)$ should be close to $J(\mathbf{\rho}_t)$. Thus the action mode of the re-solving algorithm in round $t$ should also be close to leading to a reward upper bound $(T - t) J(\mathbf{\rho}_t)$ for the remaining rounds. As for the detailed proof, we build our analysis on the framework of [CLY22], which involves a Lagrangian regret decomposition and analyzes the terms one by one. The major reason why we do not put the technique intuitions in the main body is the space limit, and we will try to add more of this in the next version.
>
> **Ref:**
> [CLY22] Guanting Chen, Xiaocheng Li, and Yinyu Ye. An improved analysis of lp-based control for revenue management.
>
>
> **Q5: The readability of the appendix.**
>
> We have tried our best to make the appendix simple to understand. Yet, since our proof is complicated, we have to involve some heavy notations and multiple layers of logical nesting. We here further state the organization of the appendix, which may benefit the understanding.
>
> In Appendix A, we state our regret results with continuous randomness. In Appendix B, we prove the regret worst-case result, Theorem 2.1. Appendix C.1 devotes to proving the main result, Theorem 3.1, where C.1.1 presents the Lagrangian regret decomposition, and C.1.2 and C.1.3 analyze different terms in the decomposition. Appendix C.2 - C.6 complement missing proofs of lemmas arising in Appendix C.1. Appendix D focuses on proving the regret results in the partial information setting, where Appendix D.1 derives Theorem 4.1, and Appendix D.2 proves the crucial Lemma 4.1 for the partial feedback model. Appendix D.3 and D.4 prove lemmas arising in previous parts of Appendix D. Appendix E deals with Theorem 5.1 and 5.2, which show that the re-solving method is near-optimal even in worst cases. At last, Appendix F complements the missing parts of Appendix A with continuous randomness.
>
> We will work hard to further improve the writing of the appendix in the next version.
>
> As for the reliance on previous works, we will include essential results as propositions in the appendix in the next version, e.g., the stability result of [CLY22].
>
> **Ref:**
> [CLY22] Guanting Chen, Xiaocheng Li, and Yinyu Ye. An improved analysis of lp-based control for revenue management.

---

> > ### Comment · Reviewer_uK79 · 2023-08-21
> >
> > Thanks for the detailed response which answers most of my questions. However, I still believe the work shares many similarities with Chen et al. [2022]. I've updated my score accordingly.

---

> > > ### Author Response · Authors · 2023-08-21
> > >
> > > Thanks a lot!

---

### Official Review · Reviewer_Wp2p · 2023-07-07

**Soundness:** 3 good
**Presentation:** 2 fair
**Contribution:** 3 good
**Rating:** 4
**Confidence:** 4

**Summary:**

This paper considers the problem of contextual bandits with knapsacks and provides an algorithm that goes beyond worst-case (i.e., provide logarithmic regret) under the mild condition of unique optimal solution and non-degenerate solution. This also simultaneously enjoys an optimal worst-case regret bound when the conditions aren't met. The paper also shows a Omega(sqrt(T)) lower-bound when the instance has a unique optimal solution and a degenerate solution.

**Strengths:**

+ The paper considers a significant generalization of the results in Sankararaman and Slivkins 2021, in that it extends the logarithmic regret from two resources and best-arm optimality to arbitrary resources and unique best solution.

+ The paper also proves that the new algorithm derived from resolving the program enjoys worst-case regret bounds that are optimal (in the interesting regime of B ~ O(T)).

+ The results also applies to full policy-based contextual bandit problems, as opposed to prior work with logarithmic regret bounds which only applies to the linear contextual bandits setting.

**Weaknesses:**

- The algorithm needs to (a) solve a program at each round (can this be removed?) unlike UCB type algorithms where the program needs to be solved only when the ucb of any single quantity changes by a constant factor and (b) even in a single round the run-time of solving the program is unclear (I might be wrong, please correct me). Although the proposed results are interesting, this seems like a major downside from the algorithm front.

- [Some rewording of comparison with prior work] The proposed algorithm is very similar to that of [Flajolet and Jaillet, 2015] (the paper mentions this, although the wording could be more generous to the prior work). Likewise, the paper mentions that the setting considered in Sankararaman and Slivkins '2021 is "and
almost surely excludes all problem instances" which is not technically true. Note that the prior work paper uses the same argument of perturbing LPs in the case of d=2. Thus, the key difference is extending to d > 2 and not about the number of instances.

- Although the lower-bound presented in this paper is interesting, I think some of the lower bounds are implied from the instances presented in Sankararaman and Slivkins '2021.

---

EDIT:

I still don't know how to implment this algorithm apart from the trivial way of having one arm per (context, arm) pair. This is important, since then the algorithm is trivial, and the results then likely follow from prior works, in particular [1] which implements this for the k-armed bandit setting. Thus, the paper needs to explain clearly why this is better than that trivial reduction, and as a result, the bounds are significantly better. I believe multiple reviewers get at this same point, and i am not able to see a  convincing argument. Please make sure to incorporate this detailed comparision in the next version of the paper.


[1] Logarithmic regret bounds for Bandits with Knapsacks - Arthur Flajolet, Patrick Jaillet

**Questions:**

can you elaborate a bit more on the exact running time of the algorithm, per step? Also can similar tricks of not having to solve in each step, as in UCB, be used for the resolving heuristic?

**Limitations:**

This is a mathematical paper and no societal impact.

---

> ### Author Rebuttal · Authors · 2023-08-09
>
> We thank the reviewer for the constructive comments. The following are our responses to your questions. Please also refer to **CQ1-3 in the global comment**.
>
> **Q1: Rewording of comparison with prior work.**
>
> Thanks for the kind notice. We will adopt your suggestions, correct inaccurate expressions, and be more generous with prior works in the next version of this paper. Nevertheless, we should also mention that [FJ15] works with the bandits with knapsacks (BwK) setting, while we extend to the CBwK setting. Meanwhile, [SS21] considers the linear variant for the CBwK problem, while our solution is wider than this setting.
>
> **Ref:**
> [FJ15] Arthur Flajolet and Patrick Jaillet. Logarithmic regret bounds for bandits with knapsacks.
> [SS21] Karthik Abinav Sankararaman and Aleksandrs Slivkins. Bandits with knapsacks beyond the worst case.
>
> **Q2: Lower-bound results.**
>
> Indeed, [SS21] presents $\Omega(\sqrt{T})$ regret lower-bound results under some cases. However, their results are less persuasive than ours in that their given instances have three arms and Bernoulli rewards. Compared to them, we show that for all instances such that the fluid LP has a unique and degenerate solution, an $\Omega(\sqrt{T})$ regret lower-bound is inevitable. In this sense, our lower-bound condition is less restrictive. Further, the lower-bound proof of [SS21] is different from ours. Their key argument is that any algorithm incurs at least an $\Omega(\sqrt{T})$ regret on one of the two closely-perturbed" instances, while our proof focuses more on the structure of the fluid LP.
>
> **Ref:**
> [SS21] Karthik Abinav Sankararaman and Aleksandrs Slivkins. Bandits with knapsacks beyond the worst case.

---

### Author Rebuttal · Authors · 2023-08-09

Here, we will answer some common questions raised by the reviewers.

**CQ1: Technical novelty.**

Our work has the following technical contributions:

1. In proving the worst-case regret (Theorem 2.1), we consider a hybrid benchmark to benefit the analysis, which first appears in literature to our knowledge. We look forward that the hybrid benchmark could be of independent interest for future studies.
2. We extend the re-solving method from bandits with knapsacks (BwK) (e.g., [FJ15], [SS21]) and the framework of [CLY22] in network revenue management (NRM) to the problem of CBwK. For the former, the optimal action mode is not universal anymore and correlated with the context. Meanwhile, the latter involves two major non-trivial differences: (a) the number of actions increases from 2 to any finite number, and (b) the learning on the distribution of external factor is necessary. (a) forces us to analyze more regret terms than the analysis of [CLY22], specifically, the second term in eq.(5) on the allocation complementarity. (b) further complicates the analysis by plugging the estimation error of external factor distribution of each round into the total regret. This part of improvement leads to results in Sections 3 - 5.
3. Our analysis of the partial information feedback (Section 4) is novel. With partial information, the agent can only see a sample of the external factor when a non-null action is chosen. To guarantee high performance, our key argument here is that the frequency of accessing external factors is almost constant with high probability, given by Lemma 4.1. We adopt the property of the re-solving algorithm to show that the null action should not be taken too much, thus proving this lemma.
4. We also consider continuous randomness for the re-solving method and CBwK problem and incorporate non-parameterized distribution estimation techniques to derive results in Appendix A. This is a new combination as far as we know.

**Ref:**
[FJ15] Arthur Flajolet and Patrick Jaillet. Logarithmic regret bounds for bandits with knapsacks.
[SS21] Karthik Abinav Sankararaman and Aleksandrs Slivkins. Bandits with knapsacks beyond the worst case.
[CLY22] Guanting Chen, Xiaocheng Li, and Yinyu Ye. An improved analysis of lp-based control for revenue management.

**CQ2: Running time of the algorithm.**

In our algorithm, with discrete randomness, the running time of each round is irrelevant with $T$. The reason is that the size of the programming $\hat{J}(\mathbf{\rho}_t, \mathcal{H}_t)$ is only correlated with the number of resources, the size of the context space and action space, and the time of computing the estimated values $\hat{R}_t(\theta, a)$ and $\hat{\mathbf{C}}_t(\theta, a)$ is only correlated with the size of the external factor space. Thus, when solving the linear programming in the discrete case in each round with the interior point/ellipsoid method, the running time of our algorithm in each round is only a polynomial of the number of resources, the sizes of the context, action, and external factor spaces, and does not rely on $T$. Even in the continuous case, the time of solving the programming and the running time of each round are still irrelevant with $T$.

**CQ3: Potential algorithms with UCB and OCO.**

To combine UCB/OCO-based and re-solving-based algorithms, one could consider doing the re-solving in only $o(T)$ rounds and use UCB/OCO methods to make slight adjustments between two consecutive re-solving rounds in order to refine the action mode and compensate for the perturbation of the external factor on the consumption. A piece of solid evidence showing that such a combination could work is given by [BW20], in which they show that merely $O(\log \log T)$ times of re-solving plus a UCB-like thresholding technique could guarantee an $O(1)$ regret for the problem of quantity-based network revenue management (NRM), which can be seen as a simplification of CBwK with only accept/reject action choices and without external factors. However, extending the above algorithm to CBwK is uneasy due to the randomness of the consumption of resources. Therefore, introducing UCB/OCO methods between two re-solving rounds may help reduce the uncertainty there. If such a trial succeeds, the resulting algorithm would simultaneously own high efficiency and constant regret under mild assumptions.

**Ref:**
[BW20] Pornpawee Bumpensanti and He Wang. A re-solving heuristic with uniformly bounded loss for network revenue management.

---

### Author Response · Authors · 2023-08-15

We, the authors, wish to know how the reviewers regard our rebuttal and whether they have further questions and comments on our response.

---

> ### Comment · Area_Chair_Mcfo · 2023-08-17
>
> Thanks for your message, let me ping the reviewers. In the meantime, I've skimmed your paper and I have some more questions (in a separate message).

---

### Comment · Area_Chair_Mcfo · 2023-08-17
**questions for the authors**

Dear authors:

I have some high-level questions after skimming the paper.

How do you handle contexts? To simplify: how do you handle contexts with bandit feedback _in the absence of resource constraints_? I don't see any of the standard approaches in your setup, such as classification oracles, regression oracles, or linearity. Seems you are comparing to the best policy. So, in the worst case, would you get a regret bound that scales as  $\sqrt{\text{NumContexts}\cdot \text{NumArms}}$?

What would be the "constant" in front of your $\log(T)$ regret bound for bandit feedback? Call it $C$. The essence of this question is, what are the specific relevant parameters that $C$ is inversely dependent on? Presumably this is all "encoded" in your Assumption 3.1, but I think this should be spelled out. Let me simplify this question into two: without resource constraints (resp., without contexts), and how does your answer compare to prior work on $O(\log T)$ regret bounds for CB (resp., BwK). Without constants and without resources, do you get the standard $C = O(1/\text{gap})$ dependence? Do you have same/similar parameters in your $O(1)$ regret bound for full feedback?

Your Algorithm 1 needs to be initialized with parameter $\rho$. How do you set it? In particular, do you need to know some of the "hidden" problem parameters?

How do you get around $\Omega(\sqrt{T})$ lower bounds from [SS21]? Presumably, this is also encoded in Assumption 3.1 somehow.

Your benchmark is stronger compared to [SS21], right? (Linear relaxation vs. the total expected reward of the best distribution over arms.) So, this makes your upper-bound results even stronger, and your lower-bound results weaker. This is worth pointing out (along with some deeper implications, if any).

### Re prior work on CBwK (Sec1.2)

The first paper on CBwK is Badanidiyuru et al (2014).

The distinctions that you bring up mirror the distinctions in prior work on CB (i.e., without constraints): namely, CB with classification oracles vs. CB with regression oracles vs. CB with linearity.

A few other papers achieve $O(\log T)$ regret in BwK (under different assumptions and parameters):

- Huasen Wu, R. Srikant, Xin Liu, and Chong Jiang. Algorithms with logarithmic or sublinear regret for constrained contextual bandits. NIPS 2015.

- Alberto Vera, Siddhartha Banerjee, and Itai Gurvich. Online allocation and pricing: Constant regret via bellman inequalities. Operations Research, 2020.

- Xiaocheng Li, Chunlin Sun, and Yinyu Ye. The symmetry between arms and knapsacks: A primal-dual approach for bandits with knapsacks. ICML 2021.

Also, several papers achieve $O(\log T)$ regret in BwK problems with only one constraint, including time. Starting with: Andras Gyorgy, Levente Kocsis, Ivett Szab´o, and Csaba Szepesv´ari. Continuous time associative bandit problems. IJCAI 2007.

---

> ### Author Response · Authors · 2023-08-18
> **Response to AC's questions (part 1)**
>
> Dear AC Mcfo:
>
> Thank you for your kind remarks and questions, and we would now like to answer your questions in order. Please feel free to discuss with us any further questions.
>
> ## Q1: Bandit information feedback.
>
> To start with, we should clarify that this work does not consider the bandit information feedback model, and it is an important question how our framework could tackle this feedback model. Instead, we deal with full information feedback and partial information feedback. With full information feedback, the external factor $\gamma_t$ is revealed after each round $t$'s decision/action/arm is selected; while with partial information feedback, the external factor $\gamma_t$ is revealed only when the decision of the round $a_t$ is non-null. Both these two feedback models are stronger (for the agent) than the bandit feedback in the sense that with a sample of external factor $\gamma_t$ revealed, the reward and consumption of all contexts and arms in this round can be deduced; however, with bandit feedback, only the reward and consumption of the current $(\text{context, arm})$ pair is known. We now clarify how we deal with full/partial information feedback.
>
> In fact, we use standard distribution estimation techniques (empirical probability estimating actual probability) to deal with full/partial information feedback by estimating the probability distribution of the external factor. This technique can be seen as a counterpart of (linear) classification/regression oracles with bandit feedback. The **trade-off** here is that for the more challenging bandit feedback, corresponding oracles would suppose specific structures on the randomness, e.g., the expectation of reward/consumption is linear in the $(\text{context, arm})$ pair. In contrast, with full/partial information feedback, we do not require any distribution properties of the external factor, therefore bypassing realizability issues [HZWXZ22]. Further, when dealing with partial information feedback, a **technical challenge** is giving a lower bound on the samples of the external factor accessed by the algorithm. On this side, we prove that the frequency of the algorithm choosing non-null actions is almost constant. Here, we utilize the property of the re-solving method to give this result (Lemma 4.1), which is a technical novelty.
>
> **Ref:**
> [HZWXZ22] Yuxuan Han, Jialin Zeng, Yang Wang, Yang Xiang, and Jiheng Zhang. Optimal contextual bandits with knapsacks under realizability via regression oracles.

---

> ### Author Response · Authors · 2023-08-18
> **Response to AC's questions (part 2)**
>
> ## Q2: Constants in the regret bound.
>
> As we have just mentioned, with the difference in the feedback model, the constants underlying the big-O notation may be different/unsimilar from the ones in previous works. We now discuss these constants in detail. We start with the easier worst-case bounds. In this scenario, the reliance of the bound on key parameters is the following:
>
> - linear with $J(\mathbf{\rho}_1)$;
> - linear with $\text{NumContexts}$ and $\text{NumResources}$;
> - not explicitly related with $\text{NumArms}$.
>
> To understand the above results, we should further note that $J(\mathbf{\rho}_1)$ is implicitly correlated with $\text{NumContexts}$, $\text{NumResources}$, and $\text{NumArms}$. Nevertheless, these relations can be complicated.
>
> For our $\tilde{O}(1)$ results beyond the worst-case, the exact relation between regret bounds and key parameters is much complicated, so we omit these constants in our main statements. The following relations between the regret bound and key parameters hold for both full/partial information feedback results.
>
> - linear with $\text{NumContexts}$;
> - quadratic with $\text{NumResources}$;
> - not explicitly related with $\text{NumArms}$;
> - quadratic with $1/D$, where $D$ is the stability factor that represents the minimum $L_\infty$ distance of the LP $J(\mathbf{\rho}_1)$ to any LP with multiple optimal solutions or a degenerate optimal solution. Here, the distance between two LPs is deemed the maximum difference between the two corresponding coefficients.
>
> We now compare our results with CB/BwK/bandit results as follows:
>
> - **[On $\text{numArms}$.]** Previous regret results on CB and BwK would always scale with $\sqrt{\text{numArms}}$ or $\text{numArms}$, while our regret does not involve $\text{numArms}$ explicitly.
> - **[On $\text{numResources}$.]** For the BwK problem, the well-known $O(\log T)$ result given by [SS21] supposes that the number of resources is fixed to 2, and it is still unclear whether their analysis can be extended to an arbitrary number of resources.
> - **[On $\text{numContexts}$.]** $\text{NumContexts}$ do not always appear in previous works. However, this is inevitable in our bound, which is led by the estimation error of the context distribution.
> - **[On the stability factor $D$.]** Mapped to the bandit model, our constant scales with $O(1/D^2) = O(1/\text{gap}^2)$. To see this, note that when $\text{gap}$ is 0, the fluid LP has multiple solutions, thus $\text{gap} \leq D$. This result is similar to the first part of [SS21] ($O(1/G_{LAG}^2)$ there), and is the same with [CLY22]. We still do not know whether we can improve the constant here to $O(1/D) = O(1/\text{gap})$, and this is an important future direction.
>
> **Ref:**
> [CLY22] Guanting Chen, Xiaocheng Li, and Yinyu Ye. An improved analysis of lp-based control for revenue management.
>
>
> ## Q3: Parameter $\mathbf{\rho}$.
>
> Actually, parameter $\mathbf{\rho} = \mathbf{B}/T$ here represents the average budget vector, where $\mathbf{B}$ is the initial budget vector and $T$ is the number of rounds. (See Section 2.) These two variables are necessary inputs for the agent in the CBwK problem. Our algorithm **does not need** any "hidden" parameters specific to the problem instance (e.g., we do not need to know the stability factor $D$ in our algorithm). On this side, some previous regression-oracle-based results would set the "learning rate" according to the performance of the oracle, which is a "hidden" parameter.
>
> ## Q4: Dealing with $\Omega(T)$ lower bounds.
>
> In fact, we show in Theorem 2.1 that when the fluid LP has a unique and **degenerate** optimal solution, then an $\Omega(\sqrt{T})$ is inevitable. In contrast, we suppose in Assumption 3.1 that the fluid LP has a unique and **non-degenerate** optimal solution, which is almost a complement for the above condition. Here, the uniqueness and non-degeneracy assumption of the LP is similar/equivalent to the best-arm-optimality assumption in [SS21] (since they consider the BwK problem without context and only deal with two resources). Actually, their lower bound is also achieved when the best-arm-optimality does not hold. In this sense, our results match the ones in [SS21].
>
> ## Q5: Benchmark comparison with [SS21].
>
> In fact, our benchmark is the same as the one of [SS21] when neglecting the impact of context since in both benchmarks, we relax the reward/consumption by their means, and the budget constraint is only needed to be satisfied in expectation. Their benchmark implies the best distribution of arms as ours, and our benchmark is also a **linear programming** problem. Thus, these two benchmarks are identical if we add the context information to [SS21]'s benchmark. Actually, the choice of our benchmark is common in the CBwK literature.
>
> ## Q6: Re Prior works.
>
> We sincerely thank the AC for mentioning more prior work on this subject. We will take your suggestions and improve our related work part in the next version of this paper.

---

> > ### Comment · Area_Chair_Mcfo · 2023-08-18
> >
> > **Re partial feedback:** Thanks for the clarification! While your model of "partial feedback" is clearly specified in Preliminaries, I missed it in my superficial reading, and assumed "bandit feedback" instead. I'm afraid the reviewers might have missed it, too! One reason for this confusion is that you often refer to "bandits", and the term "partial feedback" is often used in the literature to refer to "bandit feedback".
> >
> > I find it very misleading that you talk about CBwK in your title, abstract, and Introduction _as if_ you consider bandit feedback, even though your feedback model is much more permissive. In particular, it is precisely "full feedback" when there is no "null arm", and instead the algorithm can skip rounds and/or stop early. This needs to be clarified throughout. In particular, it should not be surprising when/if your positive guarantees improve over those in prior work on CBwK.
> >
> > What is the motivation for BwK with your version of partial feedback (or full feedback)? (As opposed to an even more permissive variant when the "full feedback" is revealed _before_ the corresponding round.)
> > \
> > \
> > **Re parameters:** I was mainly asking about gap-like parameters. Some extensions of the usual notion of "gap" from stochastic bandits are usually needed for $\log(T)$-regret results in BwK and CB. Presumably, yours are "encapsulated" in $J(\mathbf{\rho}_1)$. It is unfortunate that you don't spell out the gap-like parameters more explicitly and do not separate out the dependence on the "public" parameters (NumArms, NumContexts, and NumResources). Can you work this out for the special case of no resources? Worrisome if not.
> >
> > How do your _worst-case_ bounds scale with the public parameters?
> >
> > For the full-feedback case, do your regret bounds depend on $J(\mathbf{\rho}_1)$, too, or only through $D$?
> > \
> > \
> > **Re parameters in Algorithm 1:** got it, my bad!
> > \
> > \
> > **Re benchmarks:** for a given distribution over arms, its expected reward _when deployed as an algorithm_ is generally smaller than its reward in the fluid approximation. The benchmark in [SS21] considers the former, I think.

---

> > > ### Author Response · Authors · 2023-08-19
> > > **Response to AC's further questions (part 1)**
> > >
> > > Dear AC Mcfo:
> > >
> > > Thanks again for your kind reply! We will now respond to your further considerations. Again, feel free to discuss with us any additional worries.
> > >
> > > ## Re re partial feedback.
> > >
> > > We feel sorry for causing the misleading in the term "partial feedback", which implies a potential confusion with the term "bandit feedback" in the CBwK literature. In the next version, we will change "partial feedback" to another name. Nevertheless, we should emphasize that we do not **intend** to cause misleading. Actually, we have clarified our information model in the Abstract, Lines 18-20. We further compare them with the bandit feedback model in the Introduction, Lines 80-87. Meanwhile, similar terms are also applied in previous works, e.g., [LG22].
> > >
> > > As you have said, "partial" is just "full" in this work if we do not assume a null-arm. However, an early stop without the null arm may lead to an $O(T)$ regret, and it is common sense to assume the existence of a null arm. We will clarify this in our next version. Furthermore, even though full/partial feedback models are stronger than the bandit feedback, there still lies non-trivial technical challenges to achieve an $\tilde{O}(1)$ regret, as stated in our last reply. On top of that, we believe that our results open the gate for further studies heading to the $\tilde{O}(1)$ regret with bandit information feedback for the CBwK problem. We are also working on this.
> > >
> > > A motivation for the CBwK problem with full/partial information feedback lies in the auto-bidding market with budget constraints. We will go deeper here. In this setting, an auto-bidder competes with the environment in a campaign of $T$ consecutive auctions and is endowed with/constrained by a total budget of $B$. In each auction, given the value **(context) $v_t$**, the auto-bidder chooses a bid **(arm) $b_t$**, and its utility/cost in this auction is further related to **$p_t$ (external factor)**, which is the highest bid among other bidders. Assuming that $v_t$ and $p_t$ are drawn stochastically, the problem becomes a CBwK. Now, for major ad platforms, the information $p_t$ can be revealed to the auto-bidder after each auction or as long as it bids non-zero in the auction. These two scenarios correspond to full/partial information feedback, respectively. The dynamic auto-bidding scenario is a well-established motivation for our information models in practice.
> > >
> > > As for the stronger "full" feedback model you have mentioned in which the information is revealed before each round, we think the problem will become the well-known network revenue management (NRM) by then, as we can deem the context and the external factor as a whole, and there is no further randomness after each round's decision. This model does not capture the dynamic auto-bidding scenario we just discussed. In fact, for the NRM problem, a universal constant regret bound is known, as given by [BW20].
> > >
> > > **Ref:**
> > > [LG22] Heyuan Liu and Paul Grigas. Online contextual decision-making with a smart predict-then-optimize method.
> > > [BW20] Pornpawee Bumpensanti and He Wang. A re-solving heuristic with uniformly bounded loss for network revenue management.

---

> > > ### Author Response · Authors · 2023-08-19
> > > **Response to AC's further questions (part 2)**
> > >
> > > ## Re re parameters on $O(1)$ (full information) and $O(\log T)$ (partial information) regret bounds.
> > >
> > > We think there is some misunderstanding towards our last reply, and we further clarify this.
> > >
> > > The "gap" like parameter for our $O(1)$ (full information) and $O(\log T)$ (partial information) regret bounds is $D$, **the stability factor**, rather than $J(\mathbf{\rho}_1)$. In fact, $J(\mathbf{\rho}_1)$ does not appear in these regret results (Sections 3 and 4), and it **only** lies in the **worst-case** $\tilde{O}(\sqrt{T})$ regret results in Section 5. We now discuss more the stability factor $D$.
> > >
> > > We restate that $D$ is the stability factor that represents the minimum $L_\infty$ distance of the LP $J(\rho_1)$ to any LP with multiple optimal solutions or a degenerate optimal solution. Here, the distance between two LPs is deemed the maximum difference between the two corresponding coefficients. Now, **with no resource constraints, $D$ is precisely twice the gap between the best-arm mean reward and the second-best-arm mean reward**. To see this, the fluid LP has multiple solutions only when there are more than one optimal arms, which directly leads to the above bold claim. Thus, $D$ is exactly the gap-like parameter in this work, similar to $G_{LAG}$ in [SS21].
> > >
> > > Nevertheless, we have to admit that our $O(1)$ (full information) and $O(\log T)$ (partial information) regret bounds goes with $O(1/D^2)$, or $O(1/\text{gap}^2)$. This is not optimal as $O(1/\text{gap})$ but coincides with some NRM results like [CLY22] and the first result given by [SS21]. (They have two results, first with $O(1/G_{LAG}^2)$, and the second with $O(1/G_{LAG})$.) We will further work on improving the gap bound.
> > >
> > > **Ref:**
> > > [CLY22] Guanting Chen, Xiaocheng Li, and Yinyu Ye. An improved analysis of lp-based control for revenue management.
> > >
> > > ## Re re worst-case regret parameters.
> > >
> > > On the $\tilde{O}(\sqrt{T})$ worst-case regret results in Section 5, they relate to public parameters and $J(\mathbf{\rho}_1)$ as the following. We hope these are sufficient:
> > >
> > > - linear with the value $J(\mathbf{\rho}_1)$;
> > > - linear with $\text{NumContexts}$ and $\text{NumResources}$;
> > > - not explicitly related with $\text{NumArms}$.
> > >
> > > ## Re re benchmarks.
> > >
> > > Yes, you are right, and we are sorry to misinterpret your previous claims. [SS21] discussed two benchmarks, one $T\cdot OPT_{FD}$, and one $OPT_{FD}$. Our benchmark is equivalent to $T\cdot OPT_{LP}$, while their $O(\log T)$ upper-bound regrets are achieved w.r.t. to $OPT_{FD}$. And we know $T\cdot OPT_{LP} \geq OPT_{FD}$. In this sense, our upper-bound results are stronger, while lower-bound results are a bit weaker. It is interesting to see the exact relationships between these two benchmarks under different conditions, e.g., Assumption 3.1 in our work, or when $OPT_{LP}$ has a unique but degenerate optimal solution, which is a sufficient condition for $\Omega(\sqrt{T})$ regret in our work.

---

> > > > ### Comment · Area_Chair_Mcfo · 2023-08-20
> > > >
> > > > **Re partial feedback** I think we've clarified things now! Of course, your results remain non-trivial.
> > > >
> > > > **Re null arm:** I'm just saying, one could formally replace the "null arm" with ability to stop early. Simply: simulate an algorithm that uses the null arm, but instead of actually playing it, just advance to the next round. Stop when/if some resource is exhausted.
> > > >
> > > > **Re motivation:** Your autobidding scenario makes sense to me, thanks!
> > > >
> > > > In the revision, it would be useful to cite specific ad markets / platforms for which $p_t$, the highest bid of the other bidders, is actually revealed to the advertisers after each auction. However, I think it is a reasonable enough design choice / motivation even without such examples.
> > > >
> > > > In principle, it is good to have more than one application scenario for a general model such as BwK. Otherwise, it might make more sense to consider a model tailored to this specific application. (Indeed, a number of recent papers focus on designing algorithms for autobidding under constraints.)
> > > >
> > > > **Re gap-like parameter:** Thanks, this explanation helps a lot! Please include it in your revision.
> > > >
> > > > In particular, it is very reassuring that everything comes down to $D$, which generalizes "gap" from stochastic bandits.
> > > >
> > > > To clarify a bit further:
> > > >
> > > > - What form does $D$ take without resources, but with contexts? If there something similar in prior work on contextual bandits?
> > > >
> > > > - Without contexts, has $D$ appeared in some prior work?
> > > >
> > > > **Scaling:** I still wonder how $D$ scales with NumArms, NumContexts, NumResources. For stochastic bandits, one could make such question well-defined as follows: start with just two arms, create $K$ copies of each arm, and ask how regret scales with $K$ (and we get linear scaling in $K$). So, what if we ask a similar question about $D$? Likewise, start with just two resources and then create $d$ copies of each resource; and start with just two contexts, and create $N$ copies of each context. I encourage the authors to try and work out this scaling exercise and include it in the final version.

---

> > > > > ### Author Response · Authors · 2023-08-20
> > > > > **Authors' third response to AC's questions**
> > > > >
> > > > > Dear AC Mcfo:
> > > > >
> > > > > Thanks the third time for your kind reply to our responses. We will now respond to your new questions as much as possible in the current stage. As usual, please feel free to raise further questions to discuss.
> > > > >
> > > > > ## Re re null arm.
> > > > >
> > > > > Yes, we think we get your point here. Nevertheless, we are still unsure whether the "reduction" you mentioned (like the one from PrunedUcbBwK to UcbBwK in [SS21]) does work in the CBwK problem. One primary concern is that all rounds are symmetric in the BwK problem, whereas this is not the case for CBwK due to the **randomness/uncertainty in the context sequence**. To settle our concern, let us say some round $t$ with context $\theta_t$ that our algorithm chooses the null arm. Facing this scenario, we cannot always directly head to the next round since the new context $\theta_{t + 1}$ could be different with $\theta_t$, and the decision mode could correspondingly differ. A choice for round $t$ cannot be well-derived by then.
> > > > >
> > > > > An alternative option of the above, which keeps re-drawing the arm in round $t$ until a non-null arm is chosen, may not work either (but we are not sure). A reason here is that facing some "bad" contexts, the agent should choose not to serve as they will bring small rewards but large consumptions with any non-null action. Struggling to come up with a non-null arm there could occupy the space for serving those good contexts. The struggling process may not even come to an end if the re-solving instructs to choose the null-arm with probability 1 facing a bad context. (For example, for some context, any action will consume a unit resource but brings a zero reward, yet the prior probability of the context is a tiny constant, say 0.01.) Meanwhile, we also do not think a too-early stopping (stops with $O(T)$ rounds remaining) is beneficial since good contexts could arrive late in the sequence. Stopping before they arrive may inevitably cause an $O(T)$ regret.
> > > > >
> > > > > We would like to discuss this sub-topic more if you have further questions.
> > > > >
> > > > > ## Re re motivation.
> > > > >
> > > > > Thanks a lot! We will surely put this auto-bidding motivation in our revision and assist it with appropriate references. We will also work on finding other suitable motivations for our model.
> > > > >
> > > > > ## Re re gap-like parameter.
> > > > >
> > > > > We will surely put the gap-like parameter part in our revision. Besides, we believe that you raised a good thought experiment here. We will now try to answer your questions.
> > > > >
> > > > > - **[Without resources.]** In this case, $D$ becomes half the minimum expected reward gap between the best and second-best arms across all contexts. To be clearer, let $gap(\theta) := \max_a R(\theta, a) - secondmax_a R(\theta, a)$ be the reward gap of context $\theta$, then $D = 1/2 \min_\theta gap(\theta)$. We are not sure whether a similar notion appears in the CB literature, but we will try to find that.
> > > > > - **[Without contexts.]** $D$ is still hard to analyze, even in this case. Nevertheless, with only one resource, the Lagrangian gap $G_{LAG}$ notion in [SS21] could play a similar role with $D$ since they **both come from a view of LP perturbation**. We currently cannot give the exact relationship between these two factors, but we will work on that.
> > > > >
> > > > > ## Re scaling.
> > > > >
> > > > > Thank you for your kind suggestion, and we will start the analysis from the angle you raised. Meanwhile, as a support, the Appendix E of [CLY22] implies that $D$ could take the order of $O(1/(\text{NumContexts}^2\sqrt{\text{NumResources} + \text{NumContexts}}))$, and is not correlated with $\text{NumArms}$. At this stage, we are not sure whether this can be improved.
> > > > >
> > > > > **Ref:**
> > > > > [CLY22] Guanting Chen, Xiaocheng Li, and Yinyu Ye. An improved analysis of lp-based control for revenue management.

---

> > > > > > ### Comment · Area_Chair_Mcfo · 2023-08-21
> > > > > >
> > > > > > We had a nice discussion, NO NEED TO RESPOND FURTHER.
> > > > > >
> > > > > > **Re null arm:** agreed, it is not w.l.o.g. with contexts, at least not obviously so. Thanks for setting me straight!
> > > > > >
> > > > > > So, the presence of contexts makes your "partial feedback" regime a bit more different from "full feedback". You may want to mention this when you discuss your model.
> > > > > >
> > > > > > **Re gap-like parameter:** Thanks for your explanations. I generally think explaining such special cases adds important intuition to your results, when and if.
> > > > > >
> > > > > > For comparison against $O(\log T)$ regret results for contextual bandits: one "easy" way to derive such results is to treat each "policy" (mapping from contexts to arms) as a "super-arm", and talk about the smallest gap between policies. There could be more refined results, I'm not sure. (I recall there were some $O(\log T)$ regret results for _linear_ contextual bandits, but it is unclear to me if they imply anything for your version.) No need to respond to this now, of course.
> > > > > >
> > > > > > **Re scaling:** I took a brief look at App E of [CLY22], but I can't find the technical statement you are referring too. (In fact, I'm not quite sure what is the technical statement that you had in mind -- e.g., note that the scaling exercise I proposed in the previous message is a quite particular one.)
> > > > > >
> > > > > > Regardless, a bound of this form would already  be reassuring, e.g., it rules out exponential dependence.
> > > > > >
> > > > > > Note that you seem to have a rather high dependence on numContexts, one way or another. I mean, even $\sqrt{\text{numContexts}}$ dependence can be prohibitive when contexts are high-dimensional. In other words, you target a regime of relatively small numContexts. This is fine, but worth pointing out.
> > > > > >
> > > > > > A minor point: the dependence on numArms should appear _somewhere_, simply because you'd have $K/\text{gap}$ dependence in the special case of stochastic bandits.

---

> > > > > > > ### Author Response · Authors · 2023-08-21
> > > > > > >
> > > > > > > Dear AC Mcfo,
> > > > > > >
> > > > > > > We received your reply and will revise our work according to your suggestions in the next version. Further, we want to express our sincere gratitude for your precious time, kind patience, and insightful advice on our work.
> > > > > > >
> > > > > > > Best regard,
> > > > > > > Authors of Submission 6087

---

### Author Response · Authors · 2023-08-21

We, the authors, wish to know how reviewer Wp2p and reviewer CJMH regard our rebuttal and whether they have further questions and comments on our response.

---

### Decision · Program_Chairs · 2023-09-21

**Decision:**

Reject

**Comment:**

After substantive with the authors and the reviewers, I think this is a "borderline" paper, both by reviewers' scores and by the substance. While many of the concerns might be addressable via revision and some relatively minor extensions, this is bit too much to take on faith for a conference revision. I'll encourage the authors to revise properly, and resubmit to another conference.

Let me recap the concerns identified in the discussion:

- **Feedback model:** I find it very misleading that the paper talks about CBwK in the title, abstract, and Introduction _as if_ it considers  bandit feedback, even though their "partial feedback" model is much more permissive (namely: full feedback for all non-null arms). This needs to be clarified throughout. In particular, it should not be surprising when/if the positive guarantees in this paper improve over those in prior work on CBwK.

- **Motivation:** Bidding algorithms on ad platforms is the only motivating story so far. It would be good to cite specific ad markets / platforms for which the highest bid of the other bidders  is actually revealed to the advertisers after each auction. Moreover, for a general model such as BwK, it is good to have more than one application scenario. (Otherwise, it may make more sense to consider a model tailored to this specific application, esp. since several recent papers do.)

- **Techniques:** The proposed algorithm seems very similar to that of [Flajolet and Jaillet, 2015]. Essentially, it is unclear how to implement the proposed algorithm, other than running [Flajolet and Jaillet, 2015] with one "super-arm" for each context-arm pair.

- **Dependence on NumContexts:** The algorithm does not extend to large context spaces. First, the regret bounds scale polynomially in NumContexts: there's an explicit linear dependence, and possibly another quadratic dependence implicitly. Second, the running time seems to be linear in NumContexts.

- **Parameters:** The regret bounds hinge upon some parameter $D$ which reduces to gap in the special case of stochastic bandits. The dependence on $D$ is $1/D^2$, as opposed to $1/D$ (which is optimal for bandits, and possible for BwK under some assumptions, e.g., in [SS21]). It is unclear how $D$ scales with NumContexts, NumArms and NumResources, and how this dependence compares to other $O(\log T)$ results in relevant prior work. The following thought experiment could be useful: start with 2 arms and then copy each arm $K$ times, how does the regret bound scale in $K$. Likewise for contexts and resources.

(A few additional concerns are stated in the reviews.)

### Re prior work on CBwK

Also, let me recap some comments re prior work, from the author-AC discussion.

The first paper on CBwK is Badanidiyuru et al (2014).

The distinctions that you bring up in "Related Work"  mirror the distinctions in prior work on CB (i.e., without constraints): namely, CB with classification oracles vs. CB with regression oracles vs. CB with linearity.

For the lower bounds:

- explain how do you get around $\Omega(\sqrt{T})$ lower bounds from [SS21] (as you did in the discussion). Note that

- note that the lower bounds in [SS21] are against a stronger benchmark, namely the expected reward of the best distribution over arms (when this distribution is actually deployed as an algorithm).

For the upper bounds, a few other papers achieve $O(\log T)$ regret in BwK (under different assumptions and parameters):

- Huasen Wu, R. Srikant, Xin Liu, and Chong Jiang. Algorithms with logarithmic or sublinear regret for constrained contextual bandits. NIPS 2015.

- Alberto Vera, Siddhartha Banerjee, and Itai Gurvich. Online allocation and pricing: Constant regret via bellman inequalities. Operations Research, 2020.

- Xiaocheng Li, Chunlin Sun, and Yinyu Ye. The symmetry between arms and knapsacks: A primal-dual approach for bandits with knapsacks. ICML 2021.

Also, several papers achieve $O(\log T)$ regret in BwK problems with only one constraint, including time. Starting with: Andras Gyorgy, Levente Kocsis, Ivett Szab´o, and Csaba Szepesv´ari. Continuous time associative bandit problems. IJCAI 2007.